# Nutrient asymmetry challenges the sustainability of Ukrainian agriculture
Sergiy Medinets [1,2] ✉, Oene Oenema [3], Bryan M. Spears [1], Andriy Buyanovskiy[4],
Volodymyr Medinets [2], William J. Brownlie [1], Eiko Nemitz[1], Massimo Vieno[1] & Mark A. Sutton [1] ✉

The Russian invasion of Ukraine has disrupted crop exports and global food security, overshadowing critical nutrient asymmetry and the associated environmental risks. Here we demonstrate that following nutrient shortages after independence in 1991, fertilizer use increased over 2000-2021, but has decreased sharply following the invasion in early 2022. Input-output balances of nitrogen (N), phosphorus (P) and potassium (K) for staple crops (wheat, maize and sunflower) highlight soil P and K mining since 1991, increasing N surpluses during 2000-2021 and large NPK deficits since the war began in 2022. Based on analysis of five scenarios for 2030, we show how an Integrated Nutrient Management Plan for Ukraine combining manure recycling, precision fertilization and legume expansion is urgently needed, and would maintain crop productivity, significantly reduce nutrient surpluses and improve nutrient use efficiencies up to 80–89%, substantially curtailing environmental pollution and soil degradation.

Ukraine has long been one of the world's leading exporters of wheat, maize and oilseed products. The Russian invasion of Ukraine in 2022 dramatically changed this situation. The invasion disrupted the exports of grain and vegetable oil and increased the prices for fertilizers, energy and shipping, affecting global food insecurity[1–3]. The devastating impact of the large-scale war on Ukraine's agricultural sector extends beyond the export blockage. This is related to a decrease in utilized agricultural areas (Supplementary Fig. 1), damaged infrastructure for processing, storage and irrigation, and destruction of grain stocks[4–6]. While the war has sparked discussions on global food security, energy and fertilizer availability and environmental impacts[2,5–7], to date less attention has been paid to the impacts on the sustainability of crop production in key exporting countries, including Ukraine[8,9]. Large nutrient imbalances may cause severe nutritional, economic and environmental threats, including soil degradation, especially in the long term[10–13], jeopardizing crop yields and food exports (Supplementary Figs. 2-4, Note 1.1). Here, we trace nutrient management during Ukraine's transition from the Soviet era (from 1980) through the challenges of independence (since 1991), culminating in the pre-war period of high agricultural productivity (2019-2021), and through the first two years of war (2022-2023).

Using Ukraine as a case study, we analyze the impacts of nutrient imbalances, including, to the best of our knowledge, the first county-to-national scale assessment of how war has disrupted nutrient cycles in a major breadbasket country, with implications for global food security. We

also assess five forward-looking nutrient management scenarios to inform policy for sustainable recovery by 2030, while also warning of the potential consequences if urgent action is not taken. The analysis focuses on the three staple crops (wheat, maize and sunflower) collectively covering more than 67% of Ukraine's total utilized agro-area in 2021[14].

## Results and Discussion

### Input-output nutrient balances and use efficiencies

Following Ukraine's independence in 1991, use of nitrogen (N), phosphorus (P) and potassium (K) fertilizers and numbers of livestock decreased dramatically, mainly due to the disruption of supply and marketing chains[15] (Fig. 1; Supplementary Fig. 6). Synthetic fertilizer N use gradually increased again from early 2000s following economic development, but has reduced significantly since the Russian invasion in 2022 (Fig. 1; Supplementary Figs. 7–9). Mean total N fertilizer use increased by 18-fold for sunflower, 8-fold for maize and 5-fold for wheat between 2000 and 2021, reaching 97%, 72% and 108% of the 1990 (pre-independence) levels, respectively (Fig. 1). As of 2018, Ukraine had the second highest synthetic N fertilizer rate per hectare of the world's major countries exporting wheat (after the EU) and maize (after the US) (Supplementary Figs. 2c, 4c).

Livestock numbers decreased substantially after 1990[14] (Supplementary Fig. 6). As a result, manure production was much lower than in the pre-independence period. Manure N application to cropland was also hindered by the decoupling of crop and livestock production systems through farm

[1]UK Centre for Ecology and Hydrology, Edinburgh, UK. [2]Regional Centre for Integrated Environmental Monitoring, Odesa National I. I. Mechnikov University, Odesa, Ukraine. [3]Wageningen University & Research, Wageningen, The Netherlands. [4]Department of Geography of Ukraine, Soil Science and Land Cadastre, Faculty of Geology and Geography, Odesa National I. I. Mechnikov University, Odesa, Ukraine. ✉e-mail: sermed@ceh.ac.uk; ms@ceh.ac.uk

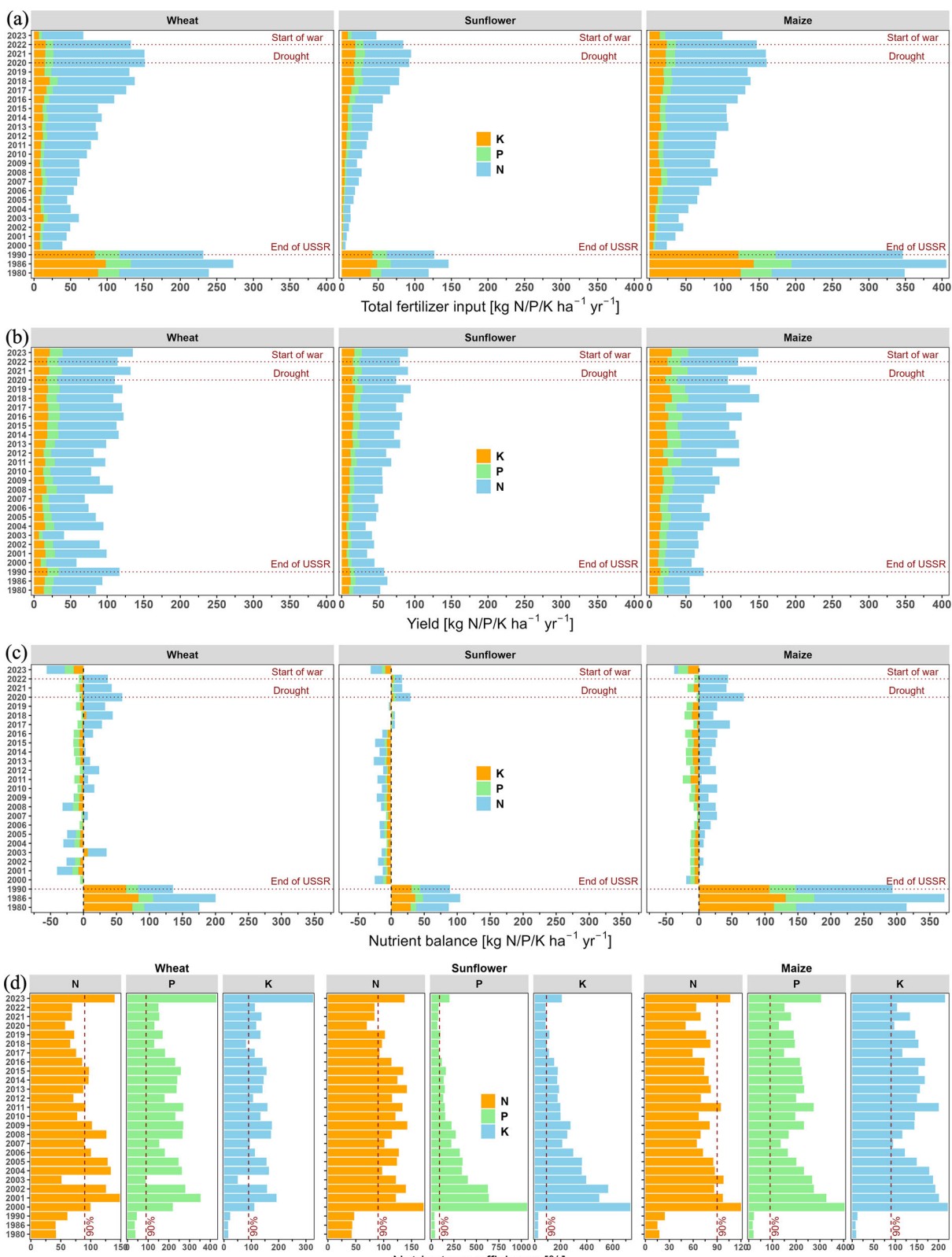

**Fig. 1 | Average annual N, P and K fertilizer inputs, yields, nutrient balances and nutrient use efficiencies for wheat, sunflower and maize crops in Ukraine over 1980, 1986, 1990 and 2000–2023.** Fertilizer inputs (**a**) show the combined amounts of synthetic and organic NPK applied; yields (**b**) represent harvested crops removed from the field; nutrient balances (**c**) indicate the difference between nutrients in inputs and those in harvested yields (see "Methods" for details); nutrient use efficiencies (**d**) are the ratios of nutrients in harvested yield to nutrient inputs (see "Methods" for details). Annual inorganic N deposition (Supplementary Fig. 5) and

annual N fixation by free-living organisms (assumed to be 5 kg N ha$^{-1}$ yr$^{-1}$)[37], both not shown in this figure, were used as N inputs for N balance and N use efficiency calculations (see "Methods" for details). To avoid risk of soil nutrient mining, 90% fertilizer use efficiency is shown as an illustrative benchmark target (see "Methods" for detail). Horizontal dashed lines represent the final year of USSR era (1990), a distinct drought year (2020) and the beginning of the large-scale war in Ukraine (2022).

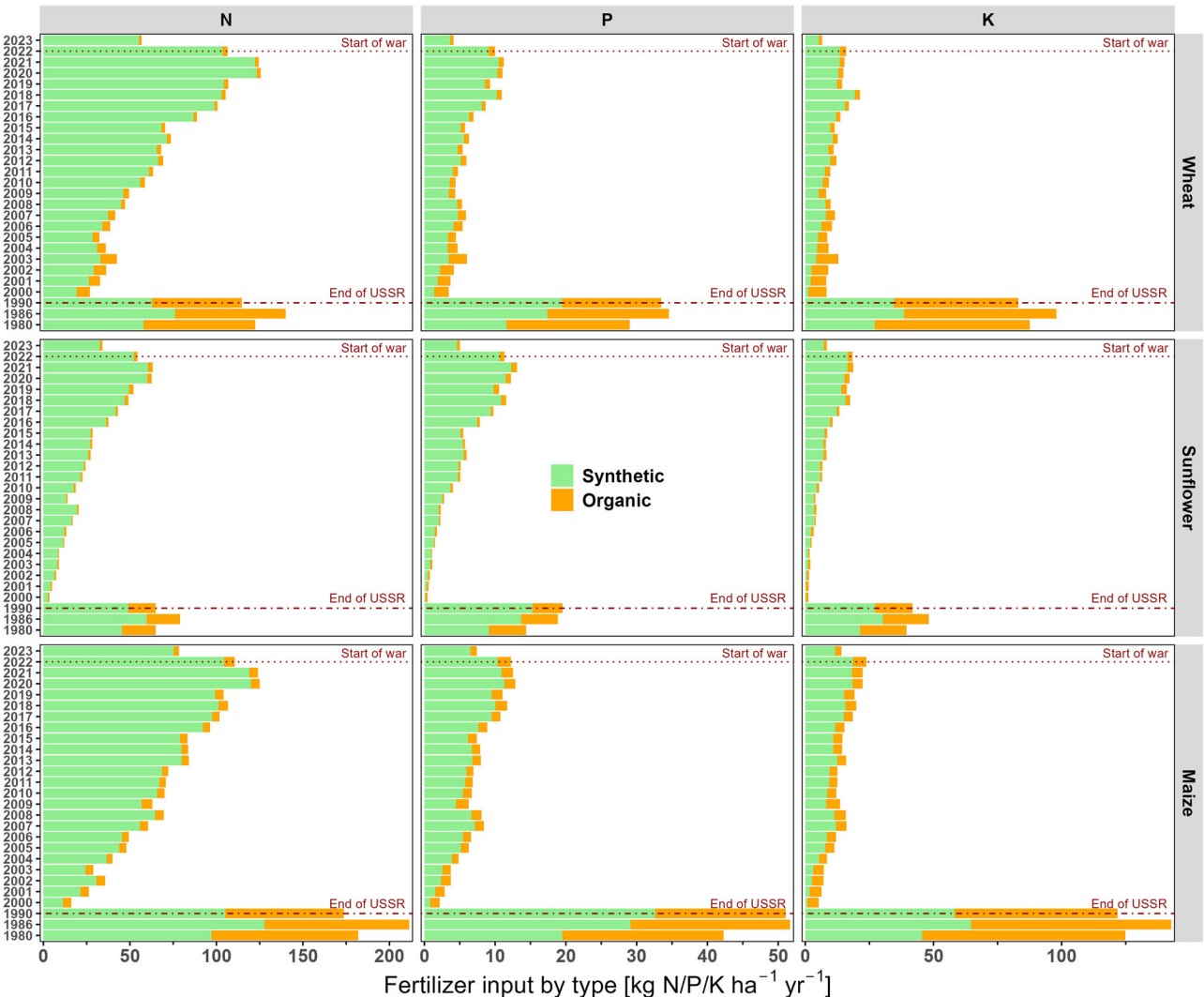

**Fig. 2 | Average annual synthetic and organic N, P, and K fertilizer application to wheat, sunflower, and maize in Ukraine during 1980, 1986, 1990, and 2000-2023.** Dotted lines represent the beginning of the large-scale war in Ukraine (2022); dash-dotted lines represent the final year of the USSR era (1990).

specialization[15]. Our estimates indicate that the fraction of manure N excreted applied to cropland decreased from around 34% in 1986-1990 to around 10% in 2021 (Supplementary Fig. 10). As a result, manure N (including other organic fertilizers) contributed only 2–4% to the total fertilizer N input to the studied crops (wheat, maize and sunflower) in 2020-2021, while this share was 24-53% during 1980-1990 (Fig. 2; Supplementary Fig. 11, Note 2.1). We estimate the economic loss from unutilized manure nutrients at around USD 2.2 billion in agriculture across Ukraine in 2021 (Fig. 3; Supplementary Note 2.1).

Following the dramatic reduction in total fertilizer use after 1990, NPK withdrawal in harvested wheat, maize, and sunflower decreased only modestly during the first ten years. Fertilizer use and NPK withdrawal in harvested crops increased again during the 2000-2021 period (Fig. 1). We attribute the modest initial decline in yields under NPK input deficits during 1990–2000 to compensation by soil nutrient mining, i.e. enhanced mineralization of organic matter reserves inherent to the rich black soils (Chernozems)[8,15]. Persistent nutrient-deficient soil management practices led to continuous depletion of soil organic matter (SOM) across Ukraine, resulting in a measurable 5–6% reduction in SOM content over that decade (Supplementary Fig. 12, Note 2.2–2.3). The increasing crop yields during the period 2000–2021 are probably due to a combination of increased fertilizer use, together with improved crop varieties and crop husbandry practices. Ukraine ranked second (after the EU) in wheat yield per hectare and third

(after the US and Argentina) in maize yield over 2019–2021 (Supplementary Figs. 2b, 4b).

Balances of NPK inputs and outputs for wheat, maize, and sunflower revealed continuous deficits for P and K and variable deficits/surpluses for N for these crops during 2000–2023 (Fig. 1). Sunflower tended to have continuous N deficits, while maize (since 2001) tended to have continuous surpluses. Wheat had relatively large N deficits in the early 2000s and surpluses from 2010 onward, increasing steadily from 2016. We estimate N surpluses of up to 59 and 68 kg N ha$^{-1}$ yr$^{-1}$ for wheat and maize, respectively, in 2020, which was a dry year (Fig. 1). The resulting economic loss associated with each 10 kg N surplus ha$^{-1}$ for the combined area of the three crops in Ukraine was around USD 524 million annually (Supplementary Note 2.1). The actual surpluses were higher if the contribution of organic N deposition from the atmosphere is considered considered[16–18] (see Methods for details). The combined NPK deficits in 2023 for the three crops exceeded those of any year since 2000 (Fig. 1; Supplementary Note 2.4).

We thus see a major asymmetry in Ukrainian agricultural nutrient balances at the country scale, with recent surpluses for N in wheat and substantial continuous deficits for both P and K, followed by the pervading NPK deficits in 2023. By asymmetry in this context, we mean the combination of imbalances (including spatially), both between inputs and outputs (including demand by different crops) and between different nutrients, leading to low nutrient use efficiency. As seen in Fig. 4, we find large

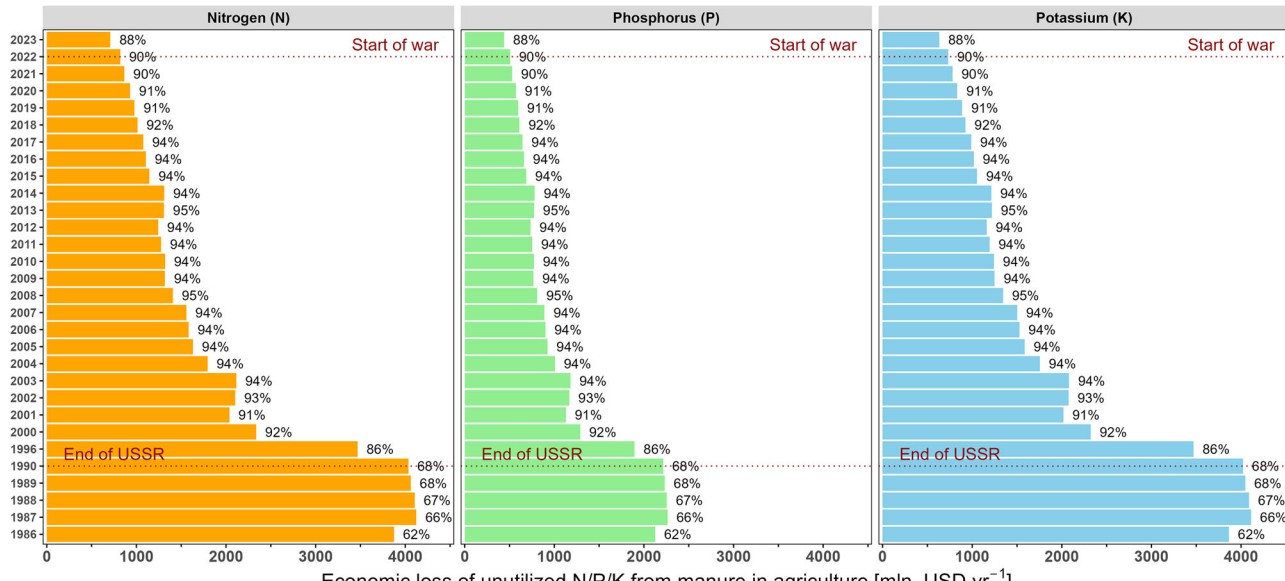

**Fig. 3 | Estimated annual economic loss from unutilized manure nutrients in Ukrainian agriculture from 1986 to 2023 (colored bars).** Values are in million USD per year, based on the average price of one kg of N, P and K in synthetic fertilizers as of June 2024 (see Supplementary Note 2.1). Dotted lines represent the final year of USSR era (1990) and the beginning of the large-scale war in Ukraine (2022). The percentage of unutilized manure is also shown next to each bar.

**Fig. 4 | Average annual N, P and K input-output balances for wheat, sunflower and maize per county in Ukraine over 2019-2021, pre-war years.** Positive values indicate nutrient surplus liable to be lost; Negative values indicate nutrient deficit. Annual inorganic N deposition (Supplementary Fig. 5) and annual N fixation by free-living organisms (assumed to be 5 kg N ha$^{-1}$ yr$^{-1}$)[37], both not shown in this figure, were used as N inputs for N balance calculations (see Methods for details). Grey-filled counties indicate no data; see Supplementary Table 1 and Fig. 17 for county details.

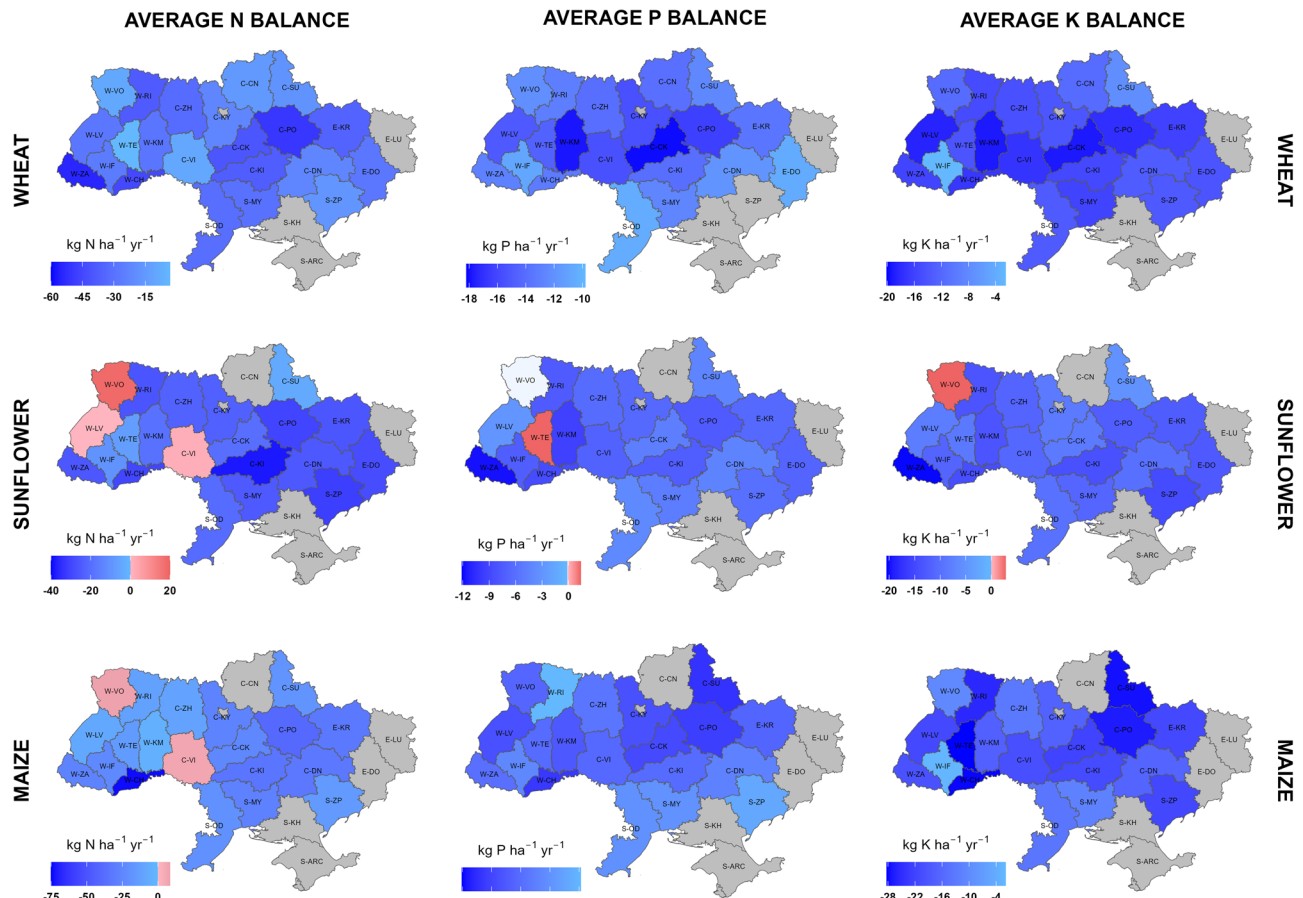

**Fig. 5 | Average annual N, P, and K input-output balances for wheat, sunflower, and maize per county in Ukraine during 2023, the 2nd year of the large-scale war.** Positive values indicate nutrient surplus liable to be lost; Negative values indicate nutrient deficit. Annual inorganic N deposition (Supplementary Fig. 5) and annual N fixation by free-living organisms (assumed to be 5 kg N ha$^{-1}$ yr$^{-1}$)[37], both not shown in this figure, were used as N inputs for N balance calculations (see Methods for details). Grey-filled counties indicate no data or excluded data (see Methods for details); see Supplementary Table 1 and Fig. 17 for county details.

variations in nutrient balances between counties, reflecting large differences between counties in NPK applications, often with asymmetric responses in yields (Supplementary Figs. 14–16, Note 2.5).

During the mid-1980s, NPK use efficiencies for most crops were mostly ≤50% due to excessive NPK inputs linked to subsidized synthetic fertilizers[8,15] and the application of relatively large amounts of animal manure, which was available in much greater quantities than currently (Fig. 2; Supplementary Fig. 7). During the 1990s - early 2000s inputs of NPK via synthetic and organic (largely manure) fertilizers greatly decreased, mainly due to the disruption of supply and marketing chains, while the crops largely relied on soil nutrient stocks[15], leading to soil degradation (Supplementary Fig. 12, Note 2.2–2.3). As a results, fertilizer NPK use efficiencies exceeded 100% during this period. Later, NPK use efficiencies of maize (since 2004) and wheat (since 2010) decreased again but remained variable for N (60-97%) and well above 100% for P and K; the opposite tendency was found for sunflower (Fig. 1). Thus, NPK use efficiencies surpassing 90-100% indicate continuous soil nutrient depletion[19], particularly for K and P, but also for N, as evidenced by steady decline in soil organic matter content during this period (Fig. 1; Supplementary Fig. 12, Note 2.6). The spatial patterns of nutrient use efficiencies varied significantly across crops and between major georegions of Ukraine (Supplementary Figs. 18–20, Note 2.7).

### Impact of the war on nutrient balances and use efficiencies
Following the Russian invasion of Ukraine on 24 February 2022, slight decreases during 2022 were observed in fertilizer application and crop production. This limited immediate impact could be attributed to the resilience of fertilizer and agrochemical supply chains, which benefited from

advanced planning and restocking routines typically completed by late winter/ early spring. Consequently, fertilizers for winter/ spring crops contracted in 2021 were largely accumulated in warehouses in Ukraine and/ or even delivered to farmers despite the onset of hostilities. However, the impact of the war was larger in 2023. Total fertilizer applications decreased by 37-54% for the three crops compared with 2021, while the utilized agro-area decreased by 22-34%. Despite this, favourable weather conditions, enhanced soil mineralization, and, potentially, residual soil nutrients from preceding crops collectively contributed to relatively high yields in 2023, comparable to those in 2021. Conversely, this led to high nutrient use efficiencies, but exacerbated soil nutrient mining to levels greater than the early 2000s (Fig. 1; Supplementary Figs. 21-23). The impact has been uneven across the country, with higher soil N depletion observed in central, eastern, and south-western counties (Fig. 5). We anticipate a substantial further increase in NPK deficits, leading to soil nutrient mining in Ukrainian agriculture, ultimately degrading soils and decreasing yields if the war continues (see scenario S-w below). This indicates the need for farmers to be supported with economic and technical assistance to prevent an eventual collapse of food production.

### Implications for food security
The evolving nutrient management landscape in Ukraine holds profound implications not only for national food security but also for the stability of the global food supply chain. As one of the world's top exporters of wheat, maize and sunflower oil, Ukraine plays a critical role in feeding over 400 million people worldwide, particularly in import-dependent regions[20,21] (Supplementary Figs. 2–4, Note 1.1). However, the ongoing war has

disrupted fertilizer supply chains and reduced agricultural inputs, aggravating imbalances of N, P and K (Figs. 1, 5; Supplementary Fig. 8). These imbalances are marked by spatial and crop-type asymmetry in N (surpluses in some areas, and deficits in others) and persistent deficits in P and K (Figs. 4, 5). The resulting degradation of Ukraine's Chernozem soils (Supplementary Fig. 12, Note 2.2), is crucial as these store about 7% of the world's soil carbon and underpin around 90% of the country's agricultural output[14,22]. Without immediate intervention, this trend could trigger substantial losses of soil organic matter, especially in the regions affected by the war, risking long-term productivity declines for key crops, such as wheat and maize[23].

Given that Ukrainian grain exports traditionally support food security in regions such as North Africa and the Middle East (Supplementary Fig. 3), disruptions in crop output could exacerbate hunger and price volatility worldwide[20,21]. The ripple effects are global: disruptions in Ukrainian grain exports during the early months of the war contributed to a sharp rise in global food prices, with the FAO Food Price Index reaching its highest level on record in 2022, over 14% higher than the previous year[24]. This surge intensified global food insecurity, contributing to a wider hunger crisis with a global estimate of 691–783 million people affected in 2022[25]. Moreover, the under-utilization of manure, caused by the decoupling of livestock and crop systems, represents a missed opportunity for circular nutrient economy practices (Figs. 2, 3; Supplementary Figs. 10, 11, Note 2.1) that could reduce dependence on synthetic fertilizers while lowering reactive N and greenhouse gas emissions. Addressing these challenges requires improved access to fertilizers and long-term investments in integrated nutrient management strategies. Strengthening Ukraine's nutrient resilience is not merely a national priority but a global imperative when considering the global food security and climate interactions.

## Nutrient management scenarios

We developed forward-looking nutrient management scenarios to project how Ukraine's nutrient balances for major crops (wheat, maize, sunflower) might evolve by 2030 (Table 1, Fig. 6; Supplementary Tables 2, 3, Figs. 24, 25). These scenarios highlight both challenges and opportunities for Ukraine to achieve agricultural sustainability (see details in Supplementary Note 2.8). The scenarios include:

S-0: **Business-as-usual (BaU)**, maintaining 2021 practices prior to the invasion of Ukraine;

S-w: **Extended war disruption scenario**, assuming prolonged fertilizer shortages at 2023 levels;

S-1: **Manure-enriched precision fertilization**, replacing synthetic N with manure-N of an increased 30% share;

S-2: **Enhanced-efficiency fertilizers (EEFs)**, building upon S-1;

S-3: **Legume-based diversification**, in combination with S-2 or S-1.

In **the BaU scenario (S-0)**, substantial nutrient imbalances persist with annual N surpluses ($\sim$42 kg N ha$^{-1}$ yr$^{-1}$ for wheat and maize), causing cumulative losses of $\sim$4 million tonnes (Mt) N total over 2024-2030 across these crops growing areas in Ukraine (Table 1). Simultaneously, P and K deficits (6–10 kg ha$^{-1}$ yr$^{-1}$) lead to cumulative soil mining of $\sim$0.1 Mt of each by 2030, gradually reducing soil fertility (Supplementary Tables 2, 3, Note 2.8). Sunflower systems, nearly balanced in 2021, remain less problematic.

**The extended war disruption scenario (S-w)** severely escalates nutrient depletion, projecting cumulative deficits of 1.7 Mt N, 2.8 Mt P and 2.1 Mt K for the three crop growing areas over 2024-2030 (Table 1; Supplementary Tables 2, 3, Note 2.8). Such extreme nutrient mining risks irreversible damage to Ukraine's Chernozem soils, depleting soil organic matter and lowering productivity, echoing degradation patterns of the 1990s[15,26,27]. This scenario emphasizes the critical need for immediate interventions to sustain national and global food security.

By contrast, the three **sustainable nutrient management scenarios** demonstrate achievable improvements (Fig. 6; Supplementary Figs. 24, 25). **Scenario S-1** involves substituting part of synthetic N fertilizer with manure-N, reaching a 30% share, combined with precision fertilization strategies

(e.g., guided by a Smart Fertilizer Planner; Supplementary Note 2.9), reducing total N inputs (through synthetic N) by 10% for maize and wheat, and 5% for sunflower. By 2030, S-1 projects a 37% synthetic N reduction, a 28% increase in manure-N, halving N surpluses and approaching balanced P and K applications. This scenario envisages a significant reduction of N losses: ammonia (NH$_3$) reduce by $\sim$49%, nitrous oxide (N$_2$O) by $\sim$13% and N runoff by 50–67%, with potential yield increases of $\sim$5%, compared to BaU (S-0). Nutrient use efficiencies would increase substantially, reaching 76–84% for N and 79–87% for P and K, alongside improved soil organic matter and resilience (Table 1; Supplementary Tables 2, 3, Note 2.8).

**Scenario S-2** builds on S-1 by integrating EEFs, allowing an additional 10% reduction in total N (through synthetic N) without yield loss. By 2030, N surpluses would decrease further to 9–17 kg N ha$^{-1}$ yr$^{-1}$, elevating NUE to 85–88%, with cumulative emissions reduced significantly: NH$_3$ ($\sim$77%), N$_2$O ($\sim$48%) and nitric oxide (NO, $\sim$60%) compared to BaU (Table 1; Supplementary Tables 2, 3, Note 2.8). This substantially enhances air quality and supports climate mitigation.

**Scenario S-3**, proposes expanding legumes to 20% of grain land, leveraging biological N fixation (130–150 kg N ha$^{-1}$ yr$^{-1}$, on average[28]). This strategy could deliver an estimated 40–50 kg N ha$^{-1}$ benefit to subsequent cereals, reducing total N inputs (through synthetic N) by an additional $\sim$15%. When combined with S-1 or S-2, this results in a 52–61% reduction in synthetic N input relative to BaU by 2030, with an expectation of maintaining the same level of yields as BAU but with higher nutrient use efficiency (Table 1; Supplementary Note 2.8).

Collectively, these scenarios show how Ukraine's nutrient asymmetry can be addressed. By adopting sustainable nutrient management practices (S-1, S-2, S-3), Ukraine could maintain crop productivity, significantly reduce nutrient surpluses and improve nutrient use efficiencies to 80–89%, substantially curtailing environmental pollution and soil degradation (Fig. 3, Table 1; Supplementary Figs. 24, 25, Tables 2, 3). The urgency of transitioning from fertilizer dependence, intensified by war-induced disruptions, underscores the need for immediate policy actions and investment toward sustainable agricultural recovery (see below).

## Current challenges and urgent actions

Based on our analyses, there is strong evidence of imbalanced nutrient use in Ukraine, which is greatly exacerbated by the current Russo-Ukrainian war. The increased asymmetry between nutrient inputs and outputs, and between N, P, and K, is reducing the sustainability of crop production, while also jeopardizing the economy, environment, and future global food security. These changes originate in part also from the hundreds of thousands of casualties in Ukraine, as well as around 5 million internally displaced Ukrainians, and loss of 20% of the population who have fled the country[29]. Aside from the obvious humanitarian crisis, these human disturbances limit the ability to focus on sustainable practices. Recognizing these challenges, the observed nutrient asymmetry highlights the need to develop an Integrated Nutrient Management Plan for Ukraine, which can also have relevant messages for other countries. To support this, our analysis of scenarios points toward the following priority measures for sustainable recovery, which would also improve farm profitability:

- Adoption of precision fertilization principles for crop nutrition based on crop requirements and the '5R approach' (right source, rate, application time, application place, and application method) at the field-to-farm scale is the foundational measure underpinning all sustainable practices, as demonstrated in scenarios S-1, S-2, and S-3 (Table 1; Supplementary Tables 2, 3, Note 2.8). This approach should be supported/guided by a simple, farmer-friendly, region-tailored, crop variety-specific Smart Fertilizer Planner tool to estimate NPK input requirements based on crop yield of top-performing farms in a region over the past 3-5 seasons (Supplementary Note 2.9). After harvest, this tool would help calculate annual field-scale nutrient balances and estimate nutrient use efficiency, to guide actions for the following season.

**Table 1 | Summary of nitrogen (N) inputs, outputs, balances and N use efficiencies (NUE) for wheat, maize and sunflower under five nutrient management projections by 2030 alongside cumulative N balances over 2024–2030 for business-as-usual (BaU, S-0) and extended war disruption (S-w) scenarios, as well as scenarios for emission reductions through manure-enriched precision fertilization (S-1), enhanced-efficiency fertilizers (S-2) and legume-based diversification (S-3), as compared with the BaU (S-0) scenario**

| Crop | Scenario | N input, kg N ha⁻¹ yr⁻¹ | | | | | | N output, kg N ha⁻¹ yr⁻¹ | N balance, kg N ha⁻¹ yr⁻¹ | | N surplus/deficit, % | | NUE, % | | Cumulative balance by 2030, Mt N | Emission reduction, % | | |
| | | Total | Total org. | Crop-available | | | Total | | Total | Crop-available | Total | Crop-available | Total | Crop-available | | $NH_3$ | $N_2O$ | NO |
| | | | | Org. | Synth. | BNF (legumes) | | | | | | | | | | | | | |
| Wheat | S-0 | 136.7 | 2.2 | 0.9 | 122.3 | - | 135.4 | 93.9 | 42.8 | 41.5 | 31.3 | 30.6 | 68.7 | 69.4 | 2.1 | 0 | 0 | 0 |
| | S-1 | 123.0 | 33.0 | 19.8 | 77.0 | - | 109.8 | 93.9 | 29.1 | 15.9 | 23.7 | 14.5 | 76.3 | 85.5 | - | 49 | 13 | n/a |
| | S-2 | 110.7 | 33.0 | 19.8 | 64.7 | - | 97.5 | 93.9 | 16.8 | 3.6 | 15.2 | 3.7 | 84.8 | 96.3 | - | 77 | 48 | 60 |
| | S-3 | 109.0 | 33.0 | 19.8 | 48.1 | ~15 | 95.9 | 93.9 | 15.2 | 2.0 | 13.9 | 2.5 | 86.1 | 97.9 | - | ≥77 | ≥48 | ≥60 |
| | S-w | 66.9 | 1.5 | 0.9 | 55.3 | - | 66.3 | 93.9 | -27.0 | -27.6 | -40.4 | -41.7 | 140 | 142 | -0.9 | - | - | - |
| Maize | S-0 | 136.4 | 5.3 | 3.2 | 118.9 | - | 134.3 | 94.5 | 41.9 | 39.8 | 30.7 | 29.6 | 69.3 | 70.4 | 2.0 | 0 | 0 | 0 |
| | S-1 | 122.7 | 32.1 | 19.3 | 74.9 | - | 109.9 | 94.5 | 28.2 | 15.4 | 23.0 | 14.0 | 77.0 | 86.0 | - | 49 | 13 | n/a |
| | S-2 | 110.4 | 32.1 | 19.3 | 62.6 | - | 97.6 | 94.5 | 15.9 | 3.1 | 14.4 | 3.1 | 85.6 | 96.9 | - | 77 | 48 | 60 |
| | S-3 | 108.8 | 32.1 | 19.3 | 46.1 | ~15 | 96.0 | 94.5 | 14.3 | 1.5 | 13.2 | 1.9 | 86.8 | 98.4 | - | ≥77 | ≥48 | ≥60 |
| | S-w | 90.2 | 3.0 | 1.8 | 75.4 | - | 89.0 | 96.0 | -5.8 | -7.0 | -6.4 | -7.9 | 106 | 108 | -0.2 | - | - | - |
| Sunflower | S-0 | 75.4 | 2.7 | 1.6 | 60.5 | - | 74.3 | 63.0 | 12.4 | 11.3 | 16.4 | 15.2 | 83.6 | 84.8 | 0.5 | 0 | 0 | 0 |
| | S-1 | 75.4 | 16.3 | 9.8 | 38.1 | - | 68.9 | 63.0 | 12.4 | 5.9 | 16.4 | 8.5 | 83.6 | 91.5 | - | 48 | 13 | n/a |
| | S-2 | 71.6 | 16.3 | 9.8 | 34.3 | - | 65.1 | 63.0 | 8.6 | 2.1 | 12.0 | 3.2 | 88.0 | 96.8 | - | 77 | 48 | 60 |
| | S-3 | 70.9 | 16.3 | 9.8 | 23.6 | ~10 | 64.4 | 63.0 | 7.9 | 1.4 | 11.1 | 2.5 | 88.9 | 97.9 | - | ≥77 | ≥48 | ≥60 |
| | S-w | 45.8 | 1.5 | 0.9 | 32.6 | - | 45.2 | 63.0 | -17.2 | -17.8 | -37.6 | -39.3 | 138 | 139 | -0.6 | - | - | - |

'Total N input' includes synthetic N fertilizer, organic (manure) N fertilizer, atmospheric N deposition, and N fixation (see Methods). 'Org.' represents manure-N; 'Synth.' represents synthetic N fertilizer; BNF (legumes)' denotes the average annual input from biological N fixation by legumes in a four-year crop rotation, where legumes are grown in one of the years preceding cereals, with sunflower often serving as the final crop in the rotation; 'Crop available' refers to N available for crop uptake within year of application (and previous-year input from manure-N; see Supplementary Note 2.8 for details). 'Mt N' represents million tonnes of N. Full descriptions of the scenarios are given in Supplementary Note 2.8. Values of key categories/parameters are highlighted in bold to facilitate reader recognition.

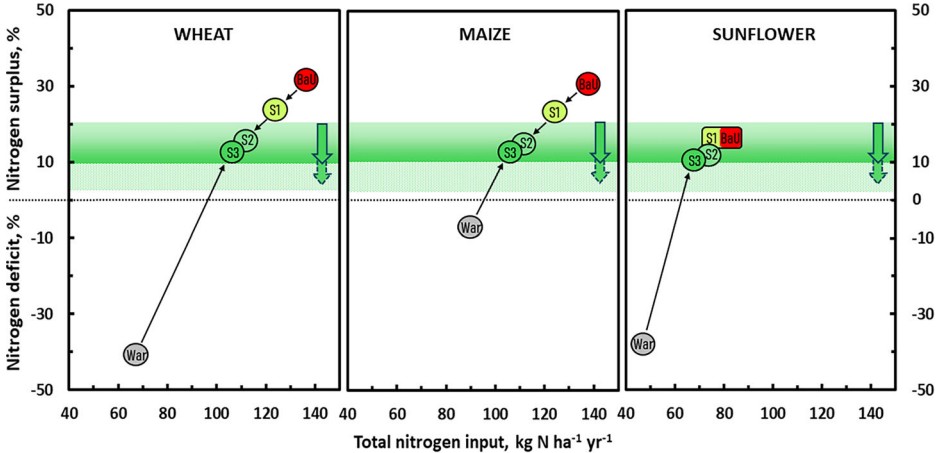

**Fig. 6 | Average nitrogen (N) surplus or deficit as a percentage of total N input for wheat, maize, and sunflower in Ukraine under five contrasting scenarios for 2030.** The business-as-usual scenario (BaU; S-0) reflects the continuation of 2021 agricultural practices prior to the Russian invasion. The extended war disruption scenario (War; S-w) assumes prolonged fertilizer shortages at 2023 levels. The manure-enriched precision fertilizer scenario (S1; S-1) involves the substitution of synthetic N with manure-N, increasing its share by 30%, combined with precision fertilizer application. The enhanced efficiency fertilizer scenario (S2; S-2) builds upon S1 by incorporating nitrification inhibitors and slow-release fertilizers. The legume-based diversification scenario (S3; S-3) introduces optimized crop rotations with legumes, in combination with S2 or S1 (see Nutrient Management Scenarios). Scenario colours indicate relative N emission reductions: red (BaU) denotes no reduction, a green gradient from S1 to S3 reflects stepwise reductions, and grey indicates the situation under the War (see Table 1; Supplementary Note 2.8). The green-shaded belt represents the acceptable N surplus range, between 10% and 20% of total N input[38]. The green-dotted belt indicates further reductions from 10% toward the minimally unavoidable N loss, expected to decline over time with improved agricultural practices. Arrows indicate the direction of surplus reduction. The dotted horizontal line marks zero N balance (equilibrium) with surpluses above and deficits below this threshold.

- Increasing manure-N recycling by 30% by 2030 is a top-priority, multi-benefit measure. When combined with precision fertilization, as shown in scenario S-1, it could lead to a 37% reduction in synthetic N use, partly through substitution with manure-N, while halving N surpluses, significantly mitigating direct N emissions (including from previously wasted manure), and helping to balance P and K applications compared to BaU (scenario S-0). This measure also boosts nutrient use efficiency (76–87%) and enhances soil organic matter and resilience (Table 1; Supplementary Tables 2, 3, Note 2.8). However, successful implementation requires government support through the development and strong coordination of a national programme to reintegrate animal husbandry and cropping systems. This could include manure–fodder exchanges between livestock and arable farms and increased use of processed manure, supported by investments in community-level manure management infrastructure. As a first step, a nationwide inventory of manure management practices should be conducted across georegions to improve manure collection, promote low-emission storage on livestock farms, and ensure complete recycling of manure nutrients onto croplands with minimal environmental pollution (see Supplementary Fig. 10, Note 2.1).
- Application of EEFs, including urease and/or nitrification inhibitors as well as slow-release formulations, is a targeted measure aimed at maximizing environmental benefits by mitigating N pollution. When combined with the previous two measures (as illustrated in scenario S-3), EEFs could significantly reduce N emissions by 48–64% and increase NUE to 85–88% (Table 1; Supplementary Tables 2, 3, Note 2.8), e.g., replacing conventional urea with slow-release fertilizers offers a practical and immediately actionable step toward achieving these outcomes.
- Diversification of crop rotation with N-fixing leguminous crops (including cover crops) is a conventional yet actionable strategy to reduce N fertilizer inputs and deliver broader benefits for subsequent crops, such as improved soil quality, disruption of pest/disease cycles, enhanced P availability and potential yield gains. In scenario S-3, we showed that when combined with manure-enriched precision fertilization (S-1) and/or EEFs (S-2), this approach can achieve a substantial 52–61% reduction in synthetic N use relative to the BaU

(S-0) scenario by 2030. Although potentially less profitable for farmers in the short term, practices such as cover cropping for green manure are likely to require subsidies to support widespread adoption (Table 1; Supplementary Note 2.8).

In addition, the establishment of regional advisory services to support the implementation of the outlined measures and broader Good Agricultural Practices at the field-to-farm level should be incorporated into an Integrated Nutrient Management Plan for Ukraine. These would raise farmers' awareness of nutrient management issues by offering education and knowledge transfer, with tailored, region-specific strategies supported by simplified cost-benefit analyses and providing the required information for the Smart Fertilizer Planner. Accompanying grants and loans to farmers for investment in nutrient management would catalyze change that can ultimately become self-sustaining (Supplementary Note 2.10).

Developing such a plan, evidence-backed by our scenario analysis, should not wait until the war is over. While the war makes conditions harder, all the measures listed could already start with appropriate investment. Indeed, there is now an urgent need for strengthening international action to support Ukrainian agriculture and prevent further depletion of precious nutrients with the accompanying soil degradation, while recognizing that this must be the start of a long-term commitment to supporting a transition to sustainable nutrient management. We propose that an Integrated Nutrient Management Plan for Ukraine be incorporated into the agriculture sector recovery under the Ukraine Recovery and Reconstruction (URR) budget, which currently accounts for 10.5% of the total planned URR budget of USD 524 billion[30]. We also recommend prioritizing agro-food system resilience as a key focus within the URR framework.

The actions listed are not only a priority for the Ukrainian economy, but are needed to maintain continued exports internationally, global food security, and environmental sustainability, contributing to SDG Goal 2: Zero Hunger by 2030 and supporting the transition to a circular and net-zero economy by combating nutrient pollution and reducing greenhouse gas emissions. Addressing this wider challenge is a key opportunity for United Nations-affiliated bodies such as the International Nitrogen Management System[11], the UNEP Nitrogen Working Group, and the Global

Partnership on Nutrient Management, in cooperation with the FAO, where a stronger coordination of effort is needed that links management of N, P, K and other nutrients. International action would help mobilize cross-sectoral intergovernmental policy on nutrients, accelerating the transition to a circular economy. If progress is to be made towards the Colombo Declaration[31] and Target 7 of the Global Biodiversity Framework[32], to at least halve nitrogen and wider nutrient pollution by 2030, such international support to Ukraine could become a beacon to guide necessary actions globally.

## Methods

### Agricultural data

We analyzed Ukraine's national statistical data[14] to estimate NPK balances of utilized cropland and fertilizer-derived nutrient use efficiencies of cropland used for the production of wheat, maize, and sunflower; crops grown for forage were not included in our analysis. The focus was on the period from 1980 to 2023, which includes the years of the large-scale war impact (2022-2023) and the last 10 years of the pre-independence (1980-1990) period. We used annual statistical data (yield, synthetic and organic fertilizer use per crop) reported by SSSU[14] to calculate average synthetic and organic NPK fertilizer inputs as well as yield per hectare of the utilized agricultural area (UAA) allocated to each respective crop. Where county-level data were available (since 2007 for sunflower and maize, and since 2009 for wheat), we used these to calculate UAA-weighted annual magnitudes for each respective crop (see Supplementary Note 3.1). Since 2014, national statistics have not included areas that were temporarily occupied or annexed by the Russian Federation. Between 2014 and 2021, the occupied territories varied but encompassed approximately 30% of Donetsk and 50% of Luhansk counties. Statistical data from the remaining areas under Ukrainian control were collected and published in full. The Autonomous Republic of Crimea was annexed in 2014, and no official data have been available from this region since then. Following the large-scale Russian invasion on 24th February, 2022, national statistics have ceased data collection in areas once they fall under Russian control and resume collection once those areas are liberated. All regions remaining under Ukrainian control have continued regular reporting in accordance with established procedures since the war began. In our analysis, we use average data (e.g., kg nutrient ha$^{-1}$ yr$^{-1}$) rather than total annual values (e.g., kg nutrient per county or country) to minimize spatial coverage bias and ensure comparability within the same county across different time periods (e.g., non-occupied vs. partially occupied). We assume that average values and trends are representative of the county as a whole. However, if a county is more than 75% occupied, either by total area or by area under the studied staple crops (whichever threshold is reached first), we exclude it from the analysis, assuming the data are not representative. For example, in 2023, Kharkivska (E-KR) and Luhanska (E-LU) counties in their entirety, Donetska (E-DO), for maize were excluded. Also, we excluded data for 2023 for Chernihivska (C-CN) county, as it reported an unexplained increase in fertilizer application, such as 1.6-fold (155 kg N ha$^{-1}$ yr$^{-1}$) for sunflower and 1.7-fold (245 kg N ha$^{-1}$ yr$^{-1}$) for maize in the 2nd year of the large-scale war, compared with 2021, the pre-war year, which was most likely an error.

### Nutrient contents in manure and other organic fertilizers

Since information on manure types and other organic materials applied to fields was either unavailable in national statistics before 2018 or only partially available for 2018–2023, the data during these years indicated only total manure from agricultural animals (i.e., a sum of manure from cattle, pigs, sheep and goats), total manure from poultry (i.e., a sum of manure from chickens, ducks and geese), as well as other organic materials, including other 'unspecified' organic fertilizers, silt, sapropel, peat and related substances[14]. In both cases, no data on N, P, K contents in these manures and organic fertilizers were reported in national statistics[14]. The literature data on average N, P, K contents in manure varied widely (Supplementary Table 4), while actual nutrient content in manure can be even more variable, depending on factors such as (i) animal types, as each

produces manure with unique nutrient profiles (e.g., poultry manure is often richer in N and P compared to cattle manure), (ii) the diet of the animals (e.g., nutrient-rich feeds yield manure with higher nutrient levels), (iii) age and growth stage (e.g., younger animals retain more nutrients for growth, resulting in lower nutrient levels in their manure), (iv) moisture content in manure (e.g., liquid manure generally has diluted nutrient levels compared to solid manure), and (v) storage conditions, including temperature and handling practices, which can lead to nutrient loss over time, particularly of N. Therefore, actual manure testing is recommended for precise field application, as part of the Good Agricultural Practices.

To quantify N, P and K content in the present study for the 'undefined' organic fertilizers as reported in national statistics[14], we:

1. Surveyed various literature sources for N, P and K content across manure types and ultimately decided to use data reported as typical for Ukraine[33] (see Supplementary Table 4).
2. Analyzed detailed statistics from 2018–2023 to estimate the average contribution of 'total manure from agricultural animals' (79.8%), poultry manure (10.8%) and other organic fertilizers (9.4%, including a minor share (<1%) of sapropel and peat) to the total organic fertilizer applied[14]. We used the poultry share in a sensitivity analysis to approximate the poultry proportion of total organic fertilizers applied in previous years.
3. Calculated a mean sensitivity coefficient (1.288) as the mean ratio between (a) the poultry manure share in total organic fertilizer to (b) the poultry manure share in total manure produced during 2018–2023, and applied this coefficient retrospectively to estimate poultry manure shares in total organic fertilizer for earlier years.
4. Assumed that most of the manure applied in the category 'total manure from agricultural animals', as reported in national statistics on animal husbandry[14], was likely fresh cattle manure on straw bedding due to the practical absence of manure management systems in Ukraine. We also assumed that chicken manure comprised the majority of poultry manure applied. Due to the lack of specific data and for simplicity, we applied the N, P, K contents for fresh cattle manure on straw bedding to other 'unspecified' organic fertilizers, which made up less than 10% of the total organic fertilizer applied in 2018–2023.
5. Finally, we computed weighted N, P and K contents in the total 'undefined' manure applied to the field for each study year based on the proportions of poultry manure (using specific NPK contents for chicken manure) and other organic fertilizers, including 'total manure from agricultural animals' and 'other organic fertilizer applied' (using specific N, P, K contents for fresh cattle manure on straw bedding) (see Supplementary Table 5).

### Atmospheric deposition data

We used mean annual atmospheric deposition rates of total inorganic N compounds, i.e., the sum of reduced ($NH_3$ and $NH_4^+$) and oxidized (NO, $NO_2$, $NO_3^-$, $N_2O_5$, $HNO_3$, etc.) forms, which were simulated by the EMEP-MSC-W model for 1990–2022[34] (Supplementary Fig. 5); these EMEP-reported data were used for both country and county scales. However, we must emphasize the limited applicability of current atmospheric model results, including EMEP-MSC-W, due to: (i) the inclusion of deposition estimates for inorganic N forms only, (ii) reliance on data largely based on estimated N emissions reported by countries that lack, or have poorly developed, national emission monitoring networks, including Ukraine, rather than on measured data, (iii) the lack of any estimates for organic N deposition, which have been shown to significantly contribute to total atmospheric deposition[16], including in studies conducted in Ukraine[17,18,35] (Supplementary Note 3.2). Incorporating organic N deposition requires more field studies with further in-depth investigation of the underlying chemical mechanisms before being included in atmospheric models. Meanwhile, atmospheric deposition of P and K is generally assumed to be negligible[13,36]; however, more field measurements worldwide are needed to confirm or challenge these assumptions in specific regions (Supplementary Note 3.2).

### Biological N fixation data

Although symbiotic biological N fixation is often not reported for maize, sunflower, and wheat[37], N fixation by free-living organisms is assumed to occur in all the studied cropping systems in Ukraine. Since no specific data for Chernozems are available, we included an average N fixation contribution by free-living organisms of 5 kg N ha$^{-1}$ yr$^{-1}$, as a feasible rate based on the review by Herridge et al.[37]. Available evidence suggests insignificant use of legumes in crop rotations in Ukraine.

### Nutrient balance estimations

We included nutrient inputs via organic (including manure) and synthetic fertilizers, atmospheric inorganic N deposition, and N fixation by free-living soil organisms, thereby neglecting inputs via soil weathering and mineralization. We did not consider any input from irrigation, as our study focused solely on rain-fed systems. Thus, annual agricultural balance of each crop (wheat, maize, sunflower) for each nutrient (N, P, K) was calculated as the difference between a total fertilizer input, that is, the sum of synthetic and organic fertilizer, applied as annual mass of N, P, K per hectare, plus atmospheric inorganic N deposition and N fixation by free-living organisms, minus the crop yield, expressed as the corresponding nutrient mass per hectare. Positive nutrient balances indicate nutrient inputs via fertilizers and manure exceed the nutrient withdrawal with harvested crop, while negative balances reflect a nutrient deficit, *i.e.*, nutrient withdrawal with harvested crop exceeds the nutrient inputs via fertilizers and manure. We assume that straw and other crop residues were largely returned to the soil and, therefore, were not included in the harvested crop yield (see Supplementary Note 3.3). We highlight general trends for wheat, maize, and sunflower crops, while recognizing that considering crop rotations may influence results at the site level (see Supplementary Notes 3.4).

### Nutrient use efficiency estimations

We estimated total fertilizer N, P, and K use efficiencies (NUE, PUE, KUE) for each cropping system as the ratio of nutrients in harvested yield to nutrient inputs via organic and synthetic fertilizers applied. According to EUNEP[19] the optimal NUE, accounting all N sources rather than fertilizers only, was suggested to be in a range between 50–90% with a desired yield above 80 kg N ha$^{-1}$. We also applied this framework also for P and K. Thus, NUE, PUE, and KUE above 90–100% indicate a high risk of soil nutrient mining, while those below this level indicate increasing risk of losses to the environment, with possible soil nutrient accumulation. The limitations of the generalized EUNEP approach have been recently outlined[38,39] and discussed in the Supplementary Note 3.5.

### Nutrient contents in crop yields

As information regarding NPK contents in the harvested crop yield was not available from national statistics, we made a survey to explore various sources, and finally used crop removal coefficients for each crop retrieved from the most recent and comprehensive meta-analysis conducted[8,40]. We crosschecked these coefficients for NPK[40] with other available datasets (Supplementary Table 6).

### Nutrient management scenarios

We developed five nutrient management scenarios to assess how Ukraine's crop nutrient balances might evolve by 2030 under different strategies (Table 1; Supplementary Tables 2, 3). These include: (i) Business-as-Usual (BaU) scenario (S-0), assuming a return to 2021 practices; (ii) Extended war disruption scenario (S-w), assuming prolonged fertilizer shortages at 2023 levels; (iii) Manure-enriched precision fertilization scenario (S-1), targeting manure to supply 30% of total fertilizer N by partially substituting synthetic N; (iv) Enhanced-efficiency fertilizer (EEF) scenario building on S-1 (S-2); and (v) Legume-based diversification scenario (S-3), designed to be combined with S-2 or S-1. Sustainable nutrient management scenarios (S-1, S-2, S-3) are based on maintaining the 2021 yield level. All scenario designs, assumptions, calculations, and limitations are detailed in Supplementary Note 2.8.

### Ethics statement

This study was conducted in accordance with the Research Ethics guidelines of UK Research & Innovation. Local researchers were involved in all stages of the research, including study design, data collection, and analysis, and co-authored this study. We have taken steps to ensure that the research is regionally and locally relevant and that the findings will be disseminated to the regional and local community. We confirm that there are no competing interests to declare and that all data and materials are available to others upon reasonable request.

### Reporting summary

Further information on research design is available in the Nature Portfolio Reporting Summary linked to this article.

### Data availability

The data supporting the findings of this study and scenarios are provided within the paper and its Supplementary Information. Agricultural statistical data are publicly available from the State Statistics Service of Ukraine (https://ukrstat.gov.ua/)[14]. The crop removal coefficients are publicly available from the dataset of Ludemann[40]. The datasets used in this paper to generate main text figures are available from Figshare https://doi.org/10.6084/m9.figshare.29943329[41]. All other data used in this manuscript are clearly attributed to their source publications through citation.

### Code availability

No specific code was generated within this study. We used R (version 4.4.3) with the corresponding packages to transform data, calculate statistics and generate figures, and these custom scripts are available from the corresponding authors upon request.

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

## Acknowledgements

This study was supported by UKRI National Capability funding through the UKCEH AGILE project (grant reference: NE/W004976/1), the GEF-UNEP funded project 'Towards INMS' (see www.inms.international), the Researcher at Risk Fellowship Programme of the British Academy with the Council for At-Risk Academics (CARA), the GEF-UNEP uPcycle project (https://www.upcyclelakes.org/) and the UKRI National Capability International Programme.

## Author contributions

S.M. and M.A.S. conceived the study concept and designed the study. S.M. collected data, performed data analysis and interpretation, wrote the initial draft of the paper. M.A.S. contributed to the data analysis, interpretation, text drafting and revision. O.O., B.S., A.B., V.M., W.B., M.V. and E.N. contributed to data interpretation and revision of the paper. All authors reviewed, edited and approved the final version of the paper.

## Competing interests

The authors declare no competing interests.
