## [Transparent Peer Review file · Communications Earth & Environment]

Nutrient asymmetry challenges the sustainability of Ukrainian agriculture

Corresponding Author: Dr Sergiy Medinets

Version 0:

Decision Letter:

Dear Dr Medinets,

Your manuscript titled "Nutrient asymmetry challenges the sustainability of Ukrainian agriculture" has now been seen by 4 reviewers, whose comments are appended below. You will see that they find your work of some potential interest. However, they have raised quite substantial concerns that must be addressed. In light of these comments, we cannot accept the manuscript for publication, but would be interested in considering a revised version that fully addresses these serious concerns.

We hope you will find the reviewers' comments useful as you decide how to proceed. Should additional work allow you to

- address these criticisms (that is, either to incorporate the suggestions or provide a compelling argument why the point made by the reviewer is not valid, or relevant to the editorial threshold as outlined below)

AND

- meet our editorial thresholds as outlined below, then we would be happy to look at a substantially revised manuscript.

In the following, we list our main editorial concerns.

- * Provide new insights on the long-term trend of the nutrient budget of Ukraine, including a clear justification for the novelty and importance of the analysis and in-depth discussions on the practical and feasible ways of nutrient management for food security, particularly in the context of Ukrainian war.

- * Clearly explain the assumptions made for the nutrient budget estimation in the study.

- * Provide a comprehensive assessment on the nutrient budget including the role of livestock, soil, BNF etc.

If you choose to take up this option, please either highlight all changes in the manuscript text file, or provide a list of the changes to the manuscript with your responses to the reviewers.

When resubmitting, please provide a point-by-point response to the reviewers' comments. Please submit your responses as a separate file, distinct from your cover letter where you can add responses to the Editors' comments that you do not want to be made available to the reviewers. Word files are preferred. We recommend that any figures, tables or graphs that are included in the response to reviewers are also included in the main article or Supplementary Information.

If the revision process takes significantly longer than three months, we will be happy to reconsider your paper at a later date, as long as nothing similar has been accepted for publication at Communications Earth & Environment or published elsewhere in the meantime.

Please use the following link to submit your revised manuscript, point-by-point response to the reviewers' comments with a list of your changes to the manuscript text (which should be in a separate document to any cover letter), a tracked-changes version of the manuscript (as a PDF file) and any completed checklist:

Link Redacted

Please do not hesitate to contact us if you have any questions or would like to discuss the required revisions further. Thank you for the opportunity to review your work.

Best regards,

Jinfeng Chang, PhD
Editorial Board Member
Communications Earth & Environment
orcid.org/0000-0003-4463-7778

Mengjie Wang
Associate Editor
Communications Earth & Environment

EDITORIAL POLICIES AND FORMAT

If you decide to resubmit your paper, please ensure that your manuscript complies with our editorial policies and complete and upload the checklist below as a Related Manuscript file type with the revised article:

Editorial Policy Policy requirements
(Download the link to your computer as a PDF.)

- Behavioural and social science
- Ecological, evolutionary & environmental sciences
- Life sciences

<https://www.nature.com/documents/nr-reporting-summary.zip>

For your information, you can find some guidance regarding format requirements summarized on the following checklist: (<https://www.nature.com/documents/commsj-phys-style-formatting-checklist-article.pdf>) and formatting guide (<https://www.nature.com/documents/commsj-phys-style-formatting-guide-accept.pdf>).

REVIEWER COMMENTS:

Reviewer #1 (Remarks to the Author):

Numerous studies on NPK balance at various scales and levels have been conducted internationally. The uniqueness of this study, however, lies in Ukraine's recent experience of the Russo-Ukrainian war. This research must highlight not only the distinct characteristics of the Ukrainian NPK balance but also the scientific significance of investigating these unique aspects, beyond the context of the war itself. If the study only addresses the national-level NPK balance, it would lack sufficient innovation to justify publication.

The paper proposes the development of Integrated Nutrient Management Plan in Ukraine, but the question remains: who will implement them? Is such a proposal feasible in a country embroiled in war? Should greater emphasis be placed on how the international community can assist in ensuring that ordinary citizens no longer face hunger? Furthermore, the country should also prioritize integrated nutrient management alongside its broader recovery efforts.

It is suggested that the paper strengthen the analysis of the importance of nutrient management in Ukraine for food security and the global food supply chain, as well as examining the impact of the Russo-Ukrainian war. Based on this, this paper can provide practical and feasible recommendations on how to improve the nutrient management in the war to mitigate the impacts on food security.

Reviewer #2 (Remarks to the Author):

This manuscript presents an analysis of nutrient imbalances in Ukrainian agriculture, in particular N, P, and K inputs and outputs over the period 1980-2023. The study evaluates the impact of historical and recent disruptions, including the Russian invasion, on agricultural sustainability and nutrient dynamics. The manuscript highlights substantial soil nutrient mining and imbalances that threaten long-term agricultural productivity and sustainability. The authors propose some policy interventions.

My impression of this paper is mostly good. It does perhaps not represent a major scientific advance, but it does usefully apply established methods from agronomy and environmental science to the very timely topic of Ukrainian agriculture under the severe crisis conditions imposed by war. The juxtaposition of crisis management, agri-environmental policy, and the historical perspective is novel and useful. The results are well organized and relevant. Thus I would recommend the paper for publication after some minor changes.

I have a few suggestions as follows.

1. Concerning the results, I would suggest that the authors expand a little bit on the trend of net mineralization of soil organic matter in Ukrainian soils. The description is reminiscent of what happened historically in the American midwest (see, e.g., David et al. 2001 <https://doi.org/10.1100/tsw.2001.283>). What is the importance of net mineralization for Ukraine's current agricultural production and do you have an estimate of how long the current situation can persist?
2. The inputs of N from biological fixation are mentioned a few times but not estimated. I find that the topic is inadequately addressed. How much of the considered crops are grown in rotations with biological N fixation? What is a rough estimate of these N inputs over the course of a crop rotation, or as a spatial average over Ukrainian cropland? Perhaps no change is needed in the modeling, but at least a rough estimate and a comment on it would be suitable.
3. Although the discussion of humus content in the supplementary is interesting and highly relevant, I would suggest that the authors connect this part of the text to more modern soil science concepts. At least preface this part with a short discussion of what "humus" is and isn't. It would be useful if the term is put into context of modern soil organic matter references and terminology (e.g., Sutton & Sposito 2005 <https://doi.org/10.1021/es050778q>; Lehmann & Kleber 2015 <https://doi.org/10.1038/nature16069>).

Reviewer #3 (Remarks to the Author):

The manuscript "Nutrient asymmetry challenges the sustainability of Ukrainian agriculture" by S. Medinets et al. provides an incredibly data-rich and well elaborated analysis of the development of crop production in the Ukraine during the last decades, including the impact of important geo-political events, such as the independence from, and the invasion by Russia. The results and recommendation are justified and pertinent in view of the during- and post-war recovery of the Ukraine and the important role of the Ukraine in (national and) global food security.

Main and supplementary information are well written and all graphical exhibitions are carefully made and add to the content of the manuscript. The amount of supplementary information is appreciated, but has also a downside. A reader who wishes to follow the details, this poses a challenge as it requires continuous going back and forth between main and supplementary text. For readers who wish to only read the main text, the numerous references to the supplementary information might be disrupting the 'flow' of reading. I do not suggest to remove supplementary elements, and see this more as an editorial challenge to allow more information into the main piece under the constraints of the article type.

My key request for improving the manuscript would be a critical assessment of the role of livestock. The role of manure as fertiliser in the assessed period has been well described. However, a few questions remain unclear: i/ what fate does the non-use manure have? Is it fully lost to the environment (how?). what would e.g. be the impact on nitrogen deposition rates? ii/ the 'return' to coupled crop-livestock systems is suggested. However, Suppl. Text 3.2 suggests that this has not only positive effects (higher use of manure as fertiliser) but also contributed to a decline of SOM. This should be included in the main analysis upfront. iii/ What is the share of cereals used for feed during the assessed period (beyond crop residues)? Livestock can only have a positive net input of nutrients to crops cultivated for export or food if nutrient are imported with feed, otherwise the return to the soils is inevitably less than extraction for feed.

Furthermore, the suggestion of a "Smart Fertilizer Planner tool" should be complemented by an assessment of existing tools of similar purpose, for example under the EU Common Agricultural Policy. If such tools could be extended, it would greatly increase the feasibility of such a tool.

Importance of SOM for soil fertility was demonstrated and emphasised. Why should the SFP be based on a 'zero nutrient balance'? Wouldn't it be important to aim to re-build SOM in the soils, and should for that purpose not also C included into

the SFP, which would allow a better consideration of the importance of crop residues management?

The development of the area/share of non-used UAA should be better explained more in detail and earlier in the manuscript. I learn details when reading Supplementary Text 2.9 but it is already important for the understanding of Supplementary Figure 1.

Minor comments.

Page 3, first paragraph: Starting a sentence 'This indicates' that follows an observation for maize and what and the opposite for sunflower makes it difficult to understand what exactly is meant to indicate soil nutrient depletion.

Supplementary Table 2. Broiler not Brolier.

Reviewer #4 (Remarks to the Author):

This paper could be treated as a short comment rather than a full-length research article. The objective is to analyze the imbalanced input and output of nutrients in Ukraine during the past decades. The topic is vital since nutrients serve as the fundamental resource for food production and environmental degradation. However, this paper only analyzed a few indicators that can be obtained directly from open statistics, failing to reveal the underlying causes or subsequent effects of the trends. The text looks like a report to the government rather than an academic paper that both scientists and the public will be interested in. The strong assumptions make the results questionable, the N/P deficits may lead to reduced nutrient content in food (rather than static). Besides, the war only affect the nutrients balance in recent a few years. Reduced agricultural activities could reduce both nutrient inputs and outputs, which doesn't not ncessarily indicate surplus and environmental risks. Thus, the authors should think about in-depth research question, improve methods through sampling, monitoring or modeling.

Communications Earth & Environment is committed to improving transparency in authorship. As part of our efforts in this direction, we are now requesting that all authors identified as 'corresponding author' create and link their Open Researcher and Contributor Identifier (ORCID) with their account on the Manuscript Tracking System prior to acceptance. ORCID helps the scientific community achieve unambiguous attribution of all scholarly contributions. You can create and link your ORCID from the home page of the Manuscript Tracking System by clicking on 'Modify my Springer Nature account' and following the instructions in the link below. Please also inform all co-authors that they can add their ORCIDs to their accounts and that they must do so prior to acceptance.

If you experience problems in linking your ORCID, please contact the Platform Support Helpdesk.

Version 1:

Decision Letter:

Dear Dr Medinets,

Your manuscript titled "Nutrient asymmetry challenges the sustainability of Ukrainian agriculture" has now been seen by 3 reviewers, whose comments are appended below. You will see that they find your work of some potential interest. However, they have raised quite substantial concerns that must be addressed. In light of these comments, we cannot accept the manuscript for publication, but would be interested in considering a revised version that fully addresses these serious concerns.

We hope you will find the reviewers' comments useful as you decide how to proceed. Should additional work allow you to

- address these criticisms (that is, either to incorporate the suggestions or provide a compelling argument why the point

made by the reviewer is not valid, or relevant to the editorial threshold as outlined below)
AND

- meet our editorial thresholds as outlined below,

then we would be happy to look at a substantially revised manuscript.

In the following, we list our main editorial concerns.

*Editorial threshold 1: Articulate the novelty of the study based on clear supporting evidence.

*Editorial threshold 2: Provide linkages between policy recommendations and main findings in the study.

*Editorial threshold 3: Clearly justify the chosen methods and discuss data limitations.

If you choose to take up this option, please either highlight all changes in the manuscript text file, or provide a list of the changes to the manuscript with your responses to the reviewers.

When resubmitting, please provide a point-by-point response to the reviewers' comments. Please submit your responses as a separate file, distinct from your cover letter where you can add responses to the Editors' comments that you do not want to be made available to the reviewers. Word files are preferred. We recommend that any figures, tables or graphs that are included in the response to reviewers are also included in the main article or Supplementary Information.

If the revision process takes significantly longer than three months, we will be happy to reconsider your paper at a later date, as long as nothing similar has been accepted for publication at Communications Earth & Environment or published elsewhere in the meantime.

Please use the following link to submit your revised manuscript, point-by-point response to the reviewers' comments with a list of your changes to the manuscript text (which should be in a separate document to any cover letter), a tracked-changes version of the manuscript (as a PDF file) and any completed checklist:

Link Redacted

Please do not hesitate to contact us if you have any questions or would like to discuss the required revisions further. Thank you for the opportunity to review your work.

Best regards,

Jinfeng Chang, PhD
Editorial Board Member
Communications Earth & Environment
orcid.org/0000-0003-4463-7778

Mengjie Wang
Associate Editor, Communications Earth & Environment
Consulting Editor, Communications Sustainability
Bluesky: @commsearth.nature.com; @ commssustain.nature.com

EDITORIAL POLICIES AND FORMAT

If you decide to resubmit your paper, please ensure that your manuscript complies with our editorial policies and complete and upload the checklist below as a Related Manuscript file type with the revised article:

- Behavioural and social science
- Ecological, evolutionary & environmental sciences
- Life sciences

For your information, you can find some guidance regarding format requirements summarized on the following checklist: (<https://www.nature.com/documents/commsj-phys-style-formatting-checklist-article.pdf>) and formatting guide (<https://www.nature.com/documents/commsj-phys-style-formatting-guide-accept.pdf>).

REVIEWER COMMENTS:

Reviewer #1 (Remarks to the Author):

The author has undertaken comprehensive and thoughtful revisions to this manuscript, significantly improving its quality. I agree that this paper is more appropriately positioned as a Short Communication rather than a full-length research article. Its primary aim should be to inform both the scientific community and policymakers about the nutrient imbalances resulting from Ukraine's unique circumstances and the broader environmental consequences. It should also serve as a call to action for the international community to address the associated food security risks and their implications for global food supply chains.

However, further refinements are necessary before it can meet the standards for publication.

(1) The paper's novelty remains insufficiently articulated. The stated focus on the "phased characteristics of nitrogen, phosphorus, and potassium in Ukraine" does not, in itself, constitute a novel contribution—especially given that Figures 1 and 2 do not convincingly support this claim. While the proposed mitigation strategies are practical, they do not represent a scientific innovation. These recommendations lack empirical testing, making their feasibility and effectiveness uncertain. Without case studies or scenario analyses, the suggestions remain speculative rather than evidence-based. The author should clarify more explicitly how this study advances academic understanding or contributes to global sustainability goals.

(2) If the authors assert that they "provide a unique analysis of the way in which the Russo-Ukrainian war has fundamentally changed nutrient balances for Ukraine," then the uniqueness of that analysis must be clearly demonstrated and substantiated with evidence. The analytical framework appears conventional, and the model used does not introduce any novel computational or theoretical advancements. Given the significant disruption to Ukraine's monitoring systems during the war, the biggest challenge is arguably the acquisition of reliable and comprehensive data. The manuscript would benefit from a more robust justification of the chosen methods and a clearer discussion of data limitations.

(3) The current "Action" section is overly long and includes recommendations that are fragmented and, in some cases, only tangentially related to the study's findings. This section should be streamlined to focus on the most critical proposals that directly emerge from the research results. Moreover, the paper does not evaluate the potential impacts of the proposed actions. It would be helpful to include preliminary analyses earlier in the manuscript to assess how the recommended interventions could influence: (a) Ukraine's NPK balance, (b) domestic food security, and (c) global agricultural supply chains. Such additions would strengthen the linkage between the study's findings and its policy recommendations, ensuring the latter are grounded in evidence.

(4) While the paper identifies several key trends through data analysis, it lacks in-depth investigation into the underlying drivers or mechanisms. As a result, the recommendations are overly general. For example, the study notes that fertilizer application rates and crop yields remained relatively stable during the first two years of the Russia-Ukraine war but experienced a sharp decline in 2023—with fertilizer use dropping by 37–54% compared to 2021. However, the manuscript does not explore the causes behind this delayed impact. A deeper analysis of these drivers would enhance the explanatory power and practical value of the study.

Reviewer #2 (Remarks to the Author):

Thank you to the authors for the thorough reply to the reviewer questions. I feel that all my initial concerns have been addressed and that this paper is ready for publication.

Reviewer #3 (Remarks to the Author):

Thanks to the authors who have addressed satisfyingly and to the extend possible the points raised.

Communications Earth & Environment is committed to improving transparency in authorship. As part of our efforts in this direction, we are now requesting that all authors identified as 'corresponding author' create and link their Open Researcher and Contributor Identifier (ORCID) with their account on the Manuscript Tracking System prior to acceptance. ORCID helps the scientific community achieve unambiguous attribution of all scholarly contributions. You can create and link your ORCID from the home page of the Manuscript Tracking System by clicking on 'Modify my Springer Nature account' and following the instructions in the link below. Please also inform all co-authors that they can add their ORCIDs to their accounts and that they must do so prior to acceptance.

Version 2:

Decision Letter:

Dear Dr Medinets,

Your manuscript titled "Nutrient asymmetry challenges the sustainability of Ukrainian agriculture" has now been seen by our reviewers, whose comments appear below. In light of their advice we are delighted to say that we are happy, in principle, to publish a suitably revised version in Communications Earth & Environment.

We therefore invite you to revise your paper one last time to address the remaining concerns of our reviewers. At the same time we ask that you edit your manuscript to comply with our format requirements and to maximise the accessibility and therefore the impact of your work.

EDITORIAL REQUESTS:

*****Please take care to match our formatting and policy requirements. We will check revised manuscript and return manuscripts that do not comply. Such requests will lead to delays. *****

SUBMISSION INFORMATION:

OPEN ACCESS:

Communications Earth & Environment is a fully open access journal. Articles are made freely accessible on publication. For further information about article processing charges, open access funding, and advice and support from Nature Portfolio, please visit <https://www.nature.com/commsenv/open-access>

Link Redacted

Best regards,
Mengjie Wang

Associate Editor, Communications Earth & Environment
Consulting Editor, Communications Sustainability
Bluesky: @commsearth.nature.com; @commssustain.nature.com

REVIEWERS' COMMENTS:

Reviewer #1 (Remarks to the Author):

No further comment or suggestion

** Visit Nature Portfolio's author and referees' website at www.nature.com/authors for information about policies, services and author benefits**

Detailed response from the authors to comments of the editor and peer reviewers

We thank the editor and reviewers for their constructive comments on our submitted manuscript, which have enabled us to strengthen the submission. We hope that the editor and reviewers will agree that we have addressed all the points raised.

For convenience in cross-referencing our point-by-point response, we have numbered the comments of the reviewers as follows: Reviewer 1: [R1.1...R1.n]; Reviewer 2: [R2.1...R2.n]; etc. Our replies are listed in the blue highlighted text starting with 'Response to [...]'.

RESPONSE TO THE REVIEWER COMMENTS

Reviewer #1 (Remarks to the Author):

[R1.1] Numerous studies on NPK balance at various scales and levels have been conducted internationally. The uniqueness of this study, however, lies in Ukraine's recent experience of the Russo-Ukrainian war. This research must highlight not only the distinct characteristics of the Ukrainian NPK balance but also the scientific significance of investigating these unique aspects, beyond the context of the war itself.

Response to [R1.1]. We would like to underline that this study covers three distinct periods in the evolution of Ukrainian agriculture:

- (i) the last 10 years of the pre-independence period (1980–1990),
- (ii) the 30 years of the independence period (1991–2021) prior to the large-scale war, and
- (iii) the years of the war's impact (2022–2023).

Thus, we discuss the impacts of nutrient imbalances across these contrasting periods over the timeline of 1980–2023, which extends far beyond the context of the war that began in February 2022.

Ukraine's significant contribution to the production and export of the studied crops (wheat, maize, and sunflower) makes it important for global food security. Meanwhile, unsustainable nutrient management (nutrient imbalance and the combination of imbalances, including spatially, both between inputs and outputs, and different nutrients) - varying across different periods, asymmetrically improving during the independence era, and aggravated by the war - makes this issue scientifically and politically significant in terms of environmental impact, soil degradation and food supply. This includes challenges observed long before the war, changes currently driven by the war, and potential future outcomes if no actions are taken.

To make these points more clearly, we have modified the manuscript as follows:

- 1) Added to the Introduction:
“ Ukraine has long been one of the world's leading exporters of wheat, maize and oilseed products.”
...
“Here, we trace nutrient management during Ukraine’s transition from the Soviet era (from 1980) through the challenges of independence (since 1991), culminating in the pre-war period of high agricultural productivity (2019-2021), and extending up to two years of war (2022-2023).”
- 2) Rephrased/ added some text for a sake of clarity in the Results section (see changes highlighted in this section of the manuscript).
- 3) We have strengthened the analysis of the importance of nutrient management in Ukraine for food security and the global food supply chain, including the war impact. To reflect this, we have also added the following text to a new section on ‘Implications for food security’; please see the cited text below under the [comment R1.6].

[R1.2] If the study only addresses the national-level NPK balance, it would lack sufficient innovation to justify publication.

Response to [R1.2]. With respect to the reviewer, we disagree, since our study clearly goes beyond national-level NPK balance, as we show detailed comparisons for different regions and counties of Ukraine. However, we would also disagree with the reviewer that analysis of the situation in Ukraine would mean that the study lacked innovation. Science can happen at multiple scales, and we would argue that a detailed analysis of a country situation can provide specific insights that will be missed at both local and global levels.

Thus we start by addressing NPK balances and nutrient use efficiencies for each type of studied crop at the national level (Fig. 1) and then show information at the county/regional level (Fig. 2; Supplementary Figs. 16-18, 20-22, 26); in the revised version, all these figures have been updated with new calculation results that include atmospheric N deposition and N fixation by free-living organisms (see our responses to Reviewer 2 [comment R2.2] and Reviewer 3 [comments R3.3]). These approaches then provide the foundation to address the high spatial asymmetry observed at the county and regional levels.

To the best of our knowledge, this is the first study to examine NPK balance and nutrient use efficiencies for individual crop agroecosystems (wheat, maize and sunflower) in Ukraine over the extended period of 1980–2023, at both national and county levels.

[R1.3] The paper proposes the development of Integrated Nutrient Management Plan in Ukraine, but the question remains: who will implement them? Is such a proposal feasible in a country embroiled in war?

Response to [R1.3]. We agree with the reviewer that a focus on food security and sustainability is harder while a country is at war. We have accordingly made clear that while actions should start now, they represent a long-term commitment by the government, jointly developed by scientists and policy-makers and co-designed with farmers.

Concerning who would implement measures during a time of war, we would emphasize that many functions of daily life continue. Thus, farmers continue to farm, while the government is active in supporting the functions of society that have to continue, even under difficult circumstances. At the same time, international support is needed, including that which can enable knowledge transfer and development. Accordingly, we make clear that, it is not only feasible, but also necessary to start now, while continuing the development of the Integrated Nutrient Management Plan into the longer term.

We have clarified these points in the manuscript, adding to the Section on ‘Actions’:

“Aside from the obvious humanitarian crisis, these human disturbances limit the ability to focus on sustainable practices. Recognizing these challenges, the observed nutrient asymmetry highlights the need to develop an Integrated Nutrient Management Plan for Ukraine”

...

“Developing such a plan should not wait until the war is over. While the war makes conditions harder, all the measures listed could already start with appropriate investment. Indeed, there is now an urgent need for strengthening international action to support Ukrainian agriculture and prevent further depletion of precious nutrients with the accompanying soil degradation, while recognizing that this must be the start of a long-term commitment to supporting a transition to sustainable nutrient management.”

...

In terms of resources for this plan realization, we suggest the following:

*“We propose that an **Integrated Nutrient Management Plan for Ukraine** be incorporated into the **agriculture sector recovery under the Ukraine Recovery and Reconstruction (URR) budget**, which currently accounts for 10.5% of the total planned URR budget of USD 524 billion²⁷.”*

It is helpful to illustrate these points by considering the specific actions we have highlighted:

- We have proposed the development of a region-tailored, variety-specific Smart Fertilizer Planner tool to help farmers estimate NPK input requirements based on the yields of top-performing crops. It would be feasible under the present conditions of war for work on this to begin immediately. For example, the tool could be created by researchers using crop sample analyses and input data provided by farmers, under a program supported by national and international funding mechanisms.
- Furthermore, a user-friendly app or website could be developed with the assistance of the Ministry of Digital Transformation of Ukraine or a third-party app developer, under the appropriate project or financial support. Again, work on this approach could begin immediately, even under the present conditions of war.

We make these points more clearly by briefly emphasizing *“**While the war makes conditions harder, action all the measures listed could already start, with appropriate investment**”*, as noted above.

[R1.4] Should greater emphasis be placed on how the international community can assist in ensuring that ordinary citizens no longer face hunger?

Response to [R1.4]. This question can be considered as beyond the scope of our study. Of course, the most important issue is how to stop the war and prevent further loss of life, which is also critical to ordinary citizens no longer facing hunger. Similarly, increased emphasis on how to avoid hunger internationally is important. However, this is outside the scope of our manuscript.

We note in passing that hunger is not currently a major issue for ordinary Ukrainian citizens living outside the direct conflict (frontline) zones - at least so far. However, the destruction of infrastructure, such as the energy sector (especially during winter, when there is no heating) and water supply and treatment facilities (particularly in summer, with regard to hygiene standards), as well as the bombing of residential and public areas poses a serious and direct threat to human life.

Indeed, there are many humanitarian missions and support measures for civilians and small farmers (see “Support Programs for Small Farmers” in Section 2.9: Farming During Wartime). However, in our study, we focus specifically on what has been happening to the nutrient balance and nutrient use efficiency of Ukraine’s three main staple crops. We examine this considering the relevance for sustainability and food security at country-level, pan-European and global scale. These are especially relevant given Ukraine’s significant role as a global food exporter. We also analyze how the war has impacted these factors, the potential consequences if no action is taken, and propose possible solutions.

We also mentioned the potential challenges that sustainable practice development encounters under significant humanitarian crises, and why it still remains of a high priority:

*“Based on our analyses, there is strong evidence of imbalanced nutrient use in Ukraine, **which is greatly exacerbated by the current Russo-Ukrainian war**. The increased asymmetry between nutrient inputs and outputs, and between N, P and K, is reducing the sustainability of crop production, while also jeopardizing the economy, environment and future global food security. These **changes originate in part also from the hundreds of thousands of casualties in Ukraine, as well as around 5 million internally displaced Ukrainians and loss of 20% of the population who have fled the country**²⁶. Aside from the obvious humanitarian crisis, **these human disturbances limit the ability to focus on sustainable practices**.*

Recognizing these challenges, the observed nutrient asymmetry highlights the need to develop an Integrated Nutrient Management Plan for Ukraine, which can also have relevant messages for other countries.”

[R1.5] Furthermore, the country should also prioritize integrated nutrient management alongside its broader recovery efforts.

Response to [R1.5]. We agree strongly with the reviewer. This is why we emphasize it in this paper to help drive communication on the issue. To make this more clear, we have now included the following text:

“We propose that an Integrated Nutrient Management Plan for Ukraine be incorporated into the agriculture sector recovery under the Ukraine Recovery and Reconstruction (URR) budget, which currently accounts for 10.5% of the total planned URR budget of USD 524 billion²⁷. We also recommend prioritizing agro-food system resilience as a key focus within the URR framework”.

[R1.6] It is suggested that the paper strengthen the analysis of the importance of nutrient management in Ukraine for food security and the global food supply chain, as well as examining the impact of the Russo-Ukrainian war.

Response to [R1.6]. We have strengthened the analysis of the importance of nutrient management in Ukraine for food security and the global food supply chain, including the war impact. To reflect this, we have also added the following text to a new section on ‘**Implications for food security**’:

“The evolving nutrient management landscape in Ukraine holds profound implications not only for national food security but also for the stability of the global food supply chain. As one of the world’s top exporters of wheat, maize and sunflower oil, Ukraine plays a critical role in feeding over 400 million people worldwide, particularly in import-dependent regions^{20,21} (Supplementary Fig. 2-4, Text 1.1). However, the ongoing war has disrupted fertilizer supply chains and reduced agricultural inputs, aggravating imbalances of nitrogen, phosphorus and potassium (Fig 1; Supplementary Fig. 8, 26). These imbalances are marked by spatial and crop-type asymmetry in nitrogen (surpluses in some areas, and deficits in others) and persistent deficits in phosphorus and potassium (Fig. 2; Supplementary Fig. 26). The resulting degradation of Ukraine’s Chernozem soils (Supplementary Fig. 14, Text 2.2), is significant as these store about 7% of the world’s soil carbon and underpin around 90% of the country’s agricultural output^{14,22}. Without immediate intervention, this trend could trigger substantial losses of soil organic matter, especially in the regions affected by the war, risking long-term productivity declines for key crops, such as wheat and maize²³.

Given that Ukrainian grain exports traditionally support food security in regions such as North Africa and the Middle East (Supplementary Fig. 3), disruptions in crop output could exacerbate hunger and price volatility worldwide^{20,21}. The ripple effects are global: disruptions in Ukrainian grain exports during the early months of the war contributed to a sharp rise in global food prices, with the FAO Food Price Index reaching its highest level on record in 2022, over 14% higher than the previous year²⁴. This surge intensified global food insecurity, contributing to a wider hunger crisis with a global estimate of 691-783 million people affected in 2022²⁵. Moreover, the under-utilization of manure, caused by the decoupling of livestock and crop systems, represents a missed opportunity for circular nutrient economy practices (Supplementary Fig. 10-13, Text 2.1) that could reduce dependence on synthetic fertilizers while lowering reactive nitrogen and greenhouse gas emissions. Addressing these challenges requires improved access to fertilizers and long-term investments in integrated nutrient management strategies. Strengthening Ukraine’s nutrient resilience is not merely a national priority but a global imperative, when considering the global food security and climate interactions.”

[R1.7] Based on this, this paper can provide practical and feasible recommendations on how to improve the nutrient management in the war to mitigate the impacts on food security.

Response to [R1.7]. We thank the reviewer for their comment. Indeed, this is an important outcome of the paper, which is why we outlined practical and feasible recommendations, prioritized accordingly, in the 'Actions' section, which is currently revised and amended. Please see the revised Action section, as well as our response above [comment R1.3].

Reviewer #2 (Remarks to the Author):

[R2.0] This manuscript presents an analysis of nutrient imbalances in Ukrainian agriculture, in particular N, P, and K inputs and outputs over the period 1980-2023. The study evaluates the impact of historical and recent disruptions, including the Russian invasion, on agricultural sustainability and nutrient dynamics. The manuscript highlights substantial soil nutrient mining and imbalances that threaten long-term agricultural productivity and sustainability. The authors propose some policy interventions.

My impression of this paper is mostly good. It does perhaps not represent a major scientific advance, but it does usefully apply established methods from agronomy and environmental science to the very timely topic of Ukrainian agriculture under the severe crisis conditions imposed by war. The juxtaposition of crisis management, agri-environmental policy, and the historical perspective is novel and useful. The results are well organized and relevant. Thus I would recommend the paper for publication after some minor changes.

I have a few suggestions as follows.

Response to [R2.0]. We thank the reviewer for their positive feedback. We especially appreciate their comment: *“The juxtaposition of crisis management, agri-environmental policy, and the historical perspective is novel and useful.”*

[R2.1] 1. Concerning the results, I would suggest that the authors expand a little bit on the trend of net mineralization of soil organic matter in Ukrainian soils. The description is reminiscent of what happened historically in the American midwest (see, e.g., David et al. 2001 <https://doi.org/10.1100/tsw.2001.283>). What is the importance of net mineralization for Ukraine's current agricultural production and do you have an estimate of how long the current situation can persist?

Response to [R2.1]. We agree on the importance of these aspects. For this reason, we already included “Text 2.2: Soil degradation and the current state of soils in Ukraine” in the Supplementary Material. In response to this comment, we have expanded this section, including concerns about dehumification/mineralization trends:

“Soil degradation and SOM loss. The Food and Agriculture Organization (FAO) has highlighted significant concerns regarding soil degradation in Ukraine, particularly the loss of SOM. An FAO report (2019) noted that soil erosion by wind and water affects approximately 13.4 million hectares, including 10.6 million hectares of arable land. Furthermore, the FAO indicated that degraded and unproductive arable land in Ukraine exceeds 20% (more than 6.5 million hectares) of the total arable land, with annual soil loss due to erosion ranging from 300 to 600 million tons (FAO, 2018). Soil degradation through SOM loss and erosion mainly occur as a result of annual soil cultivation, growth of annual crops, without winter crops or intercrops. As a result, soils are bare for large parts of the year, and then vulnerable to wind and water erosion.

It is noteworthy that conventionally in Soviet and later Ukrainian soil studies (up to the present), humus content rather than soil organic matter (SOM) is reported. Therefore, we will use ‘humus content’ (%) as

the originally reported data, assuming that the trend of change reflects SOM content. Also, in Soviet and Ukrainian studies, the average ratio of humus content to soil organic carbon in black soils was conventionally taken as 1.88 (e.g., Ponomareva and Plotnikova, 1980; Chesnyak, 1983; Orlov, 1990).

Baliuk et al. (2021) made an assessment of the soil status and soil threats in Ukraine. They estimated that the loss of humus (and hence SOM) and nutrients occurs on 43% of arable land, soil compaction on 39%, surface crust formation on 38%, water erosion (both surface and linear) on 20%, acidification on 14%, waterlogging on 14%, deflation on 11%, contamination with radionuclides on 11%, pesticide contamination on 9%, heavy metal contamination on 8%, salinization and alkalization on 4% of the total arable land area in Ukraine. Svitlychnyi et al. (2022) emphasized the significant role of water erosion, particularly surface wash and linear erosion on bare soils, which occurs following rainfall and snowmelt, and leads to a decline of land productivity and increase of soil degradation. Water erosion affects approximately 38.4% of agricultural land across Ukraine, with 40% of arable land being eroded (Baliuk et al., 2010). Among these eroded lands, 4.5 million hectares are classified as moderately or strongly eroded soils, including 68,000 hectares where the humus horizon has been completely lost (Baliuk and Medvedev, 2012). In the 1980s-1990s, the estimated average annual soil loss from arable lands in Ukraine exceeded 15 t ha^{-1} , ranging from 5 to $30 \text{ t ha}^{-1} \text{ yr}^{-1}$. The average annual loss of humus is 0.5 t ha^{-1} , and nutrients are lost at a rate of 0.6 t ha^{-1} , which is not compensated by the use of fertilizers (Baliuk et al., 2009).

In addition to water erosion, other detrimental processes relate to the intensification of soil cultivation and the associated loss of SOM and degradation of the soil structural-physical properties of Chernozems. Next, large and heavy machinery have induced (sub)soil compaction, leading to decreased water permeability and reduced water release in the top soil.

In natural soils, the process of SOM build-up typically prevails over SOM mineralization, gradually accumulating organic matter. However, in intensively cultivated arable soils, this balance shifts towards intensified mineralization, reducing SOM content, which leads to increased risks of soil degradation. The cause of SOM loss in Chernozems thus stem from increased soil cultivation practices, growth of annual crops without cover crops, and possibly climate change. There is a need for crop rotations involving cereals and cash crops and cover crops, combined with minimum tillage. In addition, there is a need for appropriate inputs of NPK fertilizers, depending on soil fertility status and crop rotations. Where available, animal manure should be utilized by preference (Baliuk and Medvedev, 2012).

Estimates suggest that Ukraine's Chernozem soils have lost a significant portion of their SOM and humus content over the past century. Chesnyak (1983) estimated an average loss of 35-40% of the original humus content. In the 1930s, chernozems were reported to contain 6 to 9% humus, but these levels decline to below 6% in 1980s (Chesnyak et al., 1983; Nosko, 1987; Nosko et al., 1988, 1992). Later studies (Baliuk and Medvedev, 2012; Baliuk et al, 2021) found that over 140 years since the first measurements of humus content by V.V. Dokuchaev, the loss of humus (and hence SOM) in the Forest-Steppe region averaged 22% and in the Steppe region averaged 19.5%. The largest losses occurred during the 1970s, when the area planted with annual crops such as sugar beets and sunflowers significantly increased, combined with soil tillage after harvest. However, some SOM loss was mitigated by the application of manure combined with mineral fertilizers (which also stimulated crop growth, including crop residues, which were returned to soil). In subsequent years, reduced fertilizer use led to a gradual decrease in humus content (and consequently SOM content), from a mean of 3.36% between 1986 and 1990 to a mean of 3.14% between 2006 and 2010. It should be noted that the decline in livestock number decreased the need for straw as bedding material; instead the straw was returned to soil, which helped to slow down the decline in SOM content. However, it has been recently stated based on the results of long-term agrochemical monitoring (Romanova et al., 2022; Romanova, 2023) that the humus content (and hence SOM content) of Ukrainian agricultural soils continued declining to 3.07% (2016–2020) on average across Ukraine (**Supplementary Fig. 14**), indicating ongoing SOM degradation through net mineralization and nutrient removal via harvested crops (i.e., mining). “

...

“To sum up, years of intensive soil cultivation, growth of annual crops without cover crops and with the near absence of organic fertilizer use, have contributed to significant SOM losses in Ukrainian soils and to soil degradation. The latter was related also to soil erosion and soil compaction. To improve the SOM status of Chernozems in Ukraine, it is recommended to incorporate legumes, alfalfa and perennial grass crops into crop rotations and to strictly prohibit the burning of straw and crop residues. Effective measures to increase both the quantity and quality of SOM in soil include maintaining an organic matter balance through the use of animal and green manures, as well as recycled bio-based waste, minimizing soil disturbance (e.g., no-till, strip-till, mini-till), optimizing crop rotation structures, mulching with plant residues and applying chemical ameliorants (where it is required).”

[R2.2] 2. The inputs of N from biological fixation are mentioned a few times but not estimated. I find that the topic is inadequately addressed. How much of the considered crops are grown in rotations with biological N fixation? What is a rough estimate of these N inputs over the course of a crop rotation, or as a spatial average over Ukrainian cropland? Perhaps no change is needed in the modeling, but at least a rough estimate and a comment on it would be suitable.

Response to [R2.2]. We address biological nitrogen fixation (BNF) in two ways: Firstly we consider BNF by free-living organisms in soils. Secondly, we consider BNF by legumes as part of crop rotations.

We recognize that **BNF by free-living soil organisms** can contribute to nitrogen inputs. However, there are no specific data for Chernozems to accurately estimate this. We agree that, since it occurs, it should be estimated in some way. Therefore, in the revised version, we included an average BNF contribution by free-living organisms of 5 kg N ha⁻¹, as a likely feasible rate based on the review by Herridge *et al.* (2008). That study noted that reported rates are highly variable and that the amounts of N₂ fixed by other symbiotic, associative, and free-living bacteria in cereal and oilseed systems are very difficult, if not impossible, to quantify.

Accordingly, we have added the following to in the Methods section:

*“Although symbiotic biological N fixation is often not reported for maize, sunflower and wheat³⁰, N fixation by free-living organisms is assumed to occur in all the studied cropping systems in Ukraine. Since no specific data for Chernozems are available, we included an average N fixation contribution by free-living organisms of 5 kg N ha⁻¹ yr⁻¹, as a feasible rate based on the review by Herridge *et al.*³⁰. Available evidence suggests insignificant use of legumes in crop rotations in Ukraine.*

We included nutrient inputs via organic (including manure) and synthetic fertilizers, atmospheric inorganic N deposition and N fixation by free-living soil organisms, thereby neglecting inputs via soil weathering and mineralization.”

BNF by legumes as part of crop rotations. Please see the existing “*Text 3.3: Accounting for crop rotation practice*”, where we have briefly mentioned BNF by crops. Unfortunately, including legumes in the crop rotations of staple crops is still a rare practice in Ukraine. No data are currently available to support an estimate.

[R2.3] 3. Although the discussion of humus content in the supplementary is interesting and highly relevant, I would suggest that the authors connect this part of the text to more modern soil science concepts. At least preface this part with a short discussion of what "humus" is and isn't. It would be useful if the term is put into context of modern soil organic matter references and terminology (e.g., Sutton & Sposito 2005 <https://doi.org/10.1021/es050778q>; Lehmann & Kleber 2015 <https://doi.org/10.1038/nature16069>).

Response to [R2.3]. We find the suggestion useful. Indeed, the situation is challenging since some nations refer to humus while others refer to soil organic matter (SOM), while the definition of what humus is also varies between countries. We have written a preface in the Supplementary “**Text 2.2: Soil degradation**

and the current state of soils in Ukraine”, to clarify the situation with regard to SOM and humus. It now reads:

“Soil organic matter (SOM) as key indicator of soil quality. Commonly, a distinction is made between mineral soils and organic soils. Organic soils (or peat soils) have a very high organic matter content (>20% in at least the top 40 cm) and naturally occur under very wet conditions; the area of peat soils in Ukraine is very small and therefore peat soils are not discussed here further. Mineral soils consist of mineral compounds (including sand, silt and clay), organic matter and soil organisms. On a volume basis, minerals make up about 45-50%, soil organic matter (SOM) about 2 to 5%, soil organisms (including bacteria and fungi) ~1%, while the remaining 40-50% are soil pores filled with air and water. Soil organic matters gives mineral soils often a dark color; the darker the soil the higher the SOM content. The dark color of the SOM-rich and dominant soil type in Ukraine is reflected in the name: Chernozems, which means black soils. About 60% of the total agricultural land and about 73% of the arable land in Ukraine are covered by Chernozems. These deep soils are highly fertile, have neutral soil pH (pH 6.5 to 8.0) and are productive naturally, because of the parent material, the climatic conditions, natural grassland vegetation and soil formation processes. Conversely, soils with low SOM content are considered to be less fertile and are vulnerable to erosion and degradation (Russell, 1973; Lehmann and Kleber, 2015).

The soil organic matter content and soil pH are considered to be the most important indicators for soil quality. Soil scientists have for long tried to unravel the process dynamics of organic matter and its dark color in soils. The traditional concept of "humus" as a chemically distinct, persistent material formed through the synthesis of large-molecular-weight "humic substances" via humification of organic compounds has been fundamentally challenged by modern soil science. Historically, humus was perceived as a stable end-product of the decomposition of crop and root residues, animal manure and soil organisms. It was thought to be dominated by recalcitrant macromolecules which can be isolated through alkaline extraction methods (Lehmann and Kleber, 2015). However, advances in analytical techniques, such as nuclear magnetic resonance spectroscopy and mass spectrometry, reveal that these "humic substances" are artifacts of extraction procedures rather than intrinsic soil components. Instead, SOM is now understood as a dynamic continuum of progressively decomposing organic compounds, influenced by microbial activity, mineral interactions and environmental conditions (Sutton and Sposito, 2005; Lehmann and Kleber, 2015). This emergent view aligns with the "soil continuum model" (SCM), which emphasizes microbial accessibility and protection mechanisms over inherent chemical recalcitrance. Sutton and Sposito (2005) further dismantle the traditional paradigm by proposing that humic materials are transient, supramolecular associations of low molecular mass components stabilized by hydrophobic and hydrogen-bonding interactions, rather than discrete polymers. Consequently, the term "humus" in modern terminology reflects not a chemically unique entity but a heterogeneous mixture of organic fragments at varying decomposition stages, integrated with microbial biomass and mineral matrices. This shift underscores the need to abandon outdated "humification" frameworks and prioritize research on microbial ecology, spatial architecture and solubility dynamics to advance predictions of soil carbon cycling, water quality and climate feedbacks.”

Reviewer #3 (Remarks to the Author):

[R3.1] The manuscript "Nutrient asymmetry challenges the sustainability of Ukrainian agriculture" by S. Medinets et al. provides an incredibly data-rich and well elaborated analysis of the development of crop production in the Ukraine during the last decades, including the impact of important geopolitical events, such as the independence from, and the invasion by Russia. The results and recommendation are justified and pertinent in view of the during- and post-war recovery of the Ukraine and the important role of the Ukraine in (national and) global food security.

Main and supplementary information are well written and all graphical exhibitions are carefully made and add to the content of the manuscript. The amount of supplementary information is appreciated, but has also a downside. A reader who wishes to follow the details, this poses a challenge as it requires continuous going back and forth between main and supplementary text. For readers

who wish to only read the main text, the numerous references to the supplementary information might be disrupting the ‘flow’ of reading. I do not suggest to remove supplementary elements, and see this more as an editorial challenge to allow more information into the main piece under the constraints of the article type.

Response to [R3.1]. We thank the reviewer for their positive feedback and for highlighting these challenging points related to the pros and cons of using extended Supplementary material. Our idea is to keep the main text of the article short for easy digestion, focusing on the key messages, with all the details provided in the Supplementary for more curious readers.

[R3.2] My key request for improving the manuscript would be a critical assessment of the role of livestock. The role of manure as fertiliser in the assessed period has been well described. However, a few questions remain unclear: i/ what fate does the non-use manure have? Is it fully lost to the environment (how?).

Response to [R3.2]. Many thanks for your questions, which reflect our own. Unfortunately, there are no definitive answers. It is likely that it was dumped somewhere, i.e. presumably lost to the environment. It now reads in the Supplementary Text 2.1:

*“We estimated that only 10% of the total amount of manure N excreted was applied to agricultural fields in 2021, while the rest (about 90%) remained unutilized (no records on its further fate are available), so that it was probably wasted, being emitted to the atmosphere as ammonia (NH_3), di-nitrogen (N_2) and nitrogen oxides (N_2O , NO_x), as well as run-off and leached to the hydrosphere, thereby contributing to environmental pollution (see **Supplementary Fig. 10**).”*

[R3.3] what would e.g. be the impact on nitrogen deposition rates?

Response to [R3.3]. In the revised version, we included the atmospheric N deposition modeled by EMEP-MSC-W for N balance and N use efficiency calculations. See the revised Method section (below) and the new Supplementary Figure 4-x.:

“We used mean annual atmospheric deposition rates of total inorganic N compounds, i.e., the sum of reduced (NH_3 and NH_4^+) and oxidized (NO , NO_2 , NO_3^- , N_2O_5 , HNO_3 , etc.) forms, which were simulated by the EMEP-MSC-W model for 1990-2022²⁸ (Supplementary Fig. 5); these EMEP-reported data were used for both country and county scales. However, we must emphasize the limited applicability of current atmospheric model results, including EMEP-MSC-W, due to: (i) the inclusion of deposition estimates for inorganic N forms only, (ii) reliance on data largely based on estimated N emissions reported by countries that lack, or have poorly developed, national emission monitoring networks, including Ukraine, rather than on measured data, (iii) the lack of any estimates for organic N deposition, which have been shown to significantly contribute to total atmospheric deposition¹⁶, including in studies conducted in Ukraine^{17,18} (Supplementary Text 3.1). Incorporating organic N deposition requires more field studies with further in-depth investigation of the underlying chemical mechanisms before being included in atmospheric models. Meanwhile, atmospheric deposition of P and K are generally assumed to be negligible^{13,29}; however, more field measurements worldwide are needed to confirm or challenge these assumptions in specific regions (Supplementary Text 3.1).”

In addition, we now also discuss the urgent need to monitor atmospheric organic N deposition and to include it in atmospheric models, as well as the need to monitor P and K deposition, which are not always negligible as conventionally suggested. We include this at the Supplementary Section **“Text 3.1: Accounting for the total atmospheric deposition of N, P and K, including organic constituents”**:

“It is well shown that organic N may significantly contribute to the total N deposition, often by more than 30% (e.g., Miyazaki et al., 2010; Cape et al., 2011; Cornell, 2011; Medinets et al., 2012, 2020; Medinets, 2014; Li et al., 2023). The highest organic fraction of the total has been reported by Medinets et al. (2020) on average at around 70% for monitoring sites (cropland, garden and natural wetland) located in Odesa region (Ukraine) over the period of 2011-2019. The authors assumed that the high ratios of organic fractions in N deposition observed in this study area might be attributed to several factors: (i) wind-induced and agri-management-induced soil dust formation from Black soils with high organic matter content

(Medinets et al., 2016), (ii) the influence of organic aerosols of marine origin due to proximity to the sea (Altieri et al., 2016) and (iii) potentially region-specific biogeochemical conditions that favor the emission or atmospheric formation of organic compounds. Intensive agricultural practices (e.g., fertilizer application, tillage) that generate additional soil dust containing urea, amines, macromolecules, and humic-like substances (McKenzie et al., 2016) might explain the higher absolute content of total N, including organics, in samples collected from and near managed sites. However, it remains debatable how to disentangle primary deposition from re-deposition, as well as to determine whether the sources are locally originated or transferred from nearby areas, and which of these should be included. Moreover, it cannot be ruled out that microbial conversion from dissolved inorganic to particulate and organic N occurred within the total deposition samples during the fortnightly-monthly sampling period, despite the addition of biocide thymol to inhibit microbial growth. Nonetheless, further accurate investigations into organic N (and other nutrient depositions) are necessary in the near future to determine their contributions across different regions, enhance atmospheric models, and incorporate these findings into balance calculations.

Based on available field data for southern Ukraine, mean annual total (organic and inorganic) atmospheric N, P and K deposition was $25.5 \pm 2.2 \text{ kg N ha}^{-1}$ (with an estimated contribution of around 30% from dissolved inorganic N and the remainder presumably from organic N fractions), $2.5 \pm 0.6 \text{ kg P ha}^{-1}$ (with a 56% contribution from dissolved inorganic P) and $3.1 \pm 1.2 \text{ kg K ha}^{-1}$ (inorganic only) [values for Lower Dniester catchment, Odeska county, 2019–2021; Medinets et al., 2020, 2024; Medinets et al., unpublished data]. For example, these values accounted for 21%, 24% and 21% of the mean total N, P and K fertilizer ($119 \pm 6 \text{ kg N ha}^{-1}$, $10.5 \pm 0.5 \text{ kg P ha}^{-1}$ and $14.8 \pm 0.3 \text{ kg K ha}^{-1}$; **Fig. 1**) applied to wheat over the same period, respectively. It is noteworthy that the mean rates presented above for dissolved inorganic P (1.4 kg P ha^{-1}) are equal to the previously global mean for $\text{PO}_4\text{-P}$ deposition reported by Tipping et al. (2014). Overall, these estimates, based on upscaling of measured deposition, demonstrate how atmospheric deposition of both P and K can be significant for areas receiving little or no input from fertilizers. Hence atmospheric deposition of both P and K needs to be considered alongside N (including the organic contributions) when calculating complete nutrient budgets.”

[R3.4]ii/ the 'return' to coupled crop-livestock systems is suggested. However, Suppl. Text 3.2 suggests that this has not only positive effects (higher use of manure as fertiliser) but also contributed to a decline of SOM.

Response to [R3.4]. It is likely that you misunderstood the point in the previous Suppl. Text 3.2 (now it is the Supplementary Text 3.3: Accounting for crop residue management), which is about crop residue management.

The application of 'wasted' (unutilized) manure has a double-win effect – minimizing environmental impacts and improving SOM, which is suggested as an essential measure for Chernozems, alongside other organic fertilizer use (see details in paragraph sub-titled “Manure application deficits” in the Suppl. Text 2.1: Mean N, P and K inputs with fertilizers and the outputs with crop yields). However, it should be applied according to regulations to avoid environmental impacts such as leaching, etc.

We have also highlighted the importance of manure recycling in the new **Implication for food security** section:

“Moreover, the under-utilization of manure, caused by the decoupling of livestock and crop systems, represents a missed opportunity for circular nutrient economy practices (Supplementary Fig. 10-13, Text 2.1) that could reduce dependence on synthetic fertilizers while lowering reactive nitrogen and greenhouse gas emissions. Addressing these challenges requires improved access to fertilizers and long-term investments in integrated nutrient management strategies.”

[R3.5] This should be included in the main analysis upfront. iii/ What is the share of cereals used for feed during the assessed period (beyond crop residues)? Livestock can only have a positive net input of nutrients to crops cultivated for export or food if nutrient are imported with feed, otherwise the return to the soils is inevitably less than extraction for feed.

Response to [R3.5]. Thanks for this note. Perhaps we should have indicated somewhere that, in this manuscript, we do not include any crops (such as cereals or corn) grown for forage purposes in this study, at least not for animal forage within Ukraine. Now we added such a note to the Methods:

“; crops grown for forage were not included in our analysis”.

[R3.6] Furthermore, the suggestion of a "Smart Fertilizer Planner tool" should be complemented by an assessment of existing tools of similar purpose, for example under the EU Common Agricultural Policy. If such tools could be extended, it would greatly increase the feasibility of such a tool. Importance of SOM for soil fertility was demonstrated and emphasised. Why should the SFP be based on a 'zero nutrient balance'? Wouldn't it be important to aim to re-build SOM in the soils, and should for that purpose not also C included into the SFP, which would allow a better consideration of the importance of crop residues management?

Response to [R3.6]. Thanks for this valuable comment! Initially, SFP is suggested as a simple tool for Ukrainian farmers, ideally with county- or region-specific features such as NPK removal rates for specific crop varieties used in the region (the most critical parameters) as well as region-specific parameters, such as soil type (texture, pH and SOM) and climatic conditions (precipitation and temperature). There are certainly some free and commercial tools available to farmers for various purposes, but they are often not specific to crop varieties in a given region and do not cover Eastern European countries outside the EU. However, in principle, with evidence-based data, they could be complimented to SFP or adapted as a separate tool for use in Eastern European countries.

We completely revised the section **“Text 2.8: Smart Fertilizer Planner”** of the Supplementary as well as added information on other tools available:

“Principle and usage. *The proposed region-tailored, variety-specific Smart Fertilizer Planner (SFP) tool is based on a simplified nutrient balance principle, incorporating a ‘5%-surplus’ for N and ‘zero-surplus’ approach for P and K. This balance is calculated between the total input, which includes synthetic and organic fertilizer applied by farmers alongside the sum of mean atmospheric inorganic N deposition and N fixation by free-living organisms (provided as a ‘constant’; N deposition is updated upon new data become available in EMEP reports), and the total output, which includes crop yield (and by-product if straw is removed from the field). Both inputs and outputs are expressed in pure elemental units: kg N ha⁻¹, kg P ha⁻¹, and kg K ha⁻¹.*

It is assumed that a 5% N surplus, together with variability in atmospheric nutrient deposition, including unaccounted organic NPK inputs, is roughly offset by unavoidable nutrient losses, which are higher for N and expected to be marginal for P and K (see 'Limitations and Assumptions' below).

The tool is suggested to be available as mobile app and a downloadable spreadsheet. It would provide farmers with easy-to-use yet precise estimation for NPK fertilizer input requirements tailored to their region and crop varieties, based on data from the top-performing crops over the past 3-5 seasons.

The SFP would help calculate annual field-scale nutrient balances and estimates nutrient use efficiency after harvest using yield data. A key element of the SFP is the collection of crop yields from varieties grown in a specific region, followed by analysis of their NPK content to develop a region-tailored, variety-specific NPK removal repository. This repository should be updated whenever new varieties are introduced in the region. Another important aspect is the analysis of NPK content in organic fertilizers applied (if relevant), to at least establish an initial inventory for the region or country. Ideally, this information should be updated regularly, or, as an advanced option, it could be provided on a regular basis by organic fertilizer suppliers. This approach can be implemented at field-to-regional levels, supporting farmers and policymakers in various counties and georegions for rain-fed cropping systems (i.e., including internationally beyond Ukraine). It can also be used for irrigated crops; in this case, the NPK content applied through irrigation water (i.e., NPK concentration in irrigated water multiplied by the amount of water applied) should be added to the input category for more precise calculations. Moreover, the SFP would be adaptable to entire crop rotation cycles, making fertilizer planning more accurate and comprehensive. In this case, it would account for nutrient surpluses or deficits from preceding crops to adjust applications for subsequent ones.

Limitations and assumptions. The proposed tool would generally assume that vegetation biomass benefits from crop residues returned to the field in previous years. In a simplest approach it may be estimated that up to 5%-surpass for N together with variation of atmospheric deposition of NPK, including unaccounted rates for organic NPK deposited (**Supplementary Text 3.1**), are roughly offset by unavoidable N emissions and potential NPK leaching or runoff (so long as urea is not used). However, the performance of the SFP may be substantially improved if evidence-based estimates for organic nutrient deposition and region-specific fertilizer-induced emission factors are available. Emissions of N are considered the primary loss pathway in black soil agroecosystems, while leaching of nitrate, phosphorus and potassium compounds is supposed to be less critical, though migration to lower soil horizons may occur (Kovda and Rozanov, 1988; Krupenikov et al., 2011; Medinets et al., 2016, 2021; Boincean and Dent, 2019).

Prerequisites and requirements. To estimate NPK removal by crop yield for specific crop varieties, a database of NPK content for currently used varieties is essential. This database should be created and regularly updated as new varieties are introduced. Such information must be accessible to farmers, either through regional advisory services, online repository or mobile app.

The minimum requirements for successful use of the SFP tool would include:

- (i) Replacing straight urea to prevent significant nitrogen losses via NH_3 volatilization.
- (ii) Returning crop residues to the field as a source for the next year's vegetation biomass or, alternatively, accurately calculating removed amounts and replenishing them with (organic) fertilizers.
- (iii) Ensuring availability of NPK content data for key crop varieties.
- (iv) Accessing appropriate soil testing to monitor exchangeable P and K concentrations.

Further development. The SFP is intended as a simple flexible solution that should be developed over time. With further research and advancements in technology, more precise, region-specific guidance could be developed, taking into account local pedoclimatic and socioeconomic conditions.

Other tools available. Also, there are some other free and commercial tools available to farmers for various purposes, they are, sometimes, specific to the georegion, but often not specific to crop varieties in a given region and, unfortunately, do not cover Eastern European countries outside the EU. However, with region-tailored, variety-specific evidence-based data, they could be complimented to SFP or adapted as a separate tool for use in Eastern European countries.

Among them are the following:

- (i) Farm Sustainability Tool for Nutrients (FaST) – a digital tool developed under the EU CAP to help farmers create nutrient management plans. It integrates existing (mainly satellite) data with manual inputs to provide customized recommendations on crop fertilization, aiming to deliver both economic and environmental benefits. It is intended for region-specific implementation across EU countries with governmental support.
- (ii) NUTRI-CHECK NET Platform – an EU-funded initiative offering a three-step approach—Plan, Check & Adjust, and Review—for nutrient management. It essentially serves as an inventory of various tools and services to support farmers, agronomists, researchers, and the broader agricultural industry in sharing knowledge and exploring available N, P, and K nutrient management approaches for wheat, maize, and potato across Europe.
- (iii) Planning Land Applications of Nutrients for Efficiency and the Environment (PLANET) – a nutrient management decision support tool for use by farmers and advisers in England, Wales and Scotland. It supports field-level nutrient planning and helps assess and demonstrate compliance with the Nitrate Vulnerable Zone (NVZ) rules.
- (iv) CAFRE Crop Nutrient Calculator – a tool developed by the College of Agriculture, Food and Rural Enterprise (CAFRE) of Northern Ireland, based on soil nutrient analysis. It helps farmers create precise fertilization plans that meet crop nutrient requirements without excess. The tool emphasizes the importance of soil pH and nutrient stocks, and promotes low-emission techniques for slurry spreading.”

[R3.7] The development of the area/share of non-used UAA should be better explained more in detail and

earlier in the manuscript. I learn details when reading Supplementary Text 2.9 but it is already important for the understanding of Supplementary Figure 1.

Response to [R3.7]. Many thanks for this point; indeed, it might be confusing. We have tried to clarify what portion of the UAA is allocated to each crop versus the total UAA by slightly rephrasing the main text, revising the title of Supplementary Figure 1, and adding a description of what constitutes the total UAA. It now reads:

“Supplementary Figure 1. Utilized agricultural areas (UAA) allocated to wheat, sunflower and maize cultivation (a) and their respective shares of Ukraine’s total UAA (b) for the years 1980, 1986, 1990 and 2000-2023 (SSSU, 2024). The unit th. ha represents thousand hectares. Dotted lines in 2022 show the start of the large-scale war; dash-dotted lines in 1990 show the end of the USSR. The total UAA refers to the total area of land used for agricultural activities, including arable land, permanent crops and permanent grassland or pasture.”

Minor comments.

[R3.8] Page 3, first paragraph: Starting a sentence ‘This indicates’ that follows an observation for maize and what and the opposite for sunflower makes it difficult to understand what exactly is meant to indicate soil nutrient depletion.

Response to [R3.8]. Many thanks for this note. We rephrased this sentence to make it clearer:

“Thus, NPK use efficiencies surpassing 90-100% indicate continuous soil nutrient depletion¹⁹, particularly for K and P, but also for N, as evidenced by steady decline in soil organic matter content during this period (Fig. 1; Supplementary Fig. 14, Text 2.6).”

[R3.9] Supplementary Table 2. Broiler not Brolier.

Response to [R3.9]. Many thanks. Fixed now.

Reviewer #4 (Remarks to the Author):

[R4.1] This paper could be treated as a short comment rather than a full-length research article. The objective is to analyze the imbalanced input and output of nutrients in Ukraine during the past decades. The topic is vital since nutrients serve as the fundamental resource for food production and environmental degradation. However, this paper only analyzed a few indicators that can be obtained directly from open statistics, failing to reveal the underlying causes or subsequent effects of the trends.

Response to [R4.1]. Many thanks for highlighting the importance of the topic and for your critical feedback on the implementation.

We have now highlighted more clearly that our study represents new insights. Ours is the first report of long-term trends for NPK balance and nutrient use efficiencies (NUE) for three staple crops in Ukraine, at county-to-country scales, across a 43-year period of substantial economic and political transformation (1980–2023).

Novelty:

(i) We are the first to chart the course of asymmetries between nitrogen (N), phosphorus (P) and potassium (K) nutrients including spatial patterns across Ukraine as one of the most important grain exporting countries globally, and over four decades:

- late Soviet period,
- post-independence era with peak production prior to the war,
- the ongoing war period.

(ii) We integrate the dynamics of soil nutrient balances for Ukraine considering each of the main components, including synthetic fertilizers, organic fertilizers (largely manure), atmospheric N deposition and N fixation by free-living organisms, and the implications of these imbalances for food security and wider sustainability.

(iii) We provide a unique analysis of the way in which the Russo-Ukrainian war has fundamentally changed nutrient balances for Ukraine, leading to emerging risks for the sustainability of Ukrainian soils and food supply.

(iv) Based on our diagnosis of nutrient imbalances and their risks, we identify an innovative (both practical and feasible) set of options for actions for more sustainable nutrient management that would set Ukraine on a path to continue a strong role in supporting global food security, while recognizing the important benefits for environmental sustainability.

Importance:

Our study contributes to national and global food security (SDG Goal 2: Zero Hunger by 2030), supports the transition to a circular and net-zero economy by combating nutrient pollution and reducing greenhouse gas emissions (aligned with Target 7 of the Global Biodiversity Framework - to halve nutrient pollution by 2030) and supports multiple SDGs (Goals 6, 12, 14, and 15).

We have modified the manuscript to emphasize these points more clearly including:

- a) Emphasized the scientific significance of our investigations, beyond the context of the war itself in the introduction. See our response to Reviewer 1 [comment R1.1].
- b) Made more clear about the uniqueness of the situation of Ukraine. See our response to Reviewer 2 [comment R2.1 and R2.3]
- c) Provided a new section on ‘Food Security Implications’. See our response to Reviewer 1 [comment R1.6] and Reviewer 4 [comment 4.2]
Further amplified our description of practical ‘Actions’ by adding challenges cause by humanitarian crisis in Ukraine and the suggestions on potential incorporation of Integrated Nutrient Management Plan for Ukraine into the agriculture sector recovery under the Ukraine Recovery and Reconstruction. See our response to Reviewer 1 [comment R1.3, R1.4, R1.5].
- d) We strengthened the analysis of the importance of nutrient management in Ukraine for the global food supply chain, including the effects of the war. See our response to Reviewer 1 [comment R1.1 and R1.5]
- e) We now included two more input categories (atmospheric N deposition and N fixation by free-living organisms) to make the balance and NUE calculations more accurate. Please see our response to Reviewer 2 on N fixation [comments R2.2], Reviewer 3 on atmospheric deposition [comment R3.3] and Reviewer 4 [comment R4.1]
- f) We discussed the urgent need to monitor atmospheric organic N deposition and to include it in atmospheric models, as well as the need to monitor P and K deposition, which are not always negligible as conventionally suggested. See our response to Reviewer 3 [comment R3.3]
- g) We expanded Supplementary “Text 2.2: Soil degradation and the current state of soils in Ukraine” by including concerns about dehumification and mineralization trends, and by clarifying the current understanding of soil organic matter (SOM) and the humus concept to better interpret the reported Ukrainian data on soil humus. See our answer to Reviewer 2 [comments R2.1 and R2.3] and Reviewer 4 [comment R4.2]
- h) We clarified information on unutilized manure and addressed Reviewer 3 concerns related to manure use. See our response to Reviewer 3 [comment R3.2 and comment R3.4]
- i) We completely revised the section “*Text 2.8: Smart Fertilizer Planner*” of the Supplementary as well as added information on other tools available. See our response to Reviewer 3 [comment R3.6] and Reviewer 4 [comment 4.5]
- j) We added a new Figure 5 ‘Mean annual deposition ($\text{kg N ha}^{-1} \text{ yr}^{-1}$) of reduced inorganic nitrogen, oxidized inorganic nitrogen and the total inorganic nitrogen in Ukraine for 1980, 1986, 1990, 1995 and 2000-2023’ and revised all main text and Supplementary Figures showing N balance and N use efficiencies using the updated calculations (Figs. 1, 2; Supplementary Figs. 16-18, 20-26)

Indeed, nutrient balance and nutrient use efficiencies are key agroecological indicators of the efficacy and sustainability of cropping systems. In this study, they were primarily calculated using open-access statistics, and in the revised version, we have further incorporated atmospheric inorganic N deposition (modeled with EMEP-MSC-W) and estimates for N fixation by free-living organisms (Herridge et al., 2008):

- atmospheric inorganic N deposition using the EMEP-MSC-W model. See the revised Method section (below) and the new Supplementary Figure 4-x.:

“We used mean annual atmospheric deposition rates of total inorganic N compounds, i.e., the sum of reduced (NH_3 and NH_4^+) and oxidized (NO , NO_2 , NO_3^- , N_2O_5 , HNO_3 , etc.) forms, which were simulated by the EMEP-MSC-W model for 1990-2022²⁸ (Supplementary Fig. 5); these EMEP-reported data were used for both country and county scales. However, we must emphasize the limited applicability of current atmospheric model results, including EMEP-MSC-W, due to: (i) the inclusion of deposition estimates for inorganic N forms only, (ii) reliance on data largely based on estimated N emissions reported by countries that lack, or have poorly developed, national emission monitoring networks, including Ukraine, rather than on measured data, (iii) the lack of any estimates for organic N deposition, which have been shown to significantly contribute to total atmospheric deposition¹⁶, including in studies conducted in Ukraine^{17,18} (Supplementary Text 3.1). Incorporating organic N deposition requires more field studies with further in-depth investigation of the underlying chemical mechanisms before being included in atmospheric models. Meanwhile, atmospheric deposition of P and K are generally assumed to be negligible^{13,29}; however, more field measurements worldwide are needed to confirm or challenge these assumptions in specific regions (Supplementary Text 3.1).”

- biological nitrogen fixation by free-living organisms, based on the approach of Herridge et al. (2008). See the revised Method section:

“Although symbiotic biological N fixation is often not reported for maize, sunflower and wheat³⁰, N fixation by free-living organisms is assumed to occur in all the studied cropping systems in Ukraine. Since no specific data for Chernozems are available, we included an average N fixation contribution by free-living organisms of $5 \text{ kg N ha}^{-1} \text{ yr}^{-1}$, as a feasible rate based on the review by Herridge et al.³⁰. ”

Dataset of N balances and N use efficiencies for all studied crops as well as related Figures (Figs. 1, 2; Supplementary Figs. 16-18, 20-22, 23-26) have been updated. In addition, we have also now highlighted about gaps in our approach more clearly. Specifically, we now also discuss the urgent need to monitor atmospheric organic N deposition and to include it in atmospheric models, as well as the need to monitor P and K deposition, which are not always negligible as conventionally suggested. We include this at the Supplementary Section **“Text 3.1: Accounting for the total atmospheric deposition of N, P and K, including organic constituents”**:

“It is well shown that organic N may significantly contribute to the total N deposition, often by more than 30% (e.g., Miyazaki et al., 2010; Cape et al., 2011; Cornell, 2011; Medinets et al., 2012, 2020; Medinets, 2014; Li et al., 2023). The highest organic fraction of the total has been reported by Medinets et al. (2020) on average at around 70% for monitoring sites (cropland, garden and natural wetland) located in Odesa region (Ukraine) over the period of 2011-2019. The authors assumed that the high ratios of organic fractions in N deposition observed in this study area might be attributed to several factors: (i) wind-induced and agri-management-induced soil dust formation from Black soils with high organic matter content (Medinets et al., 2016), (ii) the influence of organic aerosols of marine origin due to proximity to the sea (Altieri et al., 2016) and (iii) potentially region-specific biogeochemical conditions that favor the emission or atmospheric formation of organic compounds. Intensive agricultural practices (e.g., fertilizer application, tillage) that generate additional soil dust containing urea, amines, macromolecules, and humic-like substances (McKenzie et al., 2016) might explain the higher absolute content of total N, including organics, in samples collected from and near managed sites. However, it remains debatable how to disentangle primary deposition from re-deposition, as well as to determine whether the sources are locally originated or transferred from nearby areas, and which of these should be included. Moreover, it

cannot be ruled out that microbial conversion from dissolved inorganic to particulate and organic N occurred within the total deposition samples during the fortnightly-monthly sampling period, despite the addition of biocide thymol to inhibit microbial growth. Nonetheless, further accurate investigations into organic N (and other nutrient depositions) are necessary in the near future to determine their contributions across different regions, enhance atmospheric models, and incorporate these findings into balance calculations.

Based on available field data for southern Ukraine, mean annual total (organic and inorganic) atmospheric N, P and K deposition was $25.5 \pm 2.2 \text{ kg N ha}^{-1}$ (with an estimated contribution of around 30% from dissolved inorganic N and the remainder presumably from organic N fractions), $2.5 \pm 0.6 \text{ kg P ha}^{-1}$ (with a 56% contribution from dissolved inorganic P) and $3.1 \pm 1.2 \text{ kg K ha}^{-1}$ (inorganic only) [values for Lower Dniester catchment, Odeska county, 2019–2021; Medinets et al., 2020, 2024; Medinets et al., unpublished data]. For example, these values accounted for 21%, 24% and 21% of the mean total N, P and K fertilizer ($119 \pm 6 \text{ kg N ha}^{-1}$, $10.5 \pm 0.5 \text{ kg P ha}^{-1}$ and $14.8 \pm 0.3 \text{ kg K ha}^{-1}$; **Fig. 1**) applied to wheat over the same period, respectively. It is noteworthy that the mean rates presented above for dissolved inorganic P (1.4 kg P ha^{-1}) are equal to the previously global mean for $\text{PO}_4\text{-P}$ deposition reported by Tipping et al. (2014). Overall, these estimates, based on upscaling of measured deposition, demonstrate how atmospheric deposition of both P and K can be significant for areas receiving little or no input from fertilizers. Hence atmospheric deposition of both P and K needs to be considered alongside N (including the organic contributions) when calculating complete nutrient budgets.”

We fully agree with you that national statistics are not the ideal dataset for such estimations; however, due to the extreme lack of field measurement data, there is currently no alternative - and this is one of the key points we emphasize in the manuscript. National statistics are widely used in many national-to-global analyses and model simulations.

We also agree that large-scale analyses often fail to reveal underlying causes, which may vary considerably across regions. Therefore, by retrieving advanced statistics at the county scale, we were able to demonstrate the asymmetry across counties/regions in Ukraine and highlight potential differentiating factors (see the updated Supplementary Fig. 20-22 and the updated Text 2.7) as well as temporal and spatial variation in NPK balances at country and county scales in Ukraine (see updated Supplementary Fig. 23-25).

[R4.2] The text looks like a report to the government rather than an academic paper that both scientists and the public will be interested in.

Response to [R4.2]. We intentionally designed the manuscript as a Brief Communication to keep it short, concise and streamlined, aiming to highlight key messages for the public, policymakers and scientists. Unfortunately, it is well known that longer manuscripts sometimes fail to deliver clear, actionable messages. Meanwhile, readers seeking more detailed insights are encouraged to explore the extensive Supplementary Materials provided (now substantially revised as well as updated with new sections, such as *Soil degradation and SOM loss* and *Soil organic matter (SOM) as key indicator of soil quality* within the Supplementary “**Text 2.2: Soil degradation and the current state of soils in Ukraine**”:

“Soil degradation and SOM loss. The Food and Agriculture Organization (FAO) has highlighted significant concerns regarding soil degradation in Ukraine, particularly the loss of SOM. An FAO report (2019) noted that soil erosion by wind and water affects approximately 13.4 million hectares, including 10.6 million hectares of arable land. Furthermore, the FAO indicated that degraded and unproductive arable land in Ukraine exceeds 20% (more than 6.5 million hectares) of the total arable land, with annual soil loss due to erosion ranging from 300 to 600 million tons (FAO, 2018). Soil degradation through SOM loss and erosion mainly occur as a result of annual soil cultivation, growth of annual crops, without winter crops or intercrops. As a result, soils are bare for large parts of the year, and then vulnerable to wind and water erosion.

It is noteworthy that conventionally in Soviet and later Ukrainian soil studies (up to the present), humus content rather than soil organic matter (SOM) is reported. Therefore, we will use ‘humus content’ (%) as

the originally reported data, assuming that the trend of change reflects SOM content. Also, in Soviet and Ukrainian studies, the average ratio of humus content to soil organic carbon in black soils was conventionally taken as 1.88 (e.g., Ponomareva and Plotnikova, 1980; Chesnyak, 1983; Orlov, 1990).

Baliuk et al. (2021) made an assessment of the soil status and soil threats in Ukraine. They estimated that the loss of humus (and hence SOM) and nutrients occurs on 43% of arable land, soil compaction on 39%, surface crust formation on 38%, water erosion (both surface and linear) on 20%, acidification on 14%, waterlogging on 14%, deflation on 11%, contamination with radionuclides on 11%, pesticide contamination on 9%, heavy metal contamination on 8%, salinization and alkalization on 4% of the total arable land area in Ukraine. Svitlychnyi et al. (2022) emphasized the significant role of water erosion, particularly surface wash and linear erosion on bare soils, which occurs following rainfall and snowmelt, and leads to a decline of land productivity and increase of soil degradation. Water erosion affects approximately 38.4% of agricultural land across Ukraine, with 40% of arable land being eroded (Baliuk et al., 2010). Among these eroded lands, 4.5 million hectares are classified as moderately or strongly eroded soils, including 68,000 hectares where the humus horizon has been completely lost (Baliuk and Medvedev, 2012). In the 1980s-1990s, the estimated average annual soil loss from arable lands in Ukraine exceeded 15 t ha^{-1} , ranging from 5 to $30 \text{ t ha}^{-1} \text{ yr}^{-1}$. The average annual loss of humus is 0.5 t ha^{-1} , and nutrients are lost at a rate of 0.6 t ha^{-1} , which is not compensated by the use of fertilizers (Baliuk et al., 2009).

In addition to water erosion, other detrimental processes relate to the intensification of soil cultivation and the associated loss of SOM and degradation of the soil structural-physical properties of Chernozems. Next, large and heavy machinery have induced (sub)soil compaction, leading to decreased water permeability and reduced water release in the top soil.

In natural soils, the process of SOM build-up typically prevails over SOM mineralization, gradually accumulating organic matter. However, in intensively cultivated arable soils, this balance shifts towards intensified mineralization, reducing SOM content, which leads to increased risks of soil degradation. The cause of SOM loss in Chernozems thus stem from increased soil cultivation practices, growth of annual crops without cover crops, and possibly climate change. There is a need for crop rotations involving cereals and cash crops and cover crops, combined with minimum tillage. In addition, there is a need for appropriate inputs of NPK fertilizers, depending on soil fertility status and crop rotations. Where available, animal manure should be utilized by preference (Baliuk and Medvedev, 2012).

Estimates suggest that Ukraine's Chernozem soils have lost a significant portion of their SOM and humus content over the past century. Chesnyak (1983) estimated an average loss of 35-40% of the original humus content. In the 1930s, chernozems were reported to contain 6 to 9% humus, but these levels decline to below 6% in 1980s (Chesnyak et al., 1983; Nosko, 1987; Nosko et al., 1988, 1992). Later studies (Baliuk and Medvedev, 2012; Baliuk et al, 2021) found that over 140 years since the first measurements of humus content by V.V. Dokuchaev, the loss of humus (and hence SOM) in the Forest-Steppe region averaged 22% and in the Steppe region averaged 19.5%. The largest losses occurred during the 1970s, when the area planted with annual crops such as sugar beets and sunflowers significantly increased, combined with soil tillage after harvest. However, some SOM loss was mitigated by the application of manure combined with mineral fertilizers (which also stimulated crop growth, including crop residues, which were returned to soil). In subsequent years, reduced fertilizer use led to a gradual decrease in humus content (and consequently SOM content), from a mean of 3.36% between 1986 and 1990 to a mean of 3.14% between 2006 and 2010. It should be noted that the decline in livestock number decreased the need for straw as bedding material; instead the straw was returned to soil, which helped to slow down the decline in SOM content. However, it has been recently stated based on the results of long-term agrochemical monitoring (Romanova et al., 2022; Romanova, 2023) that the humus content (and hence SOM content) of Ukrainian agricultural soils continued declining to 3.07% (2016–2020) on average across Ukraine (**Supplementary Fig. 14**), indicating ongoing SOM degradation through net mineralization and nutrient removal via harvested crops (i.e., mining).”

...

“Soil organic matter (SOM) as key indicator of soil quality. Commonly, a distinction is made between mineral soils and organic soils. Organic soils (or peat soils) have a very high organic matter content (>20% in at least the top 40 cm) and naturally occur under very wet conditions; the area of peat soils in Ukraine is very small and therefore peat soils are not discussed here further. Mineral soils consist of mineral compounds (including sand, silt and clay), organic matter and soil organisms. On a volume basis, minerals make up about 45-50%, soil organic matter (SOM) about 2 to 5%, soil organisms (including bacteria and fungi) ~1%, while the remaining 40-50% are soil pores filled with air and water. Soil organic matters gives mineral soils often a dark color; the darker the soil the higher the SOM content. The dark color of the SOM-rich and dominant soil type in Ukraine is reflected in the name: Chernozems, which means black soils. About 60% of the total agricultural land and about 73% of the arable land in Ukraine are covered by Chernozems. These deep soils are highly fertile, have neutral soil pH (pH 6.5 to 8.0) and are productive naturally, because of the parent material, the climatic conditions, natural grassland vegetation and soil formation processes. Conversely, soils with low SOM content are considered to be less fertile and are vulnerable to erosion and degradation (Russell, 1973; Lehmann and Kleber, 2015).

The soil organic matter content and soil pH are considered to be the most important indicators for soil quality. Soil scientists have for long tried to unravel the process dynamics of organic matter and its dark color in soils. The traditional concept of **“humus”** as a chemically distinct, persistent material formed through the synthesis of large-molecular-weight **“humic substances”** via humification of organic compounds has been fundamentally challenged by modern soil science. Historically, humus was perceived as a stable end-product of the decomposition of crop and root residues, animal manure and soil organisms. It was thought to be dominated by recalcitrant macromolecules which can be isolated through alkaline extraction methods (Lehmann and Kleber, 2015). However, advances in analytical techniques, such as nuclear magnetic resonance spectroscopy and mass spectrometry, reveal that these “humic substances” are artifacts of extraction procedures rather than intrinsic soil components. Instead, SOM is now understood as a dynamic continuum of progressively decomposing organic compounds, influenced by microbial activity, mineral interactions and environmental conditions (Sutton and Sposito, 2005; Lehmann and Kleber, 2015). This emergent view aligns with the “soil continuum model” (SCM), which emphasizes microbial accessibility and protection mechanisms over inherent chemical recalcitrance. Sutton and Sposito (2005) further dismantle the traditional paradigm by proposing that humic materials are transient, supramolecular associations of low molecular mass components stabilized by hydrophobic and hydrogen-bonding interactions, rather than discrete polymers. Consequently, the term “humus” in modern terminology reflects not a chemically unique entity but a heterogeneous mixture of organic fragments at varying decomposition stages, integrated with microbial biomass and mineral matrices. This shift underscores the need to abandon outdated “humification” frameworks and prioritize research on microbial ecology, spatial architecture and solubility dynamics to advance predictions of soil carbon cycling, water quality and climate feedbacks.”

...

“To sum up, years of intensive soil cultivation, growth of annual crops without cover crops and with the near absence of organic fertilizer use, have contributed to significant SOM losses in Ukrainian soils and to soil degradation. The latter was related also to soil erosion and soil compaction. To improve the SOM status of Chernozems in Ukraine, it is recommended to incorporate legumes, alfalfa and perennial grass crops into crop rotations and to strictly prohibit the burning of straw and crop residues. Effective measures to increase both the quantity and quality of SOM in soil include maintaining an organic matter balance through the use of animal and green manures, as well as recycled bio-based waste, minimizing soil disturbance (e.g., no-till, strip-till, mini-till), optimizing crop rotation structures, mulching with plant residues and applying chemical ameliorants (where it is required).”

Also, we would like to underline that this study covers three distinct periods in the evolution of Ukrainian agriculture: (i) the last 10 years of the pre-independence period (1980–1990), (ii) the 30 years of the independence period (1991–2021) prior to the large-scale war, and (iii) the years of the war's impact (2022–2023). Thus, we discuss the impacts of nutrient imbalances across these contrasting periods over the timeline of 1980–2023, which extends far beyond the context of the war that began in February 2022.

Ukraine's significant contribution to the production and export of the studied crops (wheat, maize, and sunflower) makes it important for global food security. Meanwhile, unsustainable nutrient management (nutrient imbalance and the combination of imbalances, including spatially, both between inputs and outputs, and different nutrients) - varying across different periods, asymmetrically improving during the independence era, and aggravated by the war - makes this issue scientifically and politically significant in terms of environmental impact, soil degradation and food supply. This includes challenges observed long before the war, changes currently driven by the war, and potential future outcomes if no actions are taken.

To make these points more clearly, we have modified the manuscript as follows:

- 1) Added to the Introduction:

“Ukraine has long been one of the world's leading exporters of wheat, maize and oilseed products.”

...

“Here, we trace nutrient management during Ukraine’s transition from the Soviet era (from 1980) through the challenges of independence (since 1991), culminating in the pre-war period of high agricultural productivity (2019-2021), and extending up to two years of war (2022-2023).”

- 2) Rephrased/ added some text for a sake of clarity in the Results section (see changes highlighted in this section of the manuscript).

- 3) We have strengthened the analysis of the importance of nutrient management in Ukraine for food security and the global food supply chain, including the war impact. To reflect this, we have also added the following text to a new section on **‘Implications for food security’**:

“The evolving nutrient management landscape in Ukraine holds profound implications not only for national food security but also for the stability of the global food supply chain. As one of the world’s top exporters of wheat, maize and sunflower oil, Ukraine plays a critical role in feeding over 400 million people worldwide, particularly in import-dependent regions^{20,21} (Supplementary Fig. 2-4, Text 1.1). However, the ongoing war has disrupted fertilizer supply chains and reduced agricultural inputs, aggravating imbalances of nitrogen, phosphorus and potassium (Fig 1; Supplementary Fig. 8, 26). These imbalances are marked by spatial and crop-type asymmetry in nitrogen (surpluses in some areas, and deficits in others) and persistent deficits in phosphorus and potassium (Fig. 2; Supplementary Fig. 26). The resulting degradation of Ukraine’s Chernozem soils (Supplementary Fig. 14, Text 2.2), is significant as these store about 7% of the world’s soil carbon and underpin around 90% of the country’s agricultural output^{14,22}. Without immediate intervention, this trend could trigger substantial losses of soil organic matter, especially in the regions affected by the war, risking long-term productivity declines for key crops, such as wheat and maize²³.

Given that Ukrainian grain exports traditionally support food security in regions such as North Africa and the Middle East (Supplementary Fig. 3), disruptions in crop output could exacerbate hunger and price volatility worldwide^{20,21}. The ripple effects are global: disruptions in Ukrainian grain exports during the early months of the war contributed to a sharp rise in global food prices, with the FAO Food Price Index reaching its highest level on record in 2022, over 14% higher than the previous year²⁴. This surge intensified global food insecurity, contributing to a wider hunger crisis with a global estimate of 691-783 million people affected in 2022²⁵. Moreover, the under-utilization of manure, caused by the decoupling of livestock and crop systems, represents a missed opportunity for circular nutrient economy practices (Supplementary Fig. 10-13, Text 2.1) that could reduce dependence on synthetic fertilizers while lowering reactive nitrogen and greenhouse gas emissions. Addressing these challenges requires improved access to fertilizers and long-term investments in integrated nutrient management strategies. Strengthening Ukraine’s nutrient resilience is not merely a national priority but a global imperative, when considering the global food security and climate interactions.”

[R4.3] The strong assumptions make the results questionable, the N/P deficits may lead to reduced nutrient content in food (rather than static).

Response to [R4.3]. Thank you for this important observation. Indeed, the lack of region-tailored, variety-specific NPK content data represents a significant gap and a critical parameter needed for precise agricultural balance estimations. We discuss this issue in the manuscript (Supplementary Table 4) and have listed it as a priority area for future action:

“Nationwide field campaign to inventory nutrient removal rates for the key crop varieties across georegions (Supplementary Table 4).”

Furthermore, we emphasize that nutrient imbalances have already led to soil degradation, particularly P- and K-mining since the late 1990s, and N surpluses before the war, transitioning into severe deficits across all nutrients by 2023; soil organic matter loss in Ukraine has been well documented (please see the revised Supplementary “**Text 2.2: Soil degradation and the current state of soils in Ukraine**” and the “**Text 2.3: High N-mineralization rate in black soils (Chernozems)**”).

[R4.4] Besides, the war only affect the nutrients balance in recent a few years. Reduced agricultural activities could reduce both nutrient inputs and outputs, which doesn't not nesslerily indicate surplus and environmental risks.

Response to [R4.4]. We agree that in principle the war could have reduced both inputs and outputs, so that the net soil balance might not have changed. However, the detailed data we have brought together show that since 2022 the war has resulted in a substantial nutrient deficit (rather than surplus as queried by the reviewer). Going beyond this, our study actually traces the situation for over 40 years, enabling comparison of different periods.

In the revised version, we have more clearly outlined which period were analyzed and discussed:

“Here, we trace nutrient management during Ukraine’s transition from the Soviet era (from 1980) through the challenges of independence (since 1991), culminating in the pre-war period of high agricultural productivity (2019-2021), and extending up to two years of war (2022-2023).”

Concerning harvests, the reduction of utilized agricultural areas due to the war since 2022 has indeed led to a lower total yield, contributing to food insecurity, particularly in countries dependent on Ukrainian exports.

Concerning nutrient inputs:

(i) A significant reduction in nutrient input may initially be compensated by nutrient mining from the highly fertile Chernozem soils, accelerating soil degradation - as observed between 1991-early 2000s and again in 2023.

(ii) A surplus of nutrient inputs (especially through synthetic fertilizers give dates) could lead to serious environmental risks.

We have clarified these points throughout the revised version of the manuscript, including the Supplementary Material.

[R4.5] Thus, the authors should think about in-depth research question, improve methods through sampling, monitoring or modeling.

Response to [R4.5]. In this manuscript, we aimed to emphasize fundamental questions regarding the relationship between crop productivity, the maintenance of Chernozem soil fertility, wasted manure/livestock-cropland decoupling and environmental impacts - not only in the context of sustainable and economic development for Ukraine but mainly in terms of broader implications for global food security in the long-term.

Our analysis focuses on the three key export crops (wheat, maize and sunflower), which collectively pose significant environmental, economic and food security challenges.

We have strengthened the analysis of the importance of nutrient management in Ukraine for food security and the global food supply chain, including the war impact; see the new section on ‘**Implications for food security**’ cited in the [comment R4.2]

We propose urgent interventions, including an easy-to-use tool for farmers (Smart Fertilizer Planner) to optimize nutrient management, supported by a package of priority actions. See the revised section “**Text 2.8: Smart Fertilizer Planner**” of the Supplementary as well as added information on other tools available: *“Principle and usage. The proposed region-tailored, variety-specific **Smart Fertilizer Planner (SFP)** tool is based on a simplified nutrient balance principle, incorporating a ‘5%-surplus’ for N and ‘zero-surplus’ approach for P and K. This balance is calculated between the total input, which includes synthetic and organic fertilizer applied by farmers alongside the sum of mean atmospheric inorganic N deposition and N fixation by free-living organisms (provided as a ‘constant’; N deposition is updated upon new data become available in EMEP reports), and the total output, which includes crop yield (and by-product if straw is removed from the field). Both inputs and outputs are expressed in pure elemental units: kg N ha⁻¹, kg P ha⁻¹, and kg K ha⁻¹.*

It is assumed that a 5% N surplus, together with variability in atmospheric nutrient deposition, including unaccounted organic NPK inputs, is roughly offset by unavoidable nutrient losses, which are higher for N and expected to be marginal for P and K (see 'Limitations and Assumptions' below).

The tool is suggested to be available as mobile app and a downloadable spreadsheet. It would provide farmers with easy-to-use yet precise estimation for NPK fertilizer input requirements tailored to their region and crop varieties, based on data from the top-performing crops over the past 3-5 seasons.

The SFP would help calculate annual field-scale nutrient balances and estimates nutrient use efficiency after harvest using yield data. A key element of the SFP is the collection of crop yields from varieties grown in a specific region, followed by analysis of their NPK content to develop a region-tailored, variety-specific NPK removal repository. This repository should be updated whenever new varieties are introduced in the region. Another important aspect is the analysis of NPK content in organic fertilizers applied (if relevant), to at least establish an initial inventory for the region or country. Ideally, this information should be updated regularly, or, as an advanced option, it could be provided on a regular basis by organic fertilizer suppliers. This approach can be implemented at field-to-regional levels, supporting farmers and policymakers in various counties and georegions for rain-fed cropping systems (i.e., including internationally beyond Ukraine). It can also be used for irrigated crops; in this case, the NPK content applied through irrigation water (i.e., NPK concentration in irrigated water multiplied by the amount of water applied) should be added to the input category for more precise calculations. Moreover, the SFP would be adaptable to entire crop rotation cycles, making fertilizer planning more accurate and comprehensive. In this case, it would account for nutrient surpluses or deficits from preceding crops to adjust applications for subsequent ones.

Limitations and assumptions. *The proposed tool would generally assume that vegetation biomass benefits from crop residues returned to the field in previous years. In a simplest approach it may be estimated that up to 5%-surpass for N together with variation of atmospheric deposition of NPK, including unaccounted rates for organic NPK deposited (**Supplementary Text 3.1**), are roughly offset by unavoidable N emissions and potential NPK leaching or runoff (so long as urea is not used). However, the performance of the SFP may be substantially improved if evidence-based estimates for organic nutrient deposition and region-specific fertilizer-induced emission factors are available. Emissions of N are considered the primary loss pathway in black soil agroecosystems, while leaching of nitrate, phosphorus and potassium compounds is supposed to be less critical, though migration to lower soil horizons may occur (Kovda and Rozanov, 1988; Krupenikov et al., 2011; Medinets et al., 2016, 2021; Boincean and Dent, 2019).*

Prerequisites and requirements. *To estimate NPK removal by crop yield for specific crop varieties, a database of NPK content for currently used varieties is essential. This database should be created and*

regularly updated as new varieties are introduced. Such information must be accessible to farmers, either through regional advisory services, online repository or mobile app.

The minimum requirements for successful use of the SFP tool would include:

- (i) Replacing straight urea to prevent significant nitrogen losses via NH_3 volatilization.
- (ii) Returning crop residues to the field as a source for the next year's vegetation biomass or, alternatively, accurately calculating removed amounts and replenishing them with (organic) fertilizers.
- (iii) Ensuring availability of NPK content data for key crop varieties.
- (iv) Accessing appropriate soil testing to monitor exchangeable P and K concentrations.

Further development. The SFP is intended as a simple flexible solution that should be developed over time. With further research and advancements in technology, more precise, region-specific guidance could be developed, taking into account local pedoclimatic and socioeconomic conditions.

Other tools available. Also, there are some other free and commercial tools available to farmers for various purposes, they are, sometimes, specific to the georegion, but often not specific to crop varieties in a given region and, unfortunately, do not cover Eastern European countries outside the EU. However, with region-tailored, variety-specific evidence-based data, they could be complimented to SFP or adapted as a separate tool for use in Eastern European countries.

Among them are the following:

- (i) Farm Sustainability Tool for Nutrients (FaST) – a digital tool developed under the EU CAP to help farmers create nutrient management plans. It integrates existing (mainly satellite) data with manual inputs to provide customized recommendations on crop fertilization, aiming to deliver both economic and environmental benefits. It is intended for region-specific implementation across EU countries with governmental support.
- (ii) NUTRI-CHECK NET Platform – an EU-funded initiative offering a three-step approach—Plan, Check & Adjust, and Review—for nutrient management. It essentially serves as an inventory of various tools and services to support farmers, agronomists, researchers, and the broader agricultural industry in sharing knowledge and exploring available N, P, and K nutrient management approaches for wheat, maize, and potato across Europe.
- (iii) Planning Land Applications of Nutrients for Efficiency and the Environment (PLANET) – a nutrient management decision support tool for use by farmers and advisers in England, Wales and Scotland. It supports field-level nutrient planning and helps assess and demonstrate compliance with the Nitrate Vulnerable Zone (NVZ) rules.
- (iv) CAFRE Crop Nutrient Calculator – a tool developed by the College of Agriculture, Food and Rural Enterprise (CAFRE) of Northern Ireland, based on soil nutrient analysis. It helps farmers create precise fertilization plans that meet crop nutrient requirements without excess. The tool emphasizes the importance of soil pH and nutrient stocks, and promotes low-emission techniques for slurry spreading.”

Please also see our comments to Reviewer 1 [comment R1.3], where we have noted that, while actions should start now, they represent a long-term commitment by the government, jointly developed by scientists and policy-makers and co-designed with farmers.

Additionally, we highlight the critical need for strengthened international support to address these issues effectively:

“The actions listed are not only a priority for the Ukrainian economy, but are needed to maintain continued exports internationally, global food security and environment sustainability contributing to SDG Goal 2: Zero Hunger by 2030 and supporting the transition to a circular and net-zero economy by combating nutrient pollution and reducing greenhouse gas emissions. Addressing this wider challenge is a key opportunity for United Nations affiliated bodies such as the International Nitrogen Management System¹¹, Nitrogen Working Group and the Global Partnership on Nutrient Management, where a stronger coordination of effort is needed that links management of nitrogen, phosphorus, potassium and other nutrients. International action would help mobilize cross-sectoral intergovernmental policy on nutrients, accelerating the transition to a circular economy. If progress is to be made towards Target 7 of the Global

Biodiversity Framework, to at least halve nutrient pollution by 2030, such international support to Ukraine could become a beacon to guide necessary actions globally.”

Detailed response from the authors to comments of peer reviewers

We thank the editor and all reviewers for their constructive comments on our submitted manuscript, which have enabled us to strengthen the submission. In addition, we are grateful for the reviewer 2 and 3 support and are glad that the previously revised version satisfactorily meets their expectations. We hope that the editor and reviewers will agree that we have addressed all the extra points raised by the reviewer 1.

For convenience in cross-referencing our point-by-point response, we have numbered the comments of the reviewers as follows: Reviewer 1: [R1.1...R1.n]; Reviewer 2: [R2.1...R2.n]; etc. Our replies are listed in the blue highlighted text starting with ‘Response to [...]’, while cited/ revised text is additionally shown *in italics*.

REVIEWER COMMENTS:

Reviewer #1 (Remarks to the Author):

[R1.1] The author has undertaken comprehensive and thoughtful revisions to this manuscript, significantly improving its quality. I agree that this paper is more appropriately positioned as a Short Communication rather than a full-length research article. Its primary aim should be to inform both the scientific community and policymakers about the nutrient imbalances resulting from Ukraine’s unique circumstances and the broader environmental consequences. It should also serve as a call to action for the international community to address the associated food security risks and their implications for global food supply chains.

Response to [R1.1]. Many thanks for your positive evaluation of our work.

However, further refinements are necessary before it can meet the standards for publication.

[R1.2] (1) The paper’s novelty remains insufficiently articulated. The stated focus on the “phased characteristics of nitrogen, phosphorus, and potassium in Ukraine” does not, in itself, constitute a novel contribution—especially given that Figures 1 and 2 do not convincingly support this claim.

While the proposed mitigation strategies are practical, they do not represent a scientific innovation. These recommendations lack empirical testing, making their feasibility and effectiveness uncertain. Without case studies or scenario analyses, the suggestions remain speculative rather than evidence-based. The author should clarify more explicitly how this study advances academic understanding or contributes to global sustainability goals.

Response to [R1.2]. Thank you very much for your comment. We have revised the **Abstract, Introduction and Current challenges and urgent actions** sections, and added a new section on **Nutrient management scenarios** alongside a Figure, Supplementary Figures, Tables and Text, to more clearly articulate the novelty of our work. This addition complements our previously stated contribution as the first study to chart the course of asymmetries between N, P and K nutrients, including their spatial patterns across Ukraine, over four decades, alongside analysis of trends, impacts and risks for Ukraine’s agricultural sector and global food security.

Specifically, we now emphasize that this is the first county-to-national-scale assessment of how war has disrupted nutrient cycles in one of the major breadbasket countries, Ukraine, with implications for global food security. We assess five forward-looking nutrient management scenarios to inform policy for a sustainable recovery by 2030, aligned with SDG 2 and Target 7 of the Global Biodiversity Framework to halve global nutrient pollution. Notably, we integrate war-impact data (through to 2023) into an extended war disruption scenario (**S-w**) to highlight the potentially severe consequences for soil health and global food security, while also underscoring the unsustainable nature of the highly productive 2021 practices considering a peacetime business-as-usual (BaU, **S-0**) scenario.

We propose and evaluate plausible sustainable measures across three scenarios:

- **S-1:** Manure-enriched precision fertilization, replacing synthetic fertilizer with manure-N at an increased 30% share;
- **S-2:** Enhanced-efficiency fertilizers (EEFs), building upon S-1;
- **S-3:** Legume-based diversification, in combination with S-2 or S-1.

These measures are collectively intended for inclusion in an Integrated Nutrient Management Plan for Ukraine as part of the agricultural sector's recovery under the Ukraine Recovery and Reconstruction (URR) budget. This transforms the recommendations from being speculative to become evidence-based and quantitatively supported. We believe these revisions clearly demonstrate the timeliness of our study and how it advances understanding while contributing to global sustainability discussions (see revised text below).

Abstract

L21-24: *Based on analysis of five scenarios for 2030, we show how an Integrated Nutrient Management Plan for Ukraine combining manure recycling, precision fertilization and legume expansion, could maintain crop productivity, significantly reduce nutrient surpluses and improve nutrient use efficiencies to 80–89%, substantially curtailing environmental pollution and soil degradation.*

Introduction

L49-54: *Using Ukraine as a case study, we analyse the impacts of nutrient imbalances, including the first county-to-national scale assessment of how war has disrupted nutrient cycles in a major breadbasket country, with implications for global food security. We also assess five forward-looking nutrient management scenarios to inform policy for sustainable recovery by 2030, while also warning of the potential consequences if urgent action is not taken. The analysis focuses on the three staple crops (wheat, maize and sunflower) collectively covering more than 67% of Ukraine's total utilized agro-area in 2021¹⁴.*

Nutrient management scenarios

L355-405: *We developed forward-looking nutrient management scenarios to project how Ukraine's nutrient balances for major crops (wheat, maize, sunflower) might evolve by 2030 (Fig. 3; Supplementary Fig. 27, 28, Table 2-4). These scenarios highlight both challenges and opportunities for Ukraine to achieve agricultural sustainability (see details in Supplementary Text 2.8). The scenarios include:*

*S-0: **Business-as-usual (BaU)**, maintaining 2021 practices prior to the invasion of Ukraine;*

*S-w: **Extended war disruption scenario**, assuming prolonged fertilizer shortages at 2023 levels;*

*S-1: **Manure-enriched precision fertilization**, replacing synthetic N with manure-N of increased 30% share;*

*S-2: **Enhanced-efficiency fertilizers (EEFs)**, building upon S-1;*

*S-3: **Legume-based diversification**, in combination with S-2 or S-1.*

*In the **BaU scenario (S-0)**, significant nutrient imbalances persist with annual N surpluses (~42 kg N ha⁻¹ yr⁻¹ for wheat and maize), causing cumulative losses of ~4 million tonnes (Mt) N total over 2024-2030 across these crops growing areas in Ukraine. Simultaneously, P and K deficits (6–10 kg ha⁻¹ yr⁻¹) lead to cumulative soil mining of ~0.1 Mt of **extended war disruption scenario (S-w)** severely escalates nutrient depletion, projecting cumulative deficits of 1.7 Mt N, 2.8 Mt P and 2.1 Mt K for the three crop growing areas over 2024-2030 (Supplementary Table 2-4, Text 2.8). Such extreme nutrient mining risks irreversible damage to Ukraine's Chernozem soils, depleting soil organic matter and lowering productivity, echoing degradation patterns of the 1990s^{15,26,27}. This scenario emphasizes the critical need for immediate interventions to sustain national and global food security.*

*By contrast, the three **sustainable nutrient management scenarios** demonstrate achievable improvements (Fig. 3; Supplementary Fig. 27, 28). **Scenario S-1** involves substituting part of synthetic N fertilizer with manure-N, reaching a 30% share, combined with precision fertilization*

strategies (e.g., guided by a Smart Fertilizer Planner; Supplementary Text 2.8), reducing total N inputs (through synthetic N) by 10% for maize and wheat, and 5% for sunflower. By 2030, S-1 projects a 37% synthetic N reduction, a 28% increase in manure-N, halving N surpluses and approaching balanced P and K applications. This scenario envisages a significant reduction of N losses: ammonia (NH_3) reduce by ~49%, nitrous oxide (N_2O) by ~13% and N runoff by 50–67%, with potential yield increases of ~5%, compared to BaU (S-0). Nutrient use efficiencies would increase substantially, reaching 76–84% for N and 79–87% for P and K, alongside improved soil organic matter and resilience (Supplementary Table 2-4, Text 2.8).

Scenario S-2 builds on S-1 by integrating EEFs, allowing an additional 10% reduction in total N (through synthetic N) without yield loss. By 2030, N surpluses would decrease further to 9–17 kg N ha⁻¹ yr⁻¹, elevating NUE to 85–88%, with cumulative emissions reduced significantly: NH_3 (~77%), N_2O (~48%) and nitric oxide (NO, ~60%) compared to BaU (Supplementary Table 2-4, Text 2.8). This substantially enhances air quality and supports climate mitigation.

Scenario S-3, proposes expanding legumes to 20% of grain land, leveraging biological N fixation (130–150 kg N ha⁻¹ yr⁻¹, on average²⁸). This strategy could deliver an estimated 40–50 kg N ha⁻¹ benefit to subsequent cereals, reducing total N inputs (through synthetic N) by an additional ~15%. When combined with S-1 or S-2, this results in a 52–61% reduction in synthetic N input relative to BaU by 2030, with an expectation of maintaining the same level of yields as BAU but with higher nutrient use efficiency (Supplementary Text 2.8, Table 2).

Collectively, these scenarios show how Ukraine's nutrient asymmetry can be addressed. By adopting sustainable nutrient management practices (S-1, S-2, S-3), Ukraine could maintain crop productivity, significantly reduce nutrient surpluses and improve nutrient use efficiencies to 80–89%, substantially curtailing environmental pollution and soil degradation (Fig. 3; Supplementary Fig. 27, 28, Table 2-4). The urgency of transitioning from fertilizer dependence, intensified by war-induced disruptions, underscores the need for immediate policy actions and investment toward sustainable agricultural recovery (see below).

L847-862:

Fig. 3 | Average nitrogen (N) surplus or deficit as a percentage of total N input for wheat, maize and sunflower in Ukraine under five contrasting scenarios for 2030. The business-as-usual scenario (BaU; S-0) reflects the continuation of 2021 agricultural practices prior to the Russian invasion. The extended war disruption scenario (War; S-w) assumes prolonged fertilizer shortages at 2023 levels. The manure-enriched precision fertilizer scenario (S1; S-1) involves the substitution of synthetic N with manure-N, increasing its share by 30%, combined with precision fertilizer application. The enhanced efficiency fertilizer scenario (S2; S-2) builds upon S1 by incorporating nitrification inhibitors and slow-release fertilizers. The legume-based diversification scenario (S3; S-3) introduces optimized crop rotations with legumes, in combination with S2 or S1 (see Nutrient Management Scenarios). Scenario colours indicate relative N emission reductions: red (BaU) denotes no reduction, a green gradient from S1 to S3 reflects stepwise reductions and grey indicates the situation under the War (see Supplementary Table 2, Text 2.8). The green-shaded belt represents the acceptable N surplus range, between 10% and 20% of total N input³⁶. The green-dotted belt indicates further reductions from 10% toward the minimally unavoidable N loss, expected to decline over time with improved agricultural practices. Arrows indicate the direction of surplus reduction. The dotted horizontal line marks zero N balance (equilibrium) with surpluses above and deficits below this threshold.

Supplementary information

Supplementary Figure 27. Average phosphorus (P) surplus or deficit as a percentage of total P input for wheat, maize and sunflower in Ukraine under five contrasting scenarios for 2030. The business-as-usual scenario (BaU; S-0) reflects the continuation of 2021 agricultural practices prior to the Russian invasion. The extended war disruption scenario (War; S-w) assumes prolonged fertilizer shortages at 2023 levels. The manure-enriched precision fertilizer scenario (S1; S-1) involves the substitution of synthetic N with manure-N, increasing its share by 30%, combined with precision fertilizer application. The enhanced efficiency fertilizer scenario (S2; S-2) builds upon S1 by incorporating nitrification inhibitors and slow-release fertilizers. The legume-based diversification scenario (S3; S-3) introduces optimized crop rotations with legumes, in combination with S2 or S1 (see Nutrient Management Scenarios). Scenario colours indicate a surplus (white) or a deficit (grey). The green-shaded belt represents the acceptable P surplus range, between 0% and 20% of total P input (as a portion of manure-P remains crop-unavailable during the first year of application; see Supplementary Table 3, Text 2.8). Arrows indicate the direction of surplus reduction. The dotted horizontal line marks zero P balance (equilibrium) with surpluses above and deficits below this threshold.

Supplementary Figure 28. Average potassium (K) surplus or deficit as a percentage of total K input for wheat, maize and sunflower in Ukraine under five contrasting scenarios for 2030. The business-as-usual scenario (BaU; S-0) reflects the continuation of 2021 agricultural practices prior to the Russian invasion. The extended war disruption scenario (War; S-w) assumes prolonged fertilizer shortages at 2023 levels. The manure-enriched precision fertilizer scenario (S1; S-1) involves the substitution of synthetic N with manure-N, increasing its share by 30%, combined with precision fertilizer application. The enhanced efficiency fertilizer scenario (S2; S-2) builds upon S1 by incorporating nitrification inhibitors and slow-release fertilizers. The legume-based diversification scenario (S3; S-3) introduces optimized crop rotations with legumes, in combination with S2 or S1 (see Nutrient Management Scenarios). Scenario colours indicate a surplus (white) or a deficit (grey). The green-shaded belt represents the acceptable K surplus range, between 0% and 20% of total K input (as a portion of manure-K remains crop-unavailable during the first year of application; see Supplementary Table 4, Text 2.8). Arrows indicate the direction of surplus reduction. The dotted horizontal line marks zero K balance (equilibrium) with surpluses above and deficits below this threshold.

Supplementary Table 2. Summary of nitrogen (N) inputs, N outputs, N balances and N use efficiencies (NUE) for wheat, maize and sunflower under five nutrient management projections by 2030 alongside cumulative N balances over 2024–2030 for business-as-usual (BaU, S-0) and extended war disruption (S-w) scenarios, as well as potential emission reductions for manure-enriched precision fertilization (S-1), enhanced-efficiency fertilizers (S-2) and legume-based diversification (S-3) scenarios compared to the BaU (S-0) scenario. ‘Total N input’ includes synthetic N fertilizer, organic (manure) N fertilizer, atmospheric N deposition and N fixation (see Methods). ‘Org.’ represents manure-N; ‘Synth.’ represents synthetic N fertilizer; BNF (legumes) denotes the average annual legacy input from biological N fixation by legumes in a four-year crop rotation, where legumes are grown in one of the years preceding cereals, with sunflower often serving as the final crop in the rotation; ‘Crop available’ stands for N available for crop uptake within year of application (and previous-year legacy from manure-N; see Supplementary Text 2.8 for details). ‘Mt N’ represents million tonnes of N. Scenario descriptions in Supplementary Text 2.8.

Crop	Scenario	N input, kg N ha ⁻¹ yr ⁻¹						N output, kg N ha ⁻¹ yr ⁻¹	N balance, kg N ha ⁻¹ yr ⁻¹		N surplus/deficit, %		NUE, %		Cumulative balance by 2030, Mt N	Emission reduction, %		
		Total	Total org.	Crop-available					Total	Crop-available	Total	Crop-available	Total	Crop-available		NH ₃	N ₂ O	NO
				Org.	Synth.	BNF (legumes)	Total											
Wheat	S-0	136.7	2.2	0.9	122.3	-	135.4	93.9	42.8	41.5	31.3	30.6	68.7	69.4	2.1	0	0	0
	S-1	123.0	33.0	19.8	77.0	-	109.8	93.9	29.1	15.9	23.7	14.5	76.3	85.5	-	49	13	n/a
	S-2	110.7	33.0	19.8	64.7	-	97.5	93.9	16.8	3.6	15.2	3.7	84.8	96.3	-	77	48	60
	S-3*	109.0	33.0	19.8	48.1	~15	95.9	93.9	15.2	2.0	13.9	2.5	86.1	97.9	-	≥77	≥48	≥60
	S-w	66.9	1.5	0.9	55.3	-	66.3	93.9	-27.0	-27.6	-40.4	-41.7	140	142	-0.9	-	-	-
Maize	S-0	136.4	5.3	3.2	118.9	-	134.3	94.5	41.9	39.8	30.7	29.6	69.3	70.4	2.0	0	0	0
	S-1	122.7	32.1	19.3	74.9	-	109.9	94.5	28.2	15.4	23.0	14.0	77.0	86.0	-	49	13	n/a
	S-2	110.4	32.1	19.3	62.6	-	97.6	94.5	15.9	3.1	14.4	3.1	85.6	96.9	-	77	48	60
	S-3*	108.8	32.1	19.3	46.1	~15	96.0	94.5	14.3	1.5	13.2	1.9	86.8	98.4	-	≥77	≥48	≥60
	S-w	90.2	3.0	1.8	75.4	-	89.0	96.0	-5.8	-7.0	-6.4	-7.9	106	108	-0.2	-	-	-
Sunflower	S-0	75.4	2.7	1.6	60.5	-	74.3	63.0	12.4	11.3	16.4	15.2	83.6	84.8	0.5	0	0	0
	S-1	75.4	16.3	9.8	38.1	-	68.9	63.0	12.4	5.9	16.4	8.5	83.6	91.5	-	48	13	n/a
	S-2	71.6	16.3	9.8	34.3	-	65.1	63.0	8.6	2.1	12.0	3.2	88.0	96.8	-	77	48	60
	S-3*	70.9	16.3	9.8	23.6	~10	64.4	63.0	7.9	1.4	11.1	2.5	88.9	97.9	-	≥77	≥48	≥60
	S-w	45.8	1.5	0.9	32.6	-	45.2	63.0	-17.2	-17.8	-37.6	-39.3	138	139	-0.6	-	-	-

*S-3 is a qualitative scenario with indicative values, building on S-2. It assumes multilateral benefits for subsequent crops and soil health within the crop rotation, although these benefits are not quantified.

Supplementary Table 3. Summary of phosphorus (P) inputs, P outputs, P balances and P use efficiencies (PUE) for wheat, maize and sunflower under five nutrient management projections by 2030 alongside cumulative P balances over 2024–2030 for business-as-usual (BaU, S-0) and extended war disruption (S-w) scenarios. S-1 stands for the manure-enriched precision fertilization scenario; S-2 stands for the enhanced-efficiency fertilizers scenario; S-3 stands for the legume-based diversification scenario. ‘Total P input’ includes synthetic P fertilizer and organic (manure) P fertilizer (see Methods). ‘Org.’ represents manure-P; ‘Synth.’ represents synthetic P fertilizer; ‘Crop available’ stands for P available for crop uptake within year of application (and previous-year legacy from manure-P; see Supplementary Text 2.8 for details). ‘Mt P’ represents million tonnes of P. Scenario descriptions in Supplementary Text 2.8.

Crop	Scenario	P input, kg P ha ⁻¹ yr ⁻¹				P output, kg P ha ⁻¹ yr ⁻¹	P balance, kg P ha ⁻¹ yr ⁻¹		P surplus/deficit, %		PUE, %		Cumulative balance by 2030, Mt P	
		Total	Total org.	Crop-available			Total	Crop- available	Total	Crop- available	Total	Crop- available		
				Org.	Synth.	Total								
Wheat	S-0	11.2	0.7	0.48	10.5	11.01	17.23	-6.0	-6.2	-53.6	-56.5	154	157	-0.04
	S-1	20.9	10.3	7.23	10.5	17.76	17.23	3.6	0.5	17.4	3.0	82.6	97.0	-
	S-2	20.9	10.3	7.23	10.5	17.76	17.23	3.6	0.5	17.4	3.0	82.6	97.0	-
	S-3*	20.9	10.3	7.23	10.5	17.76	17.23	3.6	0.5	17.4	3.0	82.6	97.0	-
	S-w	4.1	0.5	0.32	3.64	3.96	17.23	-13.1	-13.3	-320	-335	420	435	-1.5
Maize	S-0	12.5	1.6	1.15	10.9	12.05	22.28	-9.8	-10.2	-78.2	-84.9	178	185	-0.07
	S-1	25.3	10.0	7.02	15.3	22.28	22.28	3.0	0	11.9	0	88.1	100	-
	S-2	25.3	10.0	7.02	15.3	22.28	22.28	3.0	0	11.9	0	88.1	100	-
	S-3*	25.3	10.0	7.02	15.3	22.28	22.28	3.0	0	11.9	0	88.1	100	-
	S-w	7.5	0.9	0.63	6.5	7.13	22.28	-14.8	-15.2	-197	-213	297	313	-0.8
Sunflower	S-0	13.1	0.8	0.59	12.3	12.89	10.10	3.0	2.8	22.9	21.6	77.1	78.4	0.02
	S-1	11.6	5.1	3.57	6.5	10.10	10.10	1.5	0	13.2	0	86.8	100	-
	S-2	11.6	5.1	3.57	6.5	10.10	10.10	1.5	0	13.2	0	86.8	100	-
	S-3*	11.6	5.1	3.57	6.5	10.10	10.10	1.5	0	13.2	0	86.8	100	-
	S-w	5.0	0.4	0.31	4.6	4.91	10.10	-5.1	-5.2	-102	-106	202	206	-0.5

*S-3 is a qualitative scenario with indicative values, building on S-2. It assumes multilateral benefits for subsequent crops and soil health within the crop rotation, although these benefits are not quantified.

Supplementary Table 4. Summary of potassium (K) inputs, K outputs, K balances and K use efficiencies (KUE) for wheat, maize and sunflower under five nutrient management projections by 2030 alongside cumulative K balances over 2024–2030 for business-as-usual (BaU, S-0) and extended war disruption (S-w) scenarios. S-1 stands for the manure-enriched precision fertilization scenario; S-2 stands for the enhanced-efficiency fertilizers scenario; S-3 stands for the legume-based diversification scenario. ‘Total K input’ includes synthetic K fertilizer and organic (manure) K fertilizer (see Methods). ‘Org.’ represents manure-K; ‘Synth.’ represents synthetic K fertilizer; ‘Crop available’ stands for K available for crop uptake within year of application (and previous-year legacy from manure-K; see Supplementary Text 2.8 for details). ‘Mt K’ represents million tonnes of K. Scenario descriptions in Supplementary Text 2.8.

Crop	Scenario	K input, kg K ha ⁻¹ yr ⁻¹					K output, kg K ha ⁻¹ yr ⁻¹	K balance, kg K ha ⁻¹ yr ⁻¹		K surplus/deficit, %		KUE, %		Cumulative balance by 2030, Mt K
		Total	Total org.	Crop-available				Total	Crop- available	Total	Crop- available	Total	Crop- available	
				Org.	Synth.	Total								
Wheat	S-0	15.2	1.8	1.4	13.5	14.9	20.9	-5.6	-6.0	-36.9	-40.1	137	140	-0.04
	S-1	26.4	26.4	21.1	0.0	21.1	20.9	5.5	0.2	20.9	1.1	79.1	98.9	-
	S-2	26.4	26.4	21.1	0.0	21.1	20.9	5.5	0.2	20.9	1.1	79.1	98.9	-
	S-3*	26.4	26.4	21.1	0.0	21.1	20.9	5.5	0.2	20.9	1.1	79.1	98.9	-
	S-w	6.6	1.3	1.0	5.3	6.3	20.9	-14.3	-14.6	-219	-231	319	331	-1.0
Maize	S-0	22.4	2.4	1.9	18.2	20.1	30.0	-7.6	-9.8	-33.8	-48.9	134	149	-0.05
	S-1	35.1	25.6	20.5	9.5	30.0	30.0	5.1	0.0	14.6	0	85.4	100	-
	S-2	35.1	25.6	20.5	9.5	30.0	30.0	5.1	0.0	14.6	0	85.4	100	-
	S-3*	35.1	25.6	20.5	9.5	30.0	30.0	5.1	0.0	14.6	0	85.4	100	-
	S-w	14.2	2.5	2.0	11.7	13.7	30.4	-16.2	-16.7	-114	-122	214	222	-0.5
Sunflower	S-0	18.7	2.2	1.7	16.5	18.2	17.6	1.1	0.6	5.8	3.2	94.2	96.8	0.01
	S-1	20.2	13.0	10.4	7.2	17.6	17.6	2.6	0.0	12.9	0.0	87.1	100	-
	S-2	20.2	13.0	10.4	7.2	17.6	17.6	2.6	0.0	12.9	0.0	87.1	100	-
	S-3*	20.2	13.0	10.4	7.2	17.6	17.6	2.6	0.0	12.9	0.0	87.1	100	-
	S-w	8.4	1.2	1.0	7.2	8.2	17.6	-9.2	-9.5	-110	-116	210	216	-0.6

*S-3 is a qualitative scenario with indicative values, building on S-2. It assumes multilateral benefits for subsequent crops and soil health within the crop rotation, although these benefits are not quantified.

Supplementary Text 2.8: Detailed description of nutrient management scenarios

L339-L633: Scenario design and baseline calibration

We defined a set of forward-looking scenarios to explore how Ukraine's crop nutrient balances could evolve by 2030 under different management interventions (Supplementary Table 2-4). These include;

- S-0: A Business-as-usual (BaU) scenario (assuming a return to 2021 practices);
- S-w: An extended war disruption scenario (assuming prolonged fertilizer shortages at 2023 levels);
- S-1: An improved management scenario at 2021 yield levels (manure-enriched precision fertilization);
- S2: An improved management scenario, building on S-1, plus enhanced-efficiency fertilizers);
- S3: An improved management scenario involving the legume-based diversification, combined with S-2.

The baseline for all comparisons is 2021, the last pre-war year of normal production, which we use as an analog for 'current' conventional BaU management. National nutrient balance data from 2021 for wheat, maize and sunflower, calculated using advanced national statistics (see Methods), were sourced from our dataset (Fig. 1) and cross-checked with literature.

For assessment of the scenarios, we compare them with targets for acceptability outlined by Medinets and Sutton (2025), which can be summarized as follows: (i) unavoidable N losses are currently estimated at about 10% of total N input, considered the minimum level that avoids the risk of soil N mining under conventional agricultural practices (EUNEP, 2016; Congreves et al., 2021; Mi et al., 2025); (ii) the upper acceptable threshold is suggested to be no more than twice the unavoidable N losses, aligning with the societal optimum N input framework (van Grinsven et al., 2014). However, the implementation of existing innovations and the development of future ones should not treat unavoidable losses as inevitable, but rather aim to reduce them further, e.g., through increased focus on precision N management and soil health.

S-0: Business-as-Usual (BaU) scenario

In 2021, Ukraine's average fertilizer application rates and yields resulted in moderate N surpluses for wheat and maize (and a slight surplus for sunflower), alongside P and K deficits for those crops (and only minor P and K surpluses for sunflower). For instance, on average, wheat received $136.7 \text{ kg N ha}^{-1} \text{ yr}^{-1}$ (of which $\sim 122.3 \text{ kg N}$ came from synthetic fertilizers and $\sim 2.2 \text{ kg N}$ from manure) and yielded $\sim 4.5 \text{ t ha}^{-1} \text{ yr}^{-1}$ grain, removing $\sim 93.9 \text{ kg N ha}^{-1} \text{ yr}^{-1}$ in the harvest, leaving an estimated N surplus of $\sim 42.8 \text{ kg ha}^{-1} \text{ yr}^{-1}$. Maize showed a similar N surplus ($\sim 41.9 \text{ kg ha}^{-1} \text{ yr}^{-1}$). By contrast, P and K inputs in 2021 were substantially less than crop offtake for the cereals, e.g., wheat received $11.2 \text{ kg P ha}^{-1} \text{ yr}^{-1}$ and $15.2 \text{ kg K ha}^{-1} \text{ yr}^{-1}$, but removed 17.2 kg P and $20.9 \text{ kg K ha}^{-1} \text{ yr}^{-1}$, respectively, forming a deficit of around $6 \text{ kg ha}^{-1} \text{ yr}^{-1}$ for both nutrients. Even larger deficits were observed for maize: $9.8 \text{ kg P ha}^{-1} \text{ yr}^{-1}$ and $7.6 \text{ kg K ha}^{-1} \text{ yr}^{-1}$. These imbalances align with previous studies (FAO, 2005; Lerman, 2008; von Lampe et al., 2014; Ludemann et al., 2024), which noted Ukraine's tendency toward sufficient or excessive N application, alongside under-application of P and K, which is likely a legacy of cheap domestic N fertilizers and costly imported P and K (Fig.3; Supplementary Table 2-4, Fig. 27, 28).

Surprisingly, sunflower systems were nearly balanced by 2021, with acceptable annual nutrient surpluses ($12.4 \text{ kg N ha}^{-1}$, 3.0 kg P ha^{-1} , and 1.1 kg K ha^{-1}), while maintaining nutrient use efficiencies within desirable ranges (84% for NUE, 77% for PUE and 94% for KUE).

We use the 2021 balance as the starting point for S-0 and as the reference for improvements in S-1, S-2 and qualitative S-3. For context, a 'no improvement' BaU trajectory (S-0) implies continued soil nutrient mining (annual negative P and K balances) and ongoing N losses, a situation recognized as unsustainable (Lerman et al., 2008; Boincean and Dent, 2019; Ludemann et al., 2024).

S-w: Extended war disruption scenario

The war disruption scenario (S-w) considers the case if 2023's war-induced low fertilizer use were prolonged, with continued extreme nutrient deficits. Under 2023 conditions, wheat, maize and sunflower had negative annual N balances (around 27.0 , 5.8 and $17.2 \text{ kg N ha}^{-1}$, respectively), annual P deficits (around 13.1 , 14.8 and 5.1 kg P ha^{-1}) and K deficits (roughly 14.3 , 16.2 and

9.2 kg K ha⁻¹, respectively). Projected over 2024–2030, such deficits would accumulate to roughly 40–189 kg N ha⁻¹, 35–103 kg P ha⁻¹, and 65–113 kg K ha⁻¹ being mined from soils per hectare, eventually impairing yields (Fig.3; Supplementary Table 2-4, Fig. 27, 28). Indeed, historical observations in Ukraine show that continuous nutrient deficits have led to steady declines in soil organic matter and incipient yield reductions (Chesnyak et al., 1983; Nosko et al., 1988, 1992; Lerman et al., 2008; Baliuk et al., 2009, 2021; Baliuk and Medvedev, 2012). This worst-case scenario provided the motivation for exploring the intervention scenarios described below.

S-1: Manure-enriched precision fertilization scenario

Scenario S-1 emphasizes increased manure recycling to substitute a portion of synthetic fertilizer, implemented through site-specific, crop-tailored fertilization practices. We operationalized S-1 based on the ‘Smart Fertilizer Planner’ (SFP) concept (Supplementary Text 2.8), which targets a 10-20% N surplus, with lower surpluses under low-emission agricultural practices (Medinets and Sutton, 2025), and near-zero surpluses for P and K. Literature on precision nutrient management suggests that better matching of nitrogen supply to crop demand can typically reduce N requirements by ~10% without yield loss (Sapkota et al., 2019; Brownlie et al., 2024).

Accordingly, as a starting point, we reduced the 2021 total N input rates for wheat and maize by 10%, solely through a decrease in synthetic N fertilizer use. For sunflower, total N input remains at the BaU level. We then increased the share of manure-N to 30% of total fertilizer N input by substituting part of the synthetic fertilizer, resulting in a total 37% reduction in synthetic N fertilizer use in each of the studied cropping systems. We also assume improved spatial targeting of N within fields and across regions, alongside low-emission technologies, enabling modest reductions in total N use while maintaining yields (Sutton et al., 2022; Brownlie et al., 2024), although not quantitatively modelled in the scenario.

For wheat and maize, which faced nutrient deficits in P and K, we implemented crop-specific strategies to meet the SFP targets. The remaining P shortfall, after accounting for manure, is addressed with synthetic P fertilizer: maintained at BaU levels for wheat and increased by 40% for maize to close the deficit. For K, we assume complete discontinuation of synthetic K use for wheat and a 52% reduction for maize. For sunflower, which showed only minor P and K surpluses in 2021, increased manure recycling allows for substantial reductions in synthetic P and K use, by 48% and 56%, respectively (Fig.3; Supplementary Table 2-4, Fig. 27, 28).

However, substituting synthetic fertilizers with organic (manure-based) sources requires accurate estimation of plant-available nutrients. Since manure-derived nutrients are not fully crop-available in the year of application (~40% of manure-N in year one, ~20% in year two; ~70% of manure-P and 80% of manure-K in year one (with no data about the availability in the consecutive years; Shapiro et al., 2021; Iqbal et al., 2022), correction factors are applied to determine the gross application needed to meet net surplus targets.

For example, the total N input to wheat (synthetic + organic + deposition + N fixation) in 2021 was ~136.7 kg N ha⁻¹. In S-1, we target a 10% reduction, lowering total input to ~123 kg N ha⁻¹ while maintaining ~93.9 kg N uptake. Manure-N input increases from ~2.2 kg ha⁻¹ yr⁻¹ in 2021 to ~33.0 kg ha⁻¹ yr⁻¹ in S-1 (equivalent to roughly 5–6 tonnes of mixed-origin manure per hectare, based on 2021 application patterns; Supplementary Table 6). This requires much better use of existing manure resources, which are currently wasted (Supplementary Fig. 10). Supplementary Text 2.1 and the data in Fig. 12 indicate that there would be sufficient manure available to meet this scenario according to current livestock numbers (see further details below). Synthetic N use drops from ~122.3 to ~77.1 kg ha⁻¹ yr⁻¹. Of the ~123.0 kg total N input ha⁻¹ yr⁻¹, only ~109.8 kg is estimated to be crop-available, accounting for 40% (~12 kg N ha⁻¹ yr⁻¹) first-year manure mineralization and 20% (~6 kg N ha⁻¹ yr⁻¹) carryover from the previous year. This leaves a small crop-available surplus (~16 kg N ha⁻¹) to offset unavoidable losses, assuming ~93.9 kg N uptake. The remaining crop-unavailable surplus (~13 kg N ha⁻¹ yr⁻¹) is likely retained in soil, contributing to soil organic matter (SOM) build-up (Boincean and Dent, 2019; Baliuk et al., 2021; Kraus et al., 2024). The same principle applies for P and K: wheat receives ~20.9 kg P ha⁻¹ yr⁻¹ (up from 11.2), with 17.8 kg crop-available P matching ~17.2 kg P removal, as well as ~26.4 kg K ha⁻¹ yr⁻¹ (up from 15.2), with 21.1 kg crop-available K matching ~20.9 kg K removal. Similar adjustments are applied for maize and sunflower (Fig.3; Supplementary Table 2-4, Fig. 27, 28).

Overall, S-1 is expected to deliver more sustainable nutrient balances compared with BaU, with significant N surplus reductions, improved near-neutral P and K balances and potential soil health

benefits. We also expect yield maintenance and reduced reactive N losses. This is supported by the meta-analysis of Ren et al. (2022a), who showed that substituting 20-50% of synthetic N with manure, on average, increased wheat yields by ~5% and reduced total N losses, with 49% lower ammonia (NH₃) volatilization in wheat, 13% reduced nitrous oxide (N₂O) emissions and ~50% less runoff N loss in maize systems. Additionally, improved fertilizer timing and placement (the '5R' practices) under S-1 may enhance nutrient use efficiency and reduce losses, although not explicitly modelled in this scenario.

Manure integration under S-1 addresses Ukraine's substantial underutilization of manure nutrients: only ~10% of total manure produced (excreted from farm animals) was applied to agricultural lands in 2021 (Supplementary Fig. 10). By adjusting manure use to crop-specific P and K needs, S-1 reduces reliance on costly imported fertilizers and improves soil health via added organic macro- (C, N, P, K) and micro-nutrients. We qualitatively assume gains in soil organic carbon under S-1, which could, over time, enhance nutrient retention and water-holding capacity, though these effects are not explicitly considered here.

S2: Enhanced efficiency fertilizers (EEFs) scenario

Scenario S-2 extends S-1 by incorporating enhanced efficiency fertilizers (EEFs), including nitrification and/or urease inhibitor-treated fertilizers (e.g., urea-based products), conventional fertilizers combined with single/dual inhibitors and/or slow-release formulations to further reduce N losses. The intended outcome is that a greater fraction of applied N is taken up by crops, allowing for either yield increases, N input reductions or both.

Meta-analyses and reviews (e.g., Thapa et al., 2016; Cantarella et al., 2018; Cui et al., 2018; Klimczyk et al., 2021; Ren et al., 2023; Matse et al., 2024) showed that nitrification inhibitors could reduce direct N₂O emissions by 30-50%, while urease inhibitors could lower NH₃ volatilization by 30-75% (on average around 54%). Also, Guardia et al. (2017a, b) and Recio et al. (2020) demonstrated that, alongside a 35–40% reduction in N₂O emissions, nitrification inhibitors, such as (i) DMPSA, (ii) DMPSA with or without urease inhibitor NBPT, and (iii) DMPP, were able to mitigate nitric oxide (NO) emissions by 76% from calcium ammonium nitrate (CAN), by more than 60% from urea, and by 60% from pig slurry, respectively. Yield responses vary: some studies reported slight gains (+5% on average) due to improved N retention (Lam et al., 2018), while others find no significant yield change but reduced N fertilizer needs for the same yield.

In this scenario, we take a conservative approach prioritizing environmental benefits. We assume that EEFs enable farmers to maintain the same yield with ~10% less N compared to S-1, due to reduced N losses. Accordingly, we reduce total N inputs by an additional 10% for wheat and maize, and by 5% for sunflower in S-2 compared with S-1, solely through reductions in synthetic N fertilizer, e.g., wheat in S-1 had a total N input of ~123 kg N ha⁻¹ yr⁻¹ (including deposition and N fixation); in S-2, this is reduced to ~111 kg N ha⁻¹ yr⁻¹ on an annual basis. Manure-N inputs remain the same as in S-1 (33 kg N ha⁻¹ yr⁻¹), while synthetic N is lowered from ~77 to ~65 kg N ha⁻¹ yr⁻¹. Inputs of P and K remain unchanged from S-1, assuming EEFs mainly affect N cycling (Brownlie et al., 2024 and references therein), and P and K management is based on nutrient removal with yield.

These adjustments bring the estimated N surpluses in S-2 closer to the acceptable minimum (9–17 kg N ha⁻¹ yr⁻¹), with only marginal surpluses of crop-available N (~3–4 kg N ha⁻¹ yr⁻¹). For wheat, with ~94 kg N ha⁻¹ yr⁻¹ uptake and ~111 kg N total input ha⁻¹ yr⁻¹, the surplus is ~17 kg N ha⁻¹ yr⁻¹, approximately 61% lower than the ~43 kg N ha⁻¹ yr⁻¹ surplus in the BaU scenario. Considering only crop-available N in S-2, the balance becomes even tighter, leaving only minimal residual mineral N in soil and/or subject to loss. Similar reductions in N surplus are seen for maize and sunflower (Fig.3; Supplementary Table 2-4, Fig. 27, 28).

These improved N balances are accompanied by increased crop uptake and further reduced N loss pathways, as EEFs stabilize fertilizer N (Thapa et al., 2016; Guardia et al., 2017a, b; Cantarella et al., 2018; Cui et al., 2018; Recio et al., 2020; Klimczyk et al., 2021; Ren et al., 2023; Matse et al., 2024). On average, the cumulative emission mitigation potential of S-2, building on S-1, is estimated to be ~77% for NH₃, ~48% for N₂O and at least ~60% for nitric oxide (NO). We expect similar reductions in Ukraine's arable systems, given the higher baseline N surpluses (and potentially N losses) under BaU practices.

By cutting N fertilizer rates and preventing rapid N transformations, S-2 is expected to significantly reduce gaseous emissions (NH₃, N₂O and NO) and nitrate leaching compared with S-0. One caveat

is that, if farmers fully adopt EEFs, they may also realize some yield gains; if yields increase, N uptake would rise and the apparent surplus could decrease further, potentially turning to zero or even a slight deficit if not offset.

Regardless, S-2 represents a system with very high nutrient use efficiencies, estimated at 85-88% for total applied N, 83-88% for total P and 79-87% for total K. These values are at the upper end of what has been documented globally and are typically only observed in best-practice or experimental settings (Claessens et al., 2024; Krauze et al., 2024).

S-3: Crop diversification with legumes scenario

Scenario S-3 adds to the measures included in S-2 by also exploring the impact of introducing N-fixing legume crops into rotations as a strategic measure to supply biological N and reduce dependency on synthetic fertilizers. While our core dataset and the above scenarios treat each crop independently (annual static balances), S-3 requires consideration of multi-year crop sequences for accuracy. Although we indicatively include numeric data for S-3 in Supplementary Table 2-4, we largely estimate its effects qualitatively, supplementing S-2.

We assume that by 2030, a significant portion of Ukraine's grain area (around 20%) could be sown with leguminous crops (e.g., soybeans, peas, beans or forage legumes), supported by appropriate policy incentives (see Actions). This affects nutrient flows in two main ways: (i) During legume cultivation, biological N fixation (BNF) provides most of the N for that crop, greatly reducing or eliminating the need for (synthetic) N fertilizer on those fields; (ii) The residual N benefit to the following crop allows for lower (synthetic) fertilizer rates on that subsequent crop.

Empirical data from Europe show that grain legumes like pea or faba bean can, on average, fix between 130–153 kg N ha⁻¹ in their biomass over the growing season each year (varied by species and conditions) (Zander et al., 2016), with a portion remaining in the soil after harvest. A review by Plaza-Bonilla et al. (2017) found that legumes contributed roughly 40–50 kg N ha⁻¹ to subsequent wheat crops on average. We adopted a conservative credit of ~50 kg N ha⁻¹ for the crop following a legume. In practice, this means that if wheat or maize follows a legume in rotation, farmers can reduce synthetic N by about 50 kg ha⁻¹ without yield loss.

In our scenario framing, we applied this concept by envisioning that 1 out of every 4 years a cereal is preceded by a legume, giving an effective annualized N credit of ~15-16 kg ha⁻¹ across the rotation. For simplicity, we incorporated this into S-3 by reducing N fertilizer inputs an additional ~15% beyond S-2 levels (which roughly equates to ~15 kg less N for cereal crops, in line with the credit). Thus, if wheat and maize received ~111 kg N ha⁻¹ yr⁻¹ in S-2, they might receive ~94 kg N ha⁻¹ yr⁻¹ in S-3, in addition to benefiting from 15 kg N ha⁻¹ yr⁻¹ available from the preceding legume crop. Sunflower, which often follows cereals, might indirectly benefit from a preceding legume's residual N as well (we roughly estimate this benefit at a lower rate of ~10 kg N ha⁻¹ yr⁻¹) (Fig.3; Supplementary Table 2).

The precise numbers are less important than the trend: S-3 pushes synthetic N inputs down to ~61% (when combined with S-2) and to 52–56% (when combined with S-1 only) below the 2021-based BaU scenario (S-0) for the studied staple crops by 2030, effectively replacing that portion with biologically fixed N. This means Ukraine's average synthetic N use in 2030 could be more than halved (61% reduction) for wheat, maize and sunflower compared to BaU practices, if crop diversification with legumes is combined with manure-enriched precision agriculture and EEF practices (Fig.3; Supplementary Table 2).

We keep yield assumptions in S-3 the same as in S-0, S-1 and S-2, while acknowledging the well-known rotation yield boost: cereals after legumes often yield more than cereals after cereals, due to improved soil conditions and pest/disease breaks (Zander et al., 2016; Geng et al., 2023).

Another benefit pertains to P: some legumes can acidify the rhizosphere and mobilize sparingly soluble P, effectively tapping into less-available soil P pools (Yu et al., 2021). While we did not quantify this, it implies that legume rotations might partially compensate for lower P fertilizer inputs by increasing P availability from the soil, at least in the short term (Jemo et al., 2006; Yang et al., 2021; Yu et al., 2021). However, some studies reported no increase in soil P availability under/ after legumes (Maltais-Landry et al., 2015).

Overall, S-3, when built on S-2, essentially achieves a near-zero balance for crop-available N at the national scale for key staple crops in rotation (Supplementary Table 2). The additional N fixed biologically is largely removed in harvest or remains in soil organic matter, with minimal losses

(Yang et al., 2024; Sharma et al., 2025). In the long term, a small fraction of fixed N may build up soil N capital, which is desirable for soil health.

It is worth noting that even in S-3, prudent management is essential: high legume proportions could risk N surpluses if residual N is not accounted for, or could deplete soil moisture in dry regions, affecting the next crop (Pang et al., 2018; Liu et al., 2024). Our scenario assumes best practices in legume management (optimal varieties, inoculation and proper termination to maximize N benefits).

The environmental upside of S-3 is significant: by reducing synthetic N fertilizer manufacturing and application, it lowers greenhouse gas and reactive N emissions, as well as upstream energy use. In-field N losses also drop further (Yang et al., 2024). A European study estimated that expanding legume area can reduce agricultural N₂O emissions by up to 40–50%, while also cutting fertilizer-associated CO₂ emissions (Zander et al., 2016).

While achieving S-3 at scale would require substantial shifts in cropping systems (and probably advanced markets for legumes), it represents a regenerative pathway aligned with EU sustainable agriculture goals.

Supporting data and calculations

The quantitative scenario values used in our analysis are summarized in Supplementary Table 2-4. This table presents, for each crop and scenario, the estimated total N/P/K inputs, synthetic fertilizer N/P/K inputs, manure N/P/K inputs, total crop-available N/P/K inputs, yield N/P/K outputs and resulting nutrient balances (surpluses/deficits) in absolute units, percent changes and cumulative balances over 2024–2030. We derived these by applying the percentage adjustments and allocation rules described above to the 2021 baseline data for S-0, S-1, S-2 and qualitative S-3.

E.g., Supplementary Table 2 shows the annual total N input to wheat dropping from 136.7 kg N ha⁻¹ in S-0 to ~123.0 kg N ha⁻¹ in S-1 (10% reduction) and to ~110.7 kg N in S-2 (10% further reduction vs S-1), with a further notional drop to ~94.1 kg N ha⁻¹ (excluding N input from the preceding legume's BNF) in S-3 (15% further reduction vs S-2). The annual P input rises from 11.2 kg ha⁻¹ (S-0) to 20.9 kg ha⁻¹ in S-1/S-2/S-3, eliminating the P deficit. Similar entries are provided for K, and for N/P/K in maize and sunflower (Supplementary Table 3-4).

We include the cumulative nutrient surpluses or deficits over 2024–2030 for the BaU (S-0) and the worst-case (S-w; based on 2023 data) projections, to illustrate the magnitude of soil nutrient stock changes if current trends persist. For example, under S-0, Ukraine's wheat would accumulate an N surplus of roughly $7 \times 42.8 \approx 300$ kg N ha⁻¹ over 2024–2030, whereas P and K would be depleted by $7 \times 6 = 42$ kg P ha⁻¹ and by $7 \times 5.6 \approx 39$ kg K ha⁻¹, respectively, in that period (Supplementary Table 2-4).

We stress that these are approximate estimates, as year-to-year yield fluctuations are not treated (for simplicity, we assumed 2021-level yields each year). Nonetheless, the cumulative figures highlight how quickly soil fertility could deteriorate in the absence of corrective action: losses of tens of kg per hectare of P and K each year could exhaust even high initial soil reserves within the coming decade(s) (Supplementary Table 3-4).

For N, cumulative surpluses directly translate to pollution potential and waste of nutrient resources. Under S-0 (BaU), a cumulative surplus of ~300 kg N ha⁻¹ for wheat, ~293 kg N ha⁻¹ for maize and ~87 kg N ha⁻¹ for sunflower across Ukraine's wheat (7.1 million ha), maize (6.7 million ha) and sunflower (5.5 million ha) areas as of 2021 would imply on the order of 4.6 million tonnes of reactive N potentially emitted or leached over 2024–2030, a portion of which would impact air quality (via NH₃ and NO), water quality (e.g., freshwater and Black Sea eutrophication) and climate (via N₂O) (Supplementary Table 2).

In contrast, enormous cumulative deficits translate to soil degradation potential. Under the extended war disruption scenario (S-w), assuming continued war-related disruptions, large cumulative deficits of N (189, 41 and 120 kg N ha⁻¹), P (320, 197 and 102 kg P ha⁻¹) and K (218, 114 and 110 kg K ha⁻¹) for wheat, maize and sunflower, respectively, across reduced crop areas in 2023 (wheat: 4.7 million ha; maize: 4.0 million ha; sunflower: 5.2 million ha) would imply on the order of 1.7 million tonnes of N, 2.8 million tonnes of P and 2.1 million tonnes of K potentially mined from Ukrainian soils between 2024 and 2030 (Supplementary Table 2-4).

Our sustainable scenarios, S-1 and S-2, drastically reduce the waste of N resources compared with S-0 (by 48–92% of crop-available N) or even eliminate the S-w deficit, while promoting SOM build-up due to longer retention and/or incorporation of crop-unavailable, manure-derived

organic nutrients in soil. Even more environmental and economic benefits are anticipated under crop diversification with legumes (S-3).

Assumptions and limitations

These scenario calculations are intended as illustrative projections, not precise estimates. We made several simplifying assumptions:

(i) **Yield constancy:** We held crop yields constant at 2021 levels in all scenarios (except acknowledging slight rotational boosts in S-3, and the worst-case war scenario S-w, based on 2023 yield levels). In reality, yields by 2030 could change due to genetic improvements, climate trends, other socio-economic factors or further war impacts (which may be similar or different to those projected in S-w). Higher yields would demand more nutrients but also result in higher uptake, possibly maintaining similar balances if managed well. Our constant-yield approach isolates the effect of management practices on nutrient balances. This is a necessary simplification, given the lack of field measurements in Ukraine for the practices proposed in the scenarios.

(ii) **No phased adoption modeling:** We did not explicitly model year-by-year transitions or adoption rates. Instead, we effectively assume that by 2030 the practices are fully adopted (a snapshot analysis). A more detailed dynamic model might, e.g., phase in manure use gradually or simulate yield dips in early years under nutrient deficits. Here, we compare end-state scenarios for clarity and simplicity.

(iii) **Simplified surplus interpretation:** Our nutrient balance framework does not detail all loss pathways. The 'surplus' in our tables represents a lump sum that could be lost as reactive N emissions, nitrate leaching and/or minor soil accumulation (unless organic- or manure-enriched fertilization is applied). When we report annual 'N surplus = 43 kg N ha⁻¹', it does not specify how much is lost (and how) versus temporarily stored in soil. For scenarios (namely S-0 and S-w), we interpret reductions in surplus as reductions in losses, which is reasonable if soil nutrient stocks are not building up. In the sustainability scenarios (S-1, S-2 and qualitative S-3), where manure-enriched fertilization is applied, we assume surpluses of crop-available N largely represent losses, while crop-unavailable fractions likely contribute to building soil nutrient stocks. In the case of actual yield increases under S-2 and S-3 (where balances are tight and may approach near zero or even slightly negative), we assume that slight deficits can be temporarily offset by mineralization of soil organic matter (until the yield gain is recorded). Once that happens, we assume decision-makers would in practice fine-tune synthetic and/or organic inputs to avoid sustained nutrient mining.

(iv) **Manure availability:** We assumed that manure could supply up to 30% of crop N needs by 2030. This is contingent on major improvements in manure management infrastructure. Currently, logistical and regulatory barriers in Ukraine make it difficult to collect and distribute manure at this scale. Our scenario presumes a concerted effort to invest in low emission practices for manure transport, storage and application, possibly via manure cooperatives or biogas digestate systems, e.g., at the commune/ district level as implemented in Romania under the EU and World Bank support (INPC, 2011, 2015, 2025). If only 15% substitution is achieved by 2030, the benefits would be proportionally smaller (though still significant). In addition, manure application rates and timing should comply with existing regulations under the EU Nitrates Directive (91/676/EEC) to minimize nitrate pollution of surface and groundwater, particularly within designated Nitrate Vulnerable Zones (Serra et al., 2024).

(v) **EEF feasibility:** The enhanced-efficiency fertilizer scenario (S-2) assumes that these products are effective under Ukrainian conditions (e.g., soil type, temperature) and that farmers can afford them. The cost of these advanced fertilizers/inhibitors could pose a barrier; however, if fertilizer prices remain high, even a ~10% saving in N use could economically justify their adoption. We did not perform a detailed economic analysis, but note that Lam et al. (2018) found the cost per kg of dry matter produced was similar for inhibitor-treated vs. regular urea when accounting for yield gains in their trial. Governmental subsidies may also help offset early-stage costs.

(vi) **Legume scenario generalization:** The crop diversification with legumes scenario (S-3) represents a broad-brush analysis of rotational effects. Actual outcomes would depend on which legume species are used, the area planted and rotation frequency. For example, soybeans remove much of the N they fix in the harvested grain, often leaving less residual N than peas or beans (Yang et al., 2024; Sharma et al., 2025). We based our assumptions on average residual N benefits from a mix of grain legumes. If farmers rely mainly on soybeans (a common, profitable legume), fertilizer savings for subsequent crops might be smaller than if using field pea (which leaves more N in

residues). A mix of grain and green-manure legumes might maximize benefits (Sharma et al., 2025); however, green manures require taking land out of cash-crop production for a season. Growing cash crops might be subsidized by government, as in neighboring Romania (INPC, 2011, 2015, 2022). Our S-3 scenario implicitly assumes the use of grain legumes, so that there is economic output as well.

*(vii) **Secondary environmental effects not modeled:** We did not model secondary environmental impacts such as changes in soil moisture or pest populations due to new practices. For example, additional manure application could increase soil moisture retention (S-1, S-2 and S-3); legumes could reduce the need for synthetic pesticides (some break disease cycles and lower N fertilizer use often attracts fewer pests due to reduced foliage density; S-3). These could provide additional benefits or introduce new management challenges beyond the nutrient-centric scope of our model. Despite these uncertainties, our multi-scenario approach provides a grounded estimate for policymakers and farmers of what level of improvement could be achievable by 2030. We anchored each adjustment in documented research findings to ensure plausibility. The results directly inform the policy recommendations in our main text, offering an evidence-based trajectory rather than qualitative speculation.*

[R1.3] (2) If the authors assert that they “provide a unique analysis of the way in which the Russo-Ukrainian war has fundamentally changed nutrient balances for Ukraine,” then the uniqueness of that analysis must be clearly demonstrated and substantiated with evidence. The analytical framework appears conventional, and the model used does not introduce any novel computational or theoretical advancements.

Response to [R1.3]. We are grateful for the reviewer’s considerations. Nevertheless, we would respectfully disagree. While the analytical framework and computational methods employed may appear conventional, the novelty and uniqueness of our study lie specifically in the context, scope, and implications of our scenario-based analysis regarding the ongoing Russo-Ukrainian war. To clearly demonstrate this uniqueness, we have now added an explicit **extended war disruption scenario (S-w)** to quantitatively project the unprecedented impact of prolonged war conditions on nutrient balances, soil health, and agricultural productivity in Ukraine (see revised text below and **Supplementary Text 2.8** cited in full under the response **[R1.2]**).

Specifically, our newly developed scenario (**S-w**) reveals a severe escalation of nutrient depletion if the war persists, projecting cumulative deficits of approximately 1.7 million tonnes N, 2.8 million tonnes P, and 2.1 million tonnes K by 2030 (**Supplementary Tables 2-4, Text 2.8** cited in full under the response **[R1.2]**). This substantial nutrient mining presents significant risks of irreversible soil degradation, particularly in Ukraine’s Chernozem soils, with likely profound consequences for both national and global food security. Our projections explicitly underscore the urgent need for intervention and proactive nutrient management to mitigate these war-induced impacts.

Thus, the uniqueness of our analysis does not stem primarily from methodological innovations, but rather from its groundbreaking, evidence-based exploration of war-induced disruptions in a globally significant agricultural region. This approach offers critical insights and timely evidence that have not previously been documented in the literature, directly contributing to discussions on sustainable recovery and resilience planning.

Nutrient management scenarios

L373-377: *The extended war disruption scenario (S-w) severely escalates nutrient depletion, projecting cumulative deficits of 1.7 Mt N, 2.8 Mt P and 2.1 Mt K for the three crop growing areas over 2024-2030 (Supplementary Table 2-4, Text 2.8). Such extreme nutrient mining risks irreversible damage to Ukraine’s Chernozem soils, depleting soil organic matter and lowering productivity, echoing degradation patterns of the 1990s^{15,26,27}. This scenario emphasizes the critical need for immediate interventions to sustain national and global food security.*

Supplementary Text 2.8: Detailed description of nutrient management scenarios

L364-373: S-w: Extended war disruption scenario

The war disruption scenario (S-w) considers the case if 2023's war-induced low fertilizer use were prolonged, with continued extreme nutrient deficits. Under 2023 conditions, wheat, maize and sunflower had negative annual N balances (around 27.0, 5.8 and 17.2 kg N ha⁻¹, respectively), annual P deficits (around 13.1, 14.8 and 5.1 kg P ha⁻¹) and K deficits (roughly 14.3, 16.2 and 9.2 kg K ha⁻¹, respectively). Projected over 2024–2030, such deficits would accumulate to roughly 40–189 kg N ha⁻¹, 35–103 kg P ha⁻¹, and 65–113 kg K ha⁻¹ being mined from soils per hectare, eventually impairing yields (Fig.3; Supplementary Table 2-4, Fig. 27, 28). Indeed, historical observations in Ukraine show that continuous nutrient deficits have led to steady declines in soil organic matter and incipient yield reductions (Chesnyak et al., 1983; Nosko et al., 1988, 1992; Lerman et al., 2008; Baliuk et al., 2009, 2021; Baliuk and Medvedev, 2012). This worst-case scenario provided the motivation for exploring the intervention scenarios described below.

L548-552: *In contrast, enormous cumulative deficits translate to soil degradation potential. Under the extended war disruption scenario (S-w), assuming continued war-related disruptions, large cumulative deficits of N (189, 41 and 120 kg N ha⁻¹), P (320, 197 and 102 kg P ha⁻¹) and K (218, 114 and 110 kg K ha⁻¹) for wheat, maize and sunflower, respectively, across reduced crop areas in 2023 (wheat: 4.7 million ha; maize: 4.0 million ha; sunflower: 5.2 million ha) would imply on the order of 1.7 million tonnes of N, 2.8 million tonnes of P and 2.1 million tonnes of K potentially mined from Ukrainian soils between 2024 and 2030 (Supplementary Table 2-4).*

[R1.4] Given the significant disruption to Ukraine's monitoring systems during the war, the biggest challenge is arguably the acquisition of reliable and comprehensive data.

Response to [R1.4]. We agree that there was indeed a short-term shock across all sectors and services immediately after the war began. However, within a week or so, this transformed into a 'new reality' of nearly normal functioning, especially for critically important services, including the State Statistics Service of Ukraine. Since then, they have collected data more frequently to inform the government about immediate changes and trends (although these data are not freely available to the public), including for key sectors such as agriculture.

Undoubtedly, data collection was disrupted in areas occupied or controlled by Russia, but it resumed once those areas were liberated. To clarify this, we have now included a paragraph discussing data availability from Ukraine's temporarily occupied areas, as well as the assumptions and thresholds used for including data in our analysis (see *Supplementary Text 3.1* and below):

L853-871: Text 3.1: Accounting for the data availability from the temporarily occupied areas of Ukraine

National National statistical data in Ukraine are typically collected by administrative unit, following the hierarchy of hromada (community or municipality), raion (district of county) and oblast (county), on a monthly and quarterly (3-month) basis. The data are then collated, processed and submitted to the central office in Kyiv for final verification, analysis and distribution/publication (SSSU, 2025). Since 2014, national statistics have not included areas that were temporarily occupied or annexed by the Russian Federation. Between 2014 and 2021, the occupied territories varied but encompassed approximately 30% of Donetsk and 50% of Luhansk counties. Statistical data from the remaining areas under Ukrainian control were collected and published in full. The Autonomous Republic of Crimea was annexed in 2014, and no official data have been available from this region since then.

Following the large-scale Russian invasion on 24th February, 2022, national statistics have ceased data collection in areas once they fall under Russian control and resume collection once those areas

are liberated. All regions remaining under Ukrainian control have continued regular reporting in accordance with established procedures since the war began.

In our analysis, we use average data (e.g., kg nutrient ha⁻¹ yr⁻¹) rather than total annual values (e.g., kg nutrient per county or country) to minimize spatial coverage bias and ensure comparability within the same county across different time periods (e.g., non-occupied vs. partially occupied). We assume that average values and trends are representative of the county as a whole. However, if a county is more than 75% occupied, either by total area or by area under the studied staple crops (whichever threshold is reached first), we exclude it from the analysis, assuming the data are not representative. E.g., in 2023, Kharkivska and Luhanska counties in their entirety, Donetsk for maize, and Chernihivska for maize and sunflower were excluded.

[R1.5] The manuscript would benefit from a more robust justification of the chosen methods and a clearer discussion of data limitations.

Response to [R1.5]. We have expanded the **Methods** section and **Supplementary Information** to comprehensively justify our methodology and transparently address data limitations. By explicitly recognizing these limitations and clearly demonstrating the measures we took to mitigate their impact, including cross-verification with multiple data sources and the use of conservative assumptions, we ensure the robustness and credibility of our findings. In both the **Methods** and **Supplementary Texts**, we provide detailed descriptions of data compilation and verification procedures, clearly justify our analytical framework, and discuss in depth the assumptions underlying our analysis, including their rationale, as well as the uncertainties and constraints associated with our data, methods and scenario projections, as detailed below:

- Limitations of the partial nutrient budget approach recommended by EUNEP, such as minimum desirable yield and the range of desirable surplus, are now mentioned in *Methods* (**L658–659**) and discussed in *Supplementary Text 3.6* (**L972–1009**).

Methods

L658-659: *The limitations of the generalized EUNEP approach have been recently outlined^{36,37} and discussed in the Supplementary Text 3.6.*

Supplementary Text 3.6: Limitations of the partial nutrient budget approach recommended by EUNEP

L972-1009: *The EU Nitrogen Expert Panel (EUNEP, 2016) outlined the framework with the key metrics, such as N balance (surplus, deficit or equilibrium), N use efficiency (target range: 50-90%) and N yield (minimal desirable threshold: 80 kg N ha⁻¹) to optimize nutrient management across the agro-food sector to both increase yields and minimize nitrogen pollution. However, these above-mentioned EUNEP reference values are generalized, that is not crop-specific nor adjusted for regional conditions (Medinets and Sutton, 2025; Mi et al., 2025). As real values may vary substantially depends on crop variety productivity, pedo-climatic conditions and management practices, the accurate region-tailored, crop-specific adjustment is needed based on field data/statistics. Mi et al. (2025) suggested region-dependent benchmarks for target N yield for wheat in China and the range of N surplus framing by unavoidable N losses (12-23 kg N ha⁻¹) and farmer's economic optima (80-85 kg N ha⁻¹). Whereas Medinets and Sutton (2025) recommended to set the upper limit of N surplus should not exceed twice the unavoidable N losses, estimated at about 10% of input, highlighting that future innovation should not see unavoidable losses as inevitable, but seek practices that reduce them further. Thus, due to absence of regional field study and/ or field-scale statistics and that in this study we analysed county mean data (as an input) without knowing standard error, we could not assess actual deviation in each county in the corresponding year for the studied crops to determine the threshold for desirable yields, we relied on the generalized EUNEP targets to indicatively identify the productivity level for each county per year. Also, following the recommendations of Medinets and Sutton (2025), we set acceptable N surplus range between 10 and 20%, which has been incorporated into the Smart Fertilizer Planner tool.*

The EU Nitrogen Expert Panel (EUNEP, 2016) proposed a framework with key metrics, such as N balance (surplus, deficit, or equilibrium), NUE (target range: 50–90%) and N yield (minimum desirable threshold: 80 kg N ha⁻¹) to optimize nutrient management across the agro-food sector, aiming to both increase yields and reduce N pollution.

However, these reference values are generalized; they are neither crop-specific nor adjusted for regional conditions (Medinets and Sutton, 2025; Mi et al., 2025). In practice, real values vary significantly depending on crop variety, productivity, pedo-climatic conditions and management practices. Therefore, accurate, region-specific, crop-adjusted benchmarks are needed, ideally based on field data/ statistics.

For example, Mi et al. (2025) proposed region-specific benchmarks for target N yields for wheat in China, alongside a range for N surplus bounded by unavoidable N losses (12–23 kg N ha⁻¹) and economic optima for farmers (80–85 kg N ha⁻¹). Medinets and Sutton (2025) further recommended that the upper limit for N surplus should not exceed twice the unavoidable losses, currently estimated at approximately 10% of inputs. They emphasized that future innovations should not treat these losses as inevitable but should instead pursue practices that minimize them.

Due to the absence of detailed regional field studies and field-scale statistics in Ukraine, and because our study used county-level mean data (without standard error estimates), we were unable to establish best practices based crop yields within each county for the studied crops in each year, or to assess actual yield deviations. As a result, we relied on the generalized EUNEP targets to indicatively assess productivity levels by county and year. Following Medinets and Sutton (2025), we also set an acceptable N surplus range of 10–20%, which has been incorporated into the Smart Fertilizer Planner tool.

Other uncertainties and limitations related to input sources, such as atmospheric N, P, K deposition rates, manure and other organic fertilizer characterization, crop residue management, as well as crop rotation practices are discussed in Supplementary Text 3.2-3.5.

- The statistical data collection procedure and data availability from Ukraine’s temporarily occupied areas are now discussed in *Supplementary Text 3.1 (L854–871)*

Supplementary Text 3.1: Accounting for the data availability from the temporarily occupied areas of Ukraine

L854-L871: National statistical data in Ukraine are typically collected by administrative unit, following the hierarchy of hromada (community or municipality), raion (district of county) and oblast (county), on a monthly and quarterly (3-month) basis. The data are then collated, processed and submitted to the central office in Kyiv for final verification, analysis and distribution/publication (SSSU, 2025). Since 2014, national statistics have not included areas that were temporarily occupied or annexed by the Russian Federation. Between 2014 and 2021, the occupied territories varied but encompassed approximately 30% of Donetsk and 50% of Luhansk counties. Statistical data from the remaining areas under Ukrainian control were collected and published in full. The Autonomous Republic of Crimea was annexed in 2014, and no official data have been available from this region since then.

Following the large-scale Russian invasion on 24th February, 2022, national statistics have ceased data collection in areas once they fall under Russian control and resume collection once those areas are liberated. All regions remaining under Ukrainian control have continued regular reporting in accordance with established procedures since the war began.

In our analysis, we use average data (e.g., kg nutrient ha⁻¹ yr⁻¹) rather than total annual values (e.g., kg nutrient per county or country) to minimize spatial coverage bias and ensure comparability within the same county across different time periods (e.g., non-occupied vs. partially occupied). We assume that average values and trends are representative of the county as a whole. However, if a county is more than 75% occupied, either by total area or by area under the studied staple crops (whichever threshold is reached first), we exclude it from the analysis, assuming the data are not representative. E.g., in 2023, Kharkivska and Luhanska counties in their entirety, Donetska for maize, and Chernihivska for maize and sunflower were excluded.

- Uncertainties in atmospheric deposition of N, P and K, including organic constituents, are discussed in *Supplementary Text 3.2 (L874–902)*.

Supplementary Text 3.2: Accounting for the total atmospheric deposition of N, P and K, including organic constituents

L874-902: *It is well shown that organic N may significantly contribute to the total N deposition, often by more than 30% (e.g., Miyazaki et al., 2010; Cape et al., 2011; Cornell, 2011; Medinets et al., 2012, 2020; Medinets, 2014; Li et al., 2023). The highest organic fraction of the total has been reported by Medinets et al. (2020) on average at around 70% for monitoring sites (cropland, garden and natural wetland) located in Odesa region (Ukraine) over the period of 2011-2019. The authors assumed that the high ratios of organic fractions in N deposition observed in this study area might be attributed to several factors: (i) wind-induced and agri-management-induced soil dust formation from Black soils with high organic matter content (Medinets et al., 2016), (ii) the influence of organic aerosols of marine origin due to proximity to the sea (Altieri et al., 2016) and (iii) potentially region-specific biogeochemical conditions that favor the emission or atmospheric formation of organic compounds. Intensive agricultural practices (e.g., fertilizer application, tillage) that generate additional soil dust containing urea, amines, macromolecules, and humic-like substances (McKenzie et al., 2016) might explain the higher absolute content of total N, including organics, in samples collected from and near managed sites. However, it remains debatable how to disentangle primary deposition from re-deposition, as well as to determine whether the sources are locally originated or transferred from nearby areas, and which of these should be included. Moreover, it cannot be ruled out that microbial conversion from dissolved inorganic to particulate and organic N occurred within the total deposition samples during the fortnightly-monthly sampling period, despite the addition of biocide thymol to inhibit microbial growth. Nonetheless, further accurate investigations into organic N (and other nutrient depositions) are necessary in the near future to determine their contributions across different regions, enhance atmospheric models and incorporate these findings into balance calculations. Based on available field data for southern Ukraine, mean annual total (organic and inorganic) atmospheric N, P and K deposition was 25.5 ± 2.2 kg N ha⁻¹ (with an estimated contribution of around 30% from dissolved inorganic N and the remainder presumably from organic N fractions), 2.5 ± 0.6 kg P ha⁻¹ (with a 56% contribution from dissolved inorganic P) and 3.1 ± 1.2 kg K ha⁻¹ (inorganic only) [values for Lower Dniester catchment, Odeska county, 2019–2021; Medinets et al., 2020, 2024; Medinets et al., unpublished data]. For example, these values accounted for 21%, 24% and 21% of the mean total N, P and K fertilizer (119 ± 6 kg N ha⁻¹, 10.5 ± 0.5 kg P ha⁻¹ and 14.8 ± 0.3 kg K ha⁻¹; **Fig. 1**) applied to wheat over the same period, respectively. It is noteworthy that the mean rates presented above for dissolved inorganic P (1.4 kg P ha⁻¹) are equal to the previously global mean for PO₄-P deposition reported by Tipping et al. (2014). Overall, these estimates, based on upscaling of measured deposition, demonstrate how atmospheric deposition of both P and K can be significant for areas receiving little or no input from fertilizers. Hence atmospheric deposition of both P and K needs to be considered alongside N (including the organic contributions) when calculating complete nutrient budgets.*

- Assumptions and uncertainties in the mean N, P and K content of manure and other applied organic fertilizers are described in *Supplementary Text 3.3 (L904–939; Supplementary Table 5, 6)*.

Supplementary Text 3.3: Accounting for mean N, P and K in manure and other applied organic fertilizer

L904-939: *Since information on manure types and other organic materials applied to fields was either unavailable in national statistics before 2018 or only partially available for 2018–2023, the data during these years indicated only total manure from agricultural animals (i.e., a sum of manure from cattle, pigs, sheep and goats), total manure from poultry (i.e., a sum of manure from chickens, ducks and geese), as well as other organic materials, including other ‘unspecified’ organic fertilizers, silt, sapropel, peat and related substances (SSSU, 2024). In both cases, no data on N, P and K content in these manures and organic fertilizers were reported in national statistics. The literature data on average N, P and K content in manure varied widely (see **Supplementary Table 5**), while actual nutrient content in manure can be even more variable, depending on factors such as (i) animal types, as each produces manure with unique nutrient profiles (e.g., poultry manure is often richer in N and P compared to cattle manure), (ii) the diet of the animals (e.g., nutrient-rich feeds yield manure with higher nutrient levels), (iii) age and growth stage (e.g., younger animals retain more nutrients for growth, resulting in lower nutrient levels in their*

manure), (iv) moisture content in manure (e.g., liquid manure generally has diluted nutrient levels compared to solid manure), and (v) storage conditions, including temperature and handling practices, which can lead to nutrient loss over time, particularly of N. Therefore, actual manure testing is recommended for precise field application, as part of the Good Agricultural Practices. To quantify N, P and K content in the present study for the 'undefined' organic fertilizers as reported in national statistics (SSSU, 2024), we:

1. Surveyed various literature sources for N, P and K content across manure types and ultimately decided to use data reported as typical for Ukraine by Pisarenko and Pisarenko (2022) (see **Supplementary Table 5**).
2. Analyzed detailed statistics from 2018–2023 to estimate the average contribution of 'total manure from agricultural animals' (79.8%), poultry manure (10.8%) and other organic fertilizers (9.4%, including a minor share (<1%) of spropel and peat) to the total organic fertilizer applied (SSSU, 2024). We used the poultry share in a sensitivity analysis to approximate the poultry proportion of total organic fertilizers applied in previous years.
3. Calculated a mean sensitivity coefficient (1.288) as the mean ratio between (a) the poultry manure share in total organic fertilizer to (b) the poultry manure share in total manure produced during 2018–2023, and applied this coefficient retrospectively to estimate poultry manure shares in total organic fertilizer for earlier years.
4. Assumed that most of the manure applied in the category 'total manure from agricultural animals,' as reported in national statistics on animal husbandry (SSSU, 2024), was likely fresh cattle manure on straw bedding due to the practical absence of manure management systems in Ukraine. We also assumed that chicken manure comprised the majority of poultry manure applied. Due the lack of specific data and for simplicity, we applied the N, P and K content for fresh cattle manure on straw bedding to other 'unspecified' organic fertilizers, which made up less than 10% of the total organic fertilizer applied in 2018–2023.
5. Finally, we computed weighted N, P and K contents in the total 'undefined' manure applied to the field for each study year based on the proportions of poultry manure (using specific NPK content for chicken manure) and other organic fertilizers, including 'total manure from agricultural animals' and 'other organic fertilizer applied' (using specific NPK content for fresh cattle manure on straw bedding) (see **Supplementary Table 6**).

- Uncertainties related to crop residue management are outlined in *Supplementary Text 3.4 (L942–957)*.

Supplementary Text 3.4: Accounting for crop residue management

L942-957: *Crop residues were mostly removed from the fields after harvest during the Soviet era, when collective farms (so-called kolkhozes) often consisted of both crop and livestock husbandry, and straw was necessary for livestock feeding and bedding (Buyanovskiy A., Burykina S., pers. comm.). From 1995 (in some regions from 1998), the livestock sector declined substantially (Supplementary Fig. 6), and crop residues were left or returned to the field, depending on the machinery used. Around the 2000s, most agro-enterprises (middle-scale >20 ha and <500 ha, and large-scale >500 ha) began leaving crop residues in the field, then either disking or mulching them. This was motivated by two reasons: (i) increasing the soil organic matter content and soil fertility, and (ii) a more pragmatic one - most of the new harvesting equipment came from Western countries and had straw chopper / spreaders and chaff spreaders. As a result, since 2010, a decrease of organic carbon loss in the soil has been observed (Bilanchyn et al., 2021; Buyanovskiy et al., 2021, 2022).*

However, some enterprises continued to remove crop residues from the field after crop harvest because they were: (i) engaged in coupled animal-crop husbandry, or (ii) produced pellets to be used as an alternative energy source (Buyanovskiy A., Burykina S., pers. comm). In our nutrient budget calculations, we assumed that most crop residues were left in the field, and that the NPK in crop residues remained in the field system. However, it is worth noting that the humification rate of these crop residues as well as N loss during this process, varies substantially depending on their management (depth and types of ploughing, disking, mulching) and pedoclimatic parameters (temperature, precipitation, soil moisture content, soil type etc.).

- Limitations due to the absence of statistical records for crop rotation practices are discussed in *Supplementary Text 3.5 (L959-969)*.

Supplementary Text 3.5: Accounting for crop rotation practice

L959-969: *In most cropping systems, a crop rotation scheme has been implemented, primarily to prevent soil-borne diseases (like root rot) and weed infestations. There is no standard crop rotation and each farmer/ agro-enterprise is guided by their own decision making. Crop rotations could include wheat - wheat - fallow, and wheat - sunflower - another crop, but there could be a total of 4 to 6 crop cultivations in a rotation cycle. However, there is also much evidence of monoculture cultivation of maize and wheat (e.g., Pinchuk et al., 2021; Mamonova et al., 2023). Ideally, a nutrient budget or balance should be calculated for a crop rotation cycle rather than on an annual basis for a certain crop, as an experienced agronomist is likely to estimate nutrient requirements considering what is left (crop residues, soil stock) from preceding crops. Data on such practices are not available, therefore for a statistical evaluation of the data we retain the comparisons at an annual level, which allows the overall trends to be seen most clearly. In this study, we focus on the general trends for the three main crops, while acknowledging that incorporating crop rotations may impact results at the site level.*

- Justification of the crop NPK removal coefficients is provided in *Methods (L660–663)*, and their variability and uncertainties are shown in *Supplementary Table 7*.

Methods

L660-663: *As information regarding NPK contents in the harvested crop yield was not available from national statistics, we made a survey to explore various sources, and finally used crop removal coefficients for each crop retrieved from the most recent and comprehensive meta-analysis conducted³⁸. We crosschecked these coefficients for NPK³⁸ with other available datasets (Supplementary Table 7).*

- Nutrient management scenarios are now outlined in *Methods (L664-671)*, with detailed justification and quantification presented in *Supplementary Text 2.8 (L339-573)*.

Methods

L664-671: *We developed five nutrient management scenarios to assess how Ukraine's crop nutrient balances might evolve by 2030 under different strategies. These include: (i) Business-as-Usual (BaU) scenario (S-0), assuming a return to 2021 practices; (ii) Extended war disruption scenario (S-w), assuming prolonged fertilizer shortages at 2023 levels; (iii) Manure-enriched precision fertilization scenario (S-1), targeting manure to supply 30% of total fertilizer N by partially substituting synthetic N; (iv) Enhanced-efficiency fertilizer (EEF) scenario building on S-1 (S-2); and (v) Legume-based diversification scenario (S-3), designed to be combined with S-2 or S-1. Sustainable nutrient management scenarios (S-1, S-2, S-3) are based on maintaining the 2021 yield level. All scenario designs, assumptions, calculations and limitations are detailed in Supplementary Text 2.8.*

Please see **Supplementary Text 2.8** in full under the response to **[R1.2]**.

- Assumptions and limitations of the proposed scenario projections are now discussed in detail in *Supplementary Text 2.8 (L574-632)*.

Supplementary Text 2.8: Detailed description of nutrient management scenarios

L575-633: Assumptions and limitations

These scenario calculations are intended as illustrative projections, not precise estimates. We made several simplifying assumptions:

(i) Yield constancy: We held crop yields constant at 2021 levels in all scenarios (except acknowledging slight rotational boosts in S-3, and the worst-case war scenario S-w, based on 2023 yield levels). In reality, yields by 2030 could change due to genetic improvements, climate trends, other socio-economic factors or further war impacts (which may be similar or different to those projected in S-w). Higher yields would demand more nutrients but also result in higher uptake, possibly maintaining similar balances if managed well. Our constant-yield approach isolates the effect of management practices on nutrient balances. This is a necessary simplification, given the lack of field measurements in Ukraine for the practices proposed in the scenarios.

(ii) **No phased adoption modeling:** We did not explicitly model year-by-year transitions or adoption rates. Instead, we effectively assume that by 2030 the practices are fully adopted (a snapshot analysis). A more detailed dynamic model might, e.g., phase in manure use gradually or simulate yield dips in early years under nutrient deficits. Here, we compare end-state scenarios for clarity and simplicity.

(iii) **Simplified surplus interpretation:** Our nutrient balance framework does not detail all loss pathways. The 'surplus' in our tables represents a lump sum that could be lost as reactive N emissions, nitrate leaching and/or minor soil accumulation (unless organic- or manure-enriched fertilization is applied). When we report annual 'N surplus = 43 kg N ha⁻¹', it does not specify how much is lost (and how) versus temporarily stored in soil. For scenarios (namely S-0 and S-w), we interpret reductions in surplus as reductions in losses, which is reasonable if soil nutrient stocks are not building up. In the sustainability scenarios (S-1, S-2 and qualitative S-3), where manure-enriched fertilization is applied, we assume surpluses of crop-available N largely represent losses, while crop-unavailable fractions likely contribute to building soil nutrient stocks. In the case of actual yield increases under S-2 and S-3 (where balances are tight and may approach near zero or even slightly negative), we assume that slight deficits can be temporarily offset by mineralization of soil organic matter (until the yield gain is recorded). Once that happens, we assume decision-makers would in practice fine-tune synthetic and/or organic inputs to avoid sustained nutrient mining.

(iv) **Manure availability:** We assumed that manure could supply up to 30% of crop N needs by 2030. This is contingent on major improvements in manure management infrastructure. Currently, logistical and regulatory barriers in Ukraine make it difficult to collect and distribute manure at this scale. Our scenario presumes a concerted effort to invest in low emission practices for manure transport, storage and application, possibly via manure cooperatives or biogas digestate systems, e.g., at the commune/ district level as implemented in Romania under the EU and World Bank support (INPC, 2011, 2015, 2025). If only 15% substitution is achieved by 2030, the benefits would be proportionally smaller (though still significant). In addition, manure application rates and timing should comply with existing regulations under the EU Nitrates Directive (91/676/EEC) to minimize nitrate pollution of surface and groundwater, particularly within designated Nitrate Vulnerable Zones (Serra et al., 2024).

(v) **EEF feasibility:** The enhanced-efficiency fertilizer scenario (S-2) assumes that these products are effective under Ukrainian conditions (e.g., soil type, temperature) and that farmers can afford them. The cost of these advanced fertilizers/inhibitors could pose a barrier; however, if fertilizer prices remain high, even a ~10% saving in N use could economically justify their adoption. We did not perform a detailed economic analysis, but note that Lam et al. (2018) found the cost per kg of dry matter produced was similar for inhibitor-treated vs. regular urea when accounting for yield gains in their trial. Governmental subsidies may also help offset early-stage costs.

(vi) **Legume scenario generalization:** The crop diversification with legumes scenario (S-3) represents a broad-brush analysis of rotational effects. Actual outcomes would depend on which legume species are used, the area planted and rotation frequency. For example, soybeans remove much of the N they fix in the harvested grain, often leaving less residual N than peas or beans (Yang et al., 2024; Sharma et al., 2025). We based our assumptions on average residual N benefits from a mix of grain legumes. If farmers rely mainly on soybeans (a common, profitable legume), fertilizer savings for subsequent crops might be smaller than if using field pea (which leaves more N in residues). A mix of grain and green-manure legumes might maximize benefits (Sharma et al., 2025); however, green manures require taking land out of cash-crop production for a season. Growing cash crops might be subsidized by government, as in neighboring Romania (INPC, 2011, 2015, 2022). Our S-3 scenario implicitly assumes the use of grain legumes, so that there is economic output as well.

(vii) **Secondary environmental effects not modeled:** We did not model secondary environmental impacts such as changes in soil moisture or pest populations due to new practices. For example, additional manure application could increase soil moisture retention (S-1, S-2 and S-3); legumes could reduce the need for synthetic pesticides (some break disease cycles and lower N fertilizer use often attracts fewer pests due to reduced foliage density; S-3). These could provide additional benefits or introduce new management challenges beyond the nutrient-centric scope of our model. Despite these uncertainties, our multi-scenario approach provides a grounded estimate for policymakers and farmers of what level of improvement could be achievable by 2030. We anchored

each adjustment in documented research findings to ensure plausibility. The results directly inform the policy recommendations in our main text, offering an evidence-based trajectory rather than qualitative speculation.

- Justification of the proposed *Smart Fertilizer Planner* and a comparison with other available tools are now revised and provided in *Supplementary Text 2.9 (L637-691; L703-739)*.

Supplementary Text 2.9: Smart Fertilizer Planner

L637-691: Principle and usage. *The proposed region-tailored, variety-specific Smart Fertilizer Planner (SFP) tool is based on a simplified nutrient balance principle, incorporating a '10-to-20%-surplus' for N (Medinets and Sutton, 2025) and near 'zero-surplus' approach for P and K. This balance is calculated between the total input, which includes synthetic and organic fertilizer applied by farmers alongside the sum of mean atmospheric inorganic N deposition and N fixation by free-living organisms (provided as a 'constant'; N deposition is updated upon new data become available in EMEP reports), and the total output, which includes crop yield (and by-product if straw is removed from the field). Both inputs and outputs are expressed in pure elemental units: kg N ha⁻¹, kg P ha⁻¹, and kg K ha⁻¹.*

It is assumed that a 10-20% N surplus, together with variability in atmospheric nutrient deposition, including unaccounted organic NPK deposited from the atmosphere, is roughly within the range of unavoidable nutrient losses and societal optimum input levels (van Grinsven et al., 2013; Medinets and Sutton, 2025). The unavoidable N losses depend on the agricultural practices and fertilizer types/rates used, and tend to approach near zero with agro-technological advances in future (Medinets and Sutton, 2025). Losses are typically higher for N and expected to be marginal for P and K (see 'Limitations and Assumptions' below). In the case of a manure-enriched fertilization scheme (where >10% of total fertilizer-derived N is applied as manure), correction factors for crop-availability of manure-derived nutrients should be made. It is assumed that crop-available manure-N is approximately 40% in the first year and 20% in the second year after application (Shapiro et al., 2021; Iqbal et al., 2022). This suggests that at least around 60% of applied manure-N is supposed to be effectively utilized by crops in a rotation. The remainder is presumed to accumulate in soil, contributing to soil organic matter (SOM) build-up, and/or be partially lost (Krause et al., 2024). Upon field application, N losses from manure-based systems vary widely and depend on manure type and treatment methods, and the use of low-emission application techniques (Brownlie et al., 2024, and references therein). Crop availability of manure-derived P and K is assumed to be approximately 70% and 80%, respectively, during the first year of application (Shapiro et al., 2021; Iqbal et al., 2022).

The tool is suggested to be available as mobile app and a downloadable spreadsheet. It would provide farmers with easy-to-use yet precise estimation for NPK fertilizer input requirements tailored to their region and crop varieties, based on data from the top-performing crops over the past 3-5 seasons.

The SFP would help calculate annual field-scale nutrient balances and estimates nutrient use efficiency after harvest using yield data. A key element of the SFP is the collection of crop yields from varieties grown in a specific region, followed by analysis of their NPK content to develop a region-tailored, variety-specific NPK removal repository. This repository should be updated whenever new varieties are introduced in the region. Another important aspect is the analysis of NPK content in organic fertilizers applied (if relevant), to at least establish an initial inventory for the region or country. Ideally, this information should be updated regularly, or, as an advanced option, it could be provided on a regular basis by organic fertilizer suppliers. This approach can be implemented at field-to-regional levels, supporting farmers and policymakers in various counties and georegions for rain-fed cropping systems (i.e., including internationally beyond Ukraine). It can also be used for irrigated crops; in this case, the NPK content applied through irrigation water (i.e., NPK concentration in irrigated water multiplied by the amount of water applied) should be added to the input category for more precise calculations. Moreover, the SFP would be adaptable to entire crop rotation cycles, making fertilizer planning more accurate and comprehensive. In this case, it would account for nutrient surpluses or deficits from preceding crops to adjust applications for subsequent ones.

The SFP is designed to support adoption of precision farming principles for crop nutrition based on crop requirements and pedoclimatic conditions at field-to-farm scale. Key elements include the so called '5R approach' to nutrient inputs (right source, rate, application time, application place and application method) and require the following actions to foster successful implementation, including (i) improving access to latest technology, information and required inputs; (ii) access to reliable soil analytical services, and the set-up of a soil fertility monitoring program for all cropland, (iii) evaluation of the existing crop-specific and soil fertility-based fertilization recommendations. Also, the nationwide field campaign is needed to inventory nutrient removal rates for the key crop varieties across georegions to increase the accuracy of the SFP (Supplementary Table 4). The establishment of regional advisory services would greatly facilitate adoption of precision fertilizer principles and implementation a wider range of Good Agricultural Practices (GAP) at the field-to-farm level. These would raise farmers' awareness of nutrient management issues by offering education and knowledge transfer, with tailored, region-specific strategies supported by simplified cost-benefit analyses and providing the required information for Smart Fertilizer Planner. Accompanying grants and loans to farmers for investment in nutrient management would catalyse change that can ultimately become self-sustaining (Supplementary Text 2.10).

L703-739: Prerequisites and requirements. *To estimate NPK removal by crop yield for specific crop varieties, a database of NPK content for currently used varieties is essential. This database should be created and regularly updated as new varieties are introduced. Such information must be accessible to farmers, either through regional advisory services, online repository or mobile app.*

The minimum requirements for successful use of the SFP tool would include:

- (i) Replacing straight urea to prevent significant nitrogen losses via NH₃ volatilization.*
- (ii) Returning crop residues to the field as a source for the next year's vegetation biomass or, alternatively, accurately calculating removed amounts and replenishing them with (organic) fertilizers.*
- (iii) Ensuring availability of NPK content data for key crop varieties.*
- (iv) Accessing appropriate soil testing to monitor exchangeable P and K concentrations.*

Further development. *The SFP is intended as a simple flexible solution that should be developed over time. With further research and advancements in technology, more precise, region-specific guidance could be developed, taking into account local pedoclimatic and socioeconomic conditions.*

Other tools available. *Also, there are some other free and commercial tools available to farmers for various purposes, they are, sometimes, specific to the georegion, but often not specific to crop varieties in a given region and, unfortunately, do not cover Eastern European countries outside the EU. However, with region-tailored, variety-specific evidence-based data, they could be complimented to SFP or adapted as a separate tool for use in Eastern European countries.*

Among them are the following:

- (i) Farm Sustainability Tool for Nutrients (FaST) – a digital tool developed under the EU CAP to help farmers create nutrient management plans. It integrates existing (mainly satellite) data with manual inputs to provide customized recommendations on crop fertilization, aiming to deliver both economic and environmental benefits. It is intended for region-specific implementation across EU countries with governmental support.*
- (ii) NUTRI-CHECK NET Platform – an EU-funded initiative offering a three-step approach—Plan, Check & Adjust, and Review—for nutrient management. It essentially serves as an inventory of various tools and services to support farmers, agronomists, researchers, and the broader agricultural industry in sharing knowledge and exploring available N, P, and K nutrient management approaches for wheat, maize, and potato across Europe.*
- (iii) Planning Land Applications of Nutrients for Efficiency and the Environment (PLANET) – a nutrient management decision support tool for use by farmers and advisers in England, Wales and*

Scotland. It supports field-level nutrient planning and helps assess and demonstrate compliance with the Nitrate Vulnerable Zone (NVZ) rules.

(iv) CAFRE Crop Nutrient Calculator – a tool developed by the College of Agriculture, Food and Rural Enterprise (CAFRE) of Northern Ireland, based on soil nutrient analysis. It helps farmers create precise fertilization plans that meet crop nutrient requirements without excess. The tool emphasizes the importance of soil pH and nutrient stocks, and promotes low-emission techniques for slurry spreading.

- Limitations and assumptions of the *Smart Fertilizer Planner* tool are discussed in *Supplementary Text 2.8 (L693–701)*.

Supplementary Text 2.9: Smart Fertilizer Planner

L693-701: Limitations and assumptions. *The proposed tool would generally assume that vegetation biomass benefits from crop residues returned to the field in previous years. In a simplest approach it may be estimated that up to 5%-surpass for N together with variation of atmospheric deposition of NPK, including unaccounted rates for organic NPK deposited (Supplementary Text 3.2), are roughly offset by unavoidable N emissions and potential NPK leaching or runoff (so long as straight urea is not used). However, the performance of the SFP may be substantially improved if evidence-based estimates for organic nutrient deposition and region-specific fertilizer-induced emission factors are available. Emissions of N are considered the primary loss pathway in black soil agroecosystems, while leaching of nitrate, phosphorus and potassium compounds is supposed to be less critical, though migration to lower soil horizons may occur (Kovda and Rozanov, 1988; Krupenikov et al., 2011; Medinets et al., 2016, 2021; Boincean and Dent, 2019).*

[R1.6] (3) The current “Action” section is overly long and includes recommendations that are fragmented and, in some cases, only tangentially related to the study’s findings. This section should be streamlined to focus on the most critical proposals that directly emerge from the research results. Moreover, the paper does not evaluate the potential impacts of the proposed actions. It would be helpful to include preliminary analyses earlier in the manuscript to assess how the recommended interventions could influence: (a) Ukraine’s NPK balance, (b) domestic food security, and (c) global agricultural supply chains. Such additions would strengthen the linkage between the study’s findings and its policy recommendations, ensuring the latter are grounded in evidence.

Response to [R1.6]. We appreciate the reviewer’s thoughtful comments and have substantially revised the **Current challenges and urgent actions** section to directly address their concerns. We have streamlined the recommendations, focusing exclusively on key proposals that are directly derived from our findings and now explicitly linked to new quantitative analyses presented in the manuscript. Specifically, we have added a dedicated **Nutrient Management Scenario** section that quantitatively evaluates how the proposed interventions could impact Ukraine’s NPK balances, domestic food security and global agricultural supply chains by 2030.

The new scenario analyses demonstrate clearly how implementing the following targeted interventions could significantly improve nutrient use efficiency, reduce nutrient surpluses/deficits and mitigate N losses by 2030:

- (i) Increasing manure-N recycling (up to 30%) to partially substitute synthetic N (and P, K) fertilizer, supported by precision fertilization using our proposed Smart Fertilizer Planner (Scenario S-1).
- (ii) Utilizing enhanced-efficiency fertilizers (EEFs), building upon Scenario S-1 (Scenario S-2).
- (iii) Expanding legumes on up to 20% of grain-producing lands in combination with S-2 or S-1 (Scenario S-3).

We have also explicitly projected outcomes under continued war disruptions without intervention (Scenario S-w), assuming prolonged fertilizer shortages at 2023 levels, and a business-as-usual (BaU, Scenario S-0) scenario, based on 2021 practices. These comprehensive analyses provided the evidence base for completely revising the **Current challenges and urgent actions** section, ensuring each recommendation is clearly grounded in our scenario results and broader findings.

Consequently, our revised policy recommendations (precision farming, manure recycling, enhanced-efficiency fertilizers and legume-based diversification) are now firmly evidence-based and directly informed by the identified needs for closing nutrient imbalances and sustaining agricultural productivity. We believe these revisions effectively strengthen the linkage between our study's findings and actionable policy recommendations, fully addressing your concerns (see revised text below).

Current challenges and urgent actions

L413-478: *To support this, our analysis of scenarios points toward the following priority measures for sustainable recovery which would also improve farm profitability:*

- *Adoption of precision fertilization principles for crop nutrition based on crop requirements and the '5R approach' (right source, rate, application time, application place and application method) at field-to-farm scale is the foundational measure underpinning all sustainable practices, as demonstrated in scenarios S-1, S-2 and S-3 (Supplementary Text 2.8, Table 2-4). This approach should be supported/guided by a simple, farmer-friendly, region-tailored, crop variety-specific Smart Fertilizer Planner tool to estimate NPK input requirements based on crop yield of top-performing farms in a region over the past 3-5 seasons (Supplementary Text 2.9). After harvest, this tool would help calculate annual field-scale nutrient balances and estimate nutrient use efficiency, to guide actions for the following season.*
- *Increasing manure-N recycling by 30% by 2030 is a top-priority, multi-benefit measure. When combined with precision fertilization, as shown in scenario S-1, it could lead to a 37% reduction in synthetic N use, partly through substitution with manure-N, while halving N surpluses, significantly mitigating direct N emissions (including from previously wasted manure), and helping to balance P and K applications compared to BaU (scenario S-0). This measure also boosts nutrient use efficiency (76–87%) and enhances soil organic matter and resilience (Supplementary Text 2.8, Table 2-4). However, successful implementation requires government support through the development and strong coordination of a national program to reintegrate animal husbandry and cropping systems. This could include manure–fodder exchanges between livestock and arable farms and increased use of processed manure, supported by investments in community-level manure management infrastructure. As a first step, a nationwide inventory of manure management practices should be conducted across georegions to improve manure collection, promote low-emission storage on livestock farms, and ensure complete recycling of manure nutrients onto croplands with minimal environmental pollution (see Supplementary Fig. 10, Text 2.1).*
- *Application of EEFs, including urease and/or nitrification inhibitors as well as slow-release formulations, is a targeted measure aimed at maximizing environmental benefits by mitigating N pollution. When combined with the previous two measures (as illustrated in scenario S-3), EEFs could significantly reduce N emissions by 48–64% and increase NUE to 85–88% (Supplementary Text 2.8, Table 2-4), e.g., replacing conventional urea with slow-release fertilizers offers a practical and immediately actionable step toward achieving these outcomes.*
- *Diversification of crop rotation with N fixing leguminous crops (including cover crops) is a conventional yet actionable strategy to reduce N fertilizer inputs and deliver broader benefits for subsequent crops, such as improved soil quality, disruption of pest/disease cycles, enhanced P availability and potential yield gains. In scenario S-3, we showed that when combined with manure-enriched precision fertilization (S-1) and/or EEFs (S-2), this approach can achieve a substantial 52–61% reduction in synthetic N use relative to the BaU (S-0) scenario by 2030. Although potentially less profitable for farmers in the short term, practices such as cover cropping for green manure are likely to require subsidies to support widespread adoption (Supplementary Text 2.8, Table 2).*

In addition, establishment of regional advisory services to support the implementation of the outlined measures and broader Good Agricultural Practices (GAP) at the field-to-farm level should be incorporated into an Integrated Nutrient Management Plan for Ukraine. These would raise farmers' awareness of nutrient management issues by offering education and knowledge transfer, with tailored, region-specific strategies supported by simplified cost-benefit analyses and providing the required information for Smart Fertilizer Planner. Accompanying grants and loans to farmers for investment in nutrient management would catalyse change that can ultimately become self-sustaining (Supplementary Text 2.10).

Developing such a plan, evidence-backed by our scenario analysis, should not wait until the war is over. While the war makes conditions harder, all the measures listed could already start with appropriate investment. Indeed, there is now an urgent need for strengthening international action to support Ukrainian agriculture and prevent further depletion of precious nutrients with the accompanying soil degradation, while recognizing that this must be the start of a long-term commitment to supporting a transition to sustainable nutrient management. We propose that an Integrated Nutrient Management Plan for Ukraine be incorporated into the agriculture sector recovery under the Ukraine Recovery and Reconstruction (URR) budget, which currently accounts for 10.5% of the total planned URR budget of USD 524 billion³⁰. We also recommend prioritizing agro-food system resilience as a key focus within the URR framework.

We have also made more clear the implications of study in relation to the reviewer's interest in (b) domestic food security and (c) global agricultural supply chains. Although our study is not specifically focused on these questions, it is clearly relevant:

Implications for food security

L232-254: *The evolving nutrient management landscape in Ukraine holds profound implications not only for national food security but also for the stability of the global food supply chain. As one of the world's top exporters of wheat, maize and sunflower oil, Ukraine plays a critical role in feeding over 400 million people worldwide, particularly in import-dependent regions^{20,21} (Supplementary Fig. 2-4, Text 1.1). However, the ongoing war has disrupted fertilizer supply chains and reduced agricultural inputs, aggravating imbalances of N, P and K (Fig 1; Supplementary Fig. 8, 26). These imbalances are marked by spatial and crop-type asymmetry in N (surpluses in some areas, and deficits in others) and persistent deficits in P and K (Fig. 2; Supplementary Fig. 26). The resulting degradation of Ukraine's Chernozem soils (Supplementary Fig. 14, Text 2.2), is significant as these store about 7% of the world's soil carbon and underpin around 90% of the country's agricultural output^{14,22}. Without immediate intervention, this trend could trigger substantial losses of soil organic matter, especially in the regions affected by the war, risking long-term productivity declines for key crops, such as wheat and maize²³.*

Given that Ukrainian grain exports traditionally support food security in regions such as North Africa and the Middle East (Supplementary Fig. 3), disruptions in crop output could exacerbate hunger and price volatility worldwide^{20,21}. The ripple effects are global: disruptions in Ukrainian grain exports during the early months of the war contributed to a sharp rise in global food prices, with the FAO Food Price Index reaching its highest level on record in 2022, over 14% higher than the previous year²⁴. This surge intensified global food insecurity, contributing to a wider hunger crisis with a global estimate of 691-783 million people affected in 2022²⁵. Moreover, the under-utilization of manure, caused by the decoupling of livestock and crop systems, represents a missed opportunity for circular nutrient economy practices (Supplementary Fig. 10-13, Text 2.1) that could reduce dependence on synthetic fertilizers while lowering reactive N and greenhouse gas emissions. Addressing these challenges requires improved access to fertilizers and long-term investments in integrated nutrient management strategies. Strengthening Ukraine's nutrient resilience is not merely a national priority but a global imperative, when considering the global food security and climate interactions.

[R1.7] (4) While the paper identifies several key trends through data analysis, it lacks in-depth investigation into the underlying drivers or mechanisms. As a result, the recommendations are overly general. For example, the study notes that fertilizer application rates and crop yields remained relatively stable during the first two years of the Russia-Ukraine war but experienced a sharp decline in 2023—with fertilizer use dropping by 37–54% compared to 2021. However, the manuscript does not explore the causes behind this delayed impact. A deeper analysis of these drivers would enhance the explanatory power and practical value of the study.

Response to [R1.7]. With respect, we realise that it is not possible to cover all points in depth, especially

considering the intent of our manuscript as a ‘short communication’. However, to address this request, we have now expanded our discussion on the delayed impact of the Russia-Ukraine war on fertilizer application and crop yields, providing clear explanations for observed trends (L116-124). We would like to remind that the war started in 2022, and we have covered two years of the war: 2022 (the 1st one) and 2023 (the 2nd one). Additionally, we have clarified the historical context of soil mineralization processes, highlighting their role in sustaining yields during periods of nutrient deficits, particularly in the 1990–2000 interval (L75-81). We believe these additions substantially enhance the manuscript's explanatory power and practical relevance.

L216-224: *Following the Russian invasion of Ukraine on 24 February 2022, slight decreases during 2022 were observed in fertilizer application and crop production. This limited immediate impact could be attributed to the resilience of fertilizer and agrochemical supply chains, which benefited from advanced planning and restocking routines typically completed by late winter/ early spring. Consequently, fertilizers for winter/ spring crops contracted in 2021 were largely accumulated in warehouses in Ukraine and/or even delivered to farmers despite the onset of hostilities. However, the impact of the war was larger in 2023. Total fertilizer applications decreased by 37-54% for the three crops compared with 2021, while utilized agro-area decreased by 22-34%. Despite this, favourable weather conditions, enhanced soil mineralization and, potentially, residual soil nutrients from preceding crops collectively contributed to relatively high yields in 2023, comparable to those in 2021.*

L75-81: *Following the dramatic reduction in total fertilizer use after 1990, NPK withdrawal in harvested wheat, maize and sunflower decreased only modestly during the first ten years. Fertilizer use and NPK withdrawal in harvested crops increased again during the 2000-2021 period (Fig. 1). We attribute the modest initial decline in yields under NPK input deficits during 1990-2000 to compensation by soil nutrient mining, i.e. enhanced mineralization of organic matter reserves inherent to the rich black soils (Chernozems)^{8,15}. Persistent nutrient-deficient soil management practices led to continuous depletion of soil organic matter (SOM) across Ukraine, resulting in a measurable 5-6% reduction in SOM content over that decade (Supplementary Fig. 14, Text 2.2-2.3).*

Reviewer #2 (Remarks to the Author):

[R1.8] Thank you to the authors for the thorough reply to the reviewer questions. I feel that all my initial concerns have been addressed and that this paper is ready for publication.

Response to [R1.8]. We sincerely thank the reviewer for their positive feedback and for acknowledging the revisions made in response to your earlier comments. We greatly appreciate their thoughtful input throughout the review process, which helped us improve the clarity and impact of the manuscript. We are pleased that the revised version meets their expectations.

Reviewer #3 (Remarks to the Author):

[R1.9] Thanks to the authors who have addressed satisfyingly and to the extend possible the points raised.

Response to [R1.9]. We very much thank the reviewer for their constructive comments and for recognizing our efforts to address concerns. Their suggestions were invaluable in helping us enhance the manuscript's rigor and relevance. We are grateful for their support and are glad that the revised version satisfactorily meets their expectations.

Detailed response from the authors to comments of the peer reviewers

We thank the editor and all reviewers for their constructive comments on our submitted manuscript, which have enabled us to strengthen the submission. We are grateful for the reviewer 1 support and are glad that the revised version satisfactorily meets their expectations. We believe that this paper is timely and of high importance for both global and national scales.

Reviewer #1 (Remarks to the Author):

No further comment or suggestion.

Response. We sincerely thank the reviewer for their positive feedback and for acknowledging the revisions made in response to your earlier comments. We greatly appreciate their thoughtful input throughout the review process, which helped us improve the clarity and impact of the manuscript. We are pleased that the revised version meets their expectations and is accepted for publication.